# Wireless ear EEG to monitor drowsiness

Ryan Kaveh [1,2] ✉, Carolyn Schwendeman[1,2] ✉, Leslie Pu[1], Ana C. Arias[1] & Rikky Muller [1] ✉

Neural wearables can enable life-saving drowsiness and health monitoring for pilots and drivers. While existing in-cabin sensors may provide alerts, wearables can enable monitoring across more environments. Current neural wearables are promising but most require wet-electrodes and bulky electronics. This work showcases in-ear, dry-electrode earpieces used to monitor drowsiness with compact hardware. The employed system integrates additive-manufacturing for dry, user-generic earpieces, existing wireless electronics, and offline classification algorithms. Thirty-five hours of electrophysiological data were recorded across nine subjects performing drowsiness-inducing tasks. Three classifier models were trained with user-specific, leave-one-trial-out, and leave-one-user-out splits. The support-vector-machine classifier achieved an accuracy of 93.2% while evaluating users it has seen before and 93.3% when evaluating a never-before-seen user. These results demonstrate wireless, dry, user-generic earpieces used to classify drowsiness with comparable accuracies to existing state-of-the-art, wet electrode in-ear and scalp systems. Further, this work illustrates the feasibility of population-trained classification in future electrophysiological applications.

Drowsiness and fatigue while operating heavy machinery can be life-threatening. It is estimated that over 16.5% of fatal vehicle accidents in the United States include a drowsy driver resulting in over 8000 deaths and $109 billion in damages[1–3]. In addition to private and commercial (trucking) accidents, the National Safety Council has also cited drowsiness as the most critical hazard in construction and mining. While these deaths may be prevented with common risk assessments, fatigued individuals are often unable to recognize the full extent of their impairment before it is too late[4]. Drowsiness monitoring solutions use camera-based eye-tracking, steering trajectory sensors, or electrophysiological recording devices[5–7]. While they can be a good fit in automotive scenarios, eye tracking is obscured by sunglasses and other obstructions while steering sensors can be susceptible to false alarms on rough roads. User-centered recording modalities such as body-worn cameras, photoplethysmography (PPG), electrodermal activity, electrocardiography (ECG), electrooculography (EOG), and electroencephalography (EEG) are becoming increasingly popular because they are highly portable and adaptable to professional work environments[8–11]. These modalities have been incorporated into multiple form-factors such as eye-tracking glasses[12], PPG/ExG

tracking helmets[7], and in-ear ExG sensors[13,14]. Of these methods, ExG generally achieves the highest drowsiness detection accuracies[15].

Surface EEG is a safe, non-invasive method of monitoring the brain's electrical activity from the scalp. Clinically, the most prevalent use of EEG is the monitoring and diagnosis of stereotyped neurological disorders related to sleep and epilepsy. These clinical systems generally use large, scalp-based, gold (Au) and silver/silver chloride (Ag/AgCl) electrode arrays[16–18]. Au forms a capacitive interface due to its inert nature, while Ag/AgCl forms a faradaic interface between Ag and skin. The AgCl is a slightly soluble salt that quickly saturates the skin and forms a stable electrode-skin interface. To maintain a low-impedance electrode-skin interface, contact is improved with skin preparation from an overseeing technician. While suitable for occasional, short-term monitoring, existing wet electrode arrays tend to be large and delicate for everyday use. Additionally, prolonged use of devices that require skin abrasion can result in skin irritation and lesions, further limiting their long-term use[19,20]. To promote use outside the lab and simplify clinical measurements, recent wearable EEG monitoring systems have focused on using smaller form-factor wet electrode arrays (e.g., cEEGgrid)[21] and dry electrodes that eliminate the

[1]University of California Berkeley, Berkeley, CA 94708, USA. [2]These authors contributed equally: Ryan Kaveh, Carolyn Schwendeman.
✉e-mail: ryankaveh@berkeley.edu; cschwendeman@berkeley.edu; rikky@berkeley.edu

use of hydrogels, integrating electronics and electrodes into a headset form factor, and software packages that allow for use in more everyday applications. The improved wet electrode systems (e.g., the cEEG grid) can provide unobtrusive EEG monitoring for 7+ h, but still requires hydrogel application (limiting day-to-day use). Dry electrode systems for research (e.g., CGX systems and Emotiv), commercial (e.g., Muse headband and Neurosity), and hobbyist (e.g., OpenBCI and Brainbit) have similarly demonstrated impressive EEG recordings of spontaneous and evoked neural signals and enabled disease monitoring, brain-computer interfaces (BCIs) and meditation guidance. As these commercial systems' popularity increases, more and more wireless EEG systems are being developed and deployed across different environments[22–25]. The least cumbersome systems employ dry electrodes that minimize set-up time but generally still require skin cleaning and electrode surface treatments. Furthermore, the associated software packages require training to use[23,24]. Lastly, headset electronics are better suited for research and clinical environments as opposed to public, everyday use.

Discreet, multi-channel EEG recordings from inside the ear canal have been demonstrated[26–28] with recent advancements focusing on earpiece design, electrode materials, and multi-sensor arrays. The ear canal is an ideal sensor location due to its inherent mechanical stability and wealth of potential recording modalities. In-ear sensors and electrodes are well situated to record temporal lobe activity, blood oxygen saturation, head movement, and masseter muscle activity making it ideal for multi-modal sensing if high spatial coverage is not required[29,30]. While some applications may treat muscle activity or ear canal deformation as interference signals, these signals can be useful for other general ExG workloads. It is also important to note that in and around-the-ear EEG is inherently limited in gathering spatially encoded brain-activity relative to broader scalp arrays[27–31]. Many successful designs have leveraged hydrogel coated on flex-pcb arrays or user-customized earpieces to record ExG features such as EOG, low-frequency EEG (1–30 Hz), and evoked potentials (40–80 Hz)[26–28,32,33]. These wet-electrode based, custom earpiece systems established the feasibility of in-ear monitoring for attention monitoring, seizure monitoring, whole night sleep monitoring, and sleep stage classification[34–37]. Due to their user customized approach, earpieces require a case-by-case integration schemes to minimize earpiece volume resulting in variable electrode positioning. The required skin-preparation and hydrogel also can lead the conductive bridging between electrodes, limit-user-comfort, and reduced electrode lifetime[38]. The next step to more scalable deployment of in-ear ExG recordings would be the utilization of one-size-fits-most (user-generic) earpiece designs, dry electrodes, wireless electronics, and electrode materials that do not require maintenance.

Recent user-generic earpieces equipped with wet electrodes, dry electrodes[39–42], PPG, and/or chemical sensors have achieved high degrees of accuracy for brain-state and activity classification[39,40,43–46]. Additionally, dry-electrode based in-ear ExG have recorded low frequency neural rhythms, evoked potentials, and EOG comparable to wet-electrode. While potentially more susceptible to noise due to higher electrode-skin impedance (ESI) interfaces[47], dry electrodes eliminate the use of hydrogel, simplify the earpiece application process, and can improve user comfort. To achieve a middle ground between comfort and low ESI, state-of-the-art dry electrodes employ a wide range of solutions ranging from exotic materials, conductive composites, capacitive interfaces, solid-gels, and high-surface area 3D electrodes (microneedles, fingers, and nanowires)[20,40,41,48–56]. PEDOT:PSS and IrO$_3$ are commonly used in the small-scale production of rigid electrodes due to their superior conductivity and faradaic interfaces[57–59]. Both materials promote charge transfer by leveraging doped surfaces and high effective surface areas. Conductive, flexible composites, such as silvered-glass silicone and carbon-infused

silicone, are not as conductive as PEDOT:PSS and IrO$_3$ but offer significantly greater comfort. Conductive composites are made from polymers or elastomers that can be molded into arbitrary shapes for anatomically fit electrodes and use added conductive particles to achieve a desirable ESI. The more conductive particles that are added will ultimately limit polymer cross-linking and may lead to cracking over time[60]. The clinical and industry standard materials are silver/silver chloride (Ag/AgCl) and gold due to their cost, biocompatibility, and electrical properties. Ag/AgCl can be painted on 3D electrodes to form consistent, faradaic, low-impedance interface through hair and grime. Furthermore, Ag/AgCl is also popular for consumable electrodes since the conductive particles deplete over time[61]. Gold electrodes are more inert, can be repeatedly reused, and form a capacitive interface that is not reliant on added conductive ions. While potentially more susceptible to motion artifacts and interference, gold's lifetime and chemical properties make it ideal for long-lasting ExG recording systems. Most commercial wearables and existing in-ear ExG systems use Ag/AgCl, Au, or conductive composite electrodes[24,62–64].

Electrodes are just one piece of signal acquisition. Neural recording hardware is required to digitize neural signals and transmit them to a processing unit/base-station for offline processing. Neural recording hardware for more consumer-facing products tend to be tailor-made with low bandwidth, noise, and power specifications[65–67]. These devices usually have bandwidths around 100 Hz and can achieve ultra-lower power operation (<100 μW[67]). Research focused devices, however, utilizing high resolution and bandwidth hardware enables greater investigation outside the original project description. Such versatile systems generally support higher channel counts (16–64+), commercial wireless protocols (bluetooth or Wi-Fi), higher sampling rates (500–1000 Hz), and can take advantage of different signal modalities (e.g., EMG) at the cost of higher power (>50 mW)[42,46,68]. Low-noise and high-resolution systems allows for greater flexibility, repeated interpretable signal processing (frequency analysis, time-domain averaging, etc.) and algorithm development to illuminate different feature classes, mitigate interference, and discover new potential applications. Such systems have been used to build brain-machine interfaces with P300 responses and steady-state evoked potentials[27,29,34,69,70]. When adapting existing electronics for use with wearable dry electrodes, increased ESI, system noise, and interference susceptibility bear important considerations for power requirements and any downstream machine learning algorithm[71,72]. Employing versatile, higher power electronics with more interpretable, light weight classical algorithms (e.g., logistic regression, support vector machines, random forest) is an important first step for future sensor and power optimizations. To this effect, this work uses an existing, high channel count, high bandwidth system to enable studying the relationship between the employed ExG electrode technology and drowsiness detection.

In addition to system optimisation, the choice of machine learning algorithm determines system functionality from the perspective of training, data, and processing requirements. Every-day ExG systems would ideally work out of the box, improve over time, and continue to provide feedback when wireless connectivity is poor and there is unreliable access to large processing power (construction sites, planes, and trucks). Classical algorithms such as logistic regression, SVMs, and random forest have demonstrated impressive success in classifying neural signals with limited datasets[15,25,73,74]. Neural network-based algorithms have also achieved impressive results[75–77], and are good candidates for further research. Neural network-based algorithms, on average, require more training data than SVMs, logistic regression, and random forest, making them difficult to work with on smaller data sets. Furthermore, interpretable algorithms such as logistic regression and SVMs enable greater visibility into which types of features have sufficient SNR for classification and could potentially be applied to

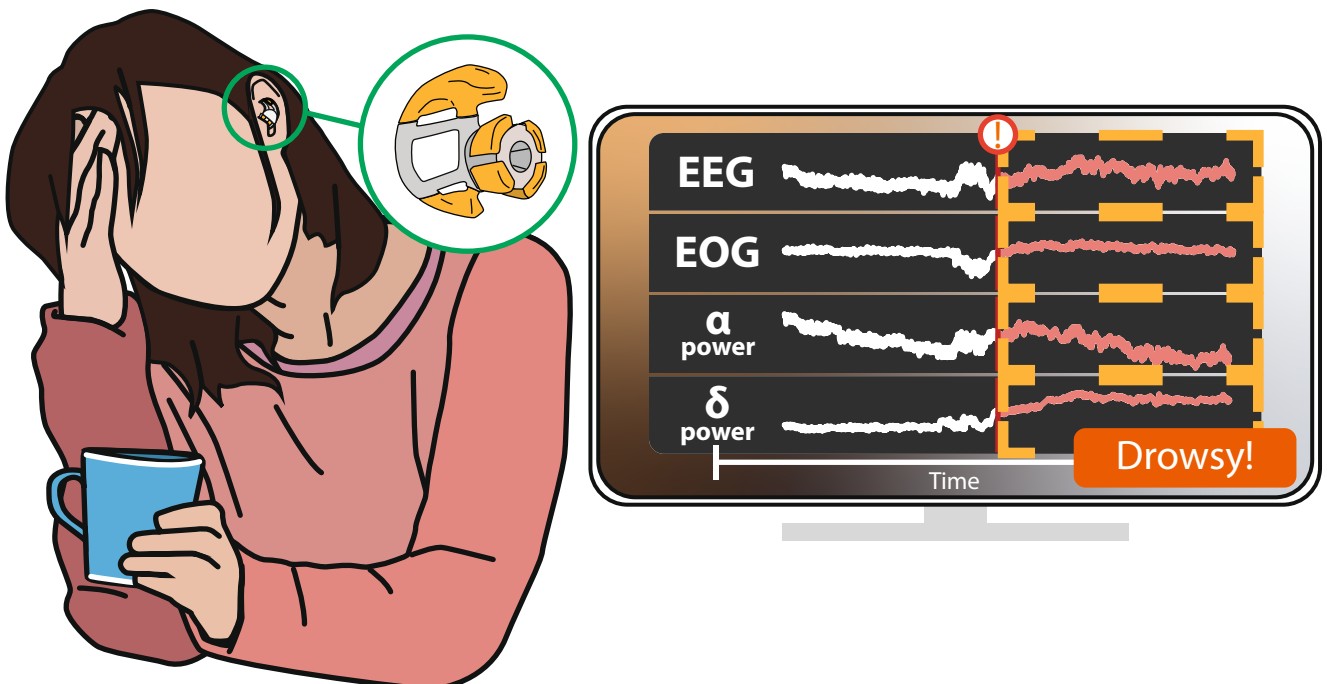

**Fig. 1 | Envisioned Ear ExG Wearable.** Envisioned systems could be discreetly worn throughout the day to comfortably record neural signals from inside the ear canal, perform drowsiness detection, and provide feedback.

different applications. Lastly, algorithms such as SVMs, logistic regression, and random forest generally require less processing power than similarly performing neural net or perceptron-based architectures, making them ideal for low-power, edge-based deployments on existing microcontrollers. Additionally, while existing in-ear ExG BCIs have achieved high classification accuracies with user-specific training and validation[35,43,76,78,79], ideal in-ear ExG wearables would leverage pre-trained algorithms so never-before-seen users can use these devices without time-consuming training. This user-generic classification has been explored in scalp-based drowsiness monitoring with great success but not yet with in-ear ExG[15].

This project is the first integration and demonstration of wireless, dry-electrode in-ear ExG sensors used for drowsiness classification. To this effect, a novel in-ear EEG sensor manufacturing method coupled to a pre-existing wireless data acquisition platform is presented and verified with open-source machine learning classification on 9-subjects. A fabrication process for dry, gold-plated electrodes suitable for repeated, comfortable, low-impedance earpieces is introduced and tested over the course of months of electrode use. This electrode technology provides a unique method for the rapid prototyping of reusable, Au electrodes that remain stable over 12 months of use. These electrodes can replace existing solutions that rely on shorter-lifespan Ag/AgCl electrodes or expensive materials such as platinum or IrO3. The earpieces are then coupled with wireless, discreet electronics capable of taking uninterrupted, low-noise neural measurements for over 40 h[46] to form a wearable, in-ear ExG system. The resulting Ear ExG BCI is then demonstrated with a nine-subject drowsiness monitoring study. Low-complexity temporal and spectral features are extracted from the recorded ExG data and used to train multiple, offline machine learning models for automated drowsiness detection. The best-performing model utilizing a support vector machine achieved an average drowsy-event detection accuracy of 93.2% when evaluating on users it has seen before and 93.3% when evaluating never-before-seen users. This system and its use of offline classifiers lay the groundwork for future, discreet, fully wireless, long term, longitudinal brain monitoring (Fig. 1).

## Results: ear ExG drowsiness monitoring platform
### Modular electrode design, fabrication, and assembly

**Earpiece design.** Easy-to-use neural wearables require a user-generic earpiece and electrode scheme designed for recording across multiple demographics and for comfortable, long-term wear. To achieve these requirements, electrode and earpiece designs were derived from refs. 46,80 and resulted in a small, medium, and large size of a single design with modular electrodes. Electrodes are positioned near the ear canal such that they do not pass the isthmus of the ear canal, which tends to develop a corkscrew shape as individuals age. This earpiece is designed to account for these age-related changes. Previous studies[30,41], have highlighted high value electrode locations that minimize channel-to-channel correlation while maximizing mechanical stability. To also maximize electrode surface area across different individuals, small, medium, and large sized earpieces were designed with slightly differing electrode sizes. The final "medium-sized" earpiece is comprised of four 60 mm² electrodes inside the ear canal and two 3 cm² electrodes on the ear's concha cymba and concha cavity (Fig. 2a). The in-ear electrodes are cantilevers that apply gentle outward pressure to achieve lower ESI over previous iterations (370 kΩ to 120 kΩ at 50 Hz[46]) and improve mechanical stability. The out-ear electrodes act as fiducial guideposts to ensure the electrodes contact the same surface with each wear. Furthermore, electrodes outside the ear are good reference and ground candidates due to their increased distance from the brain or any muscle. To improve the earpiece assembly and further increase comfort over[46], a soft earpiece body with a manifold in-ear design was 3D printed with a clear methacrylate photopolymer (Fig. 2a). Each rigid electrode is attached to this soft, elastic substrate and moves independently from the other electrodes to fit in a subject's ear (Fig. 2b). This new, modular assembly properly demonstrates the capabilities of the manifold earpiece fabrication process.

**Electrode fabrication.** A low-cost, fully electroless plating process was developed to enable rapid prototyping of arbitrary shaped electrophysiological sensors. Electrodes were 3D printed with a clear methacrylate polymer (Fig. 2c) and sandblasted to increase surface

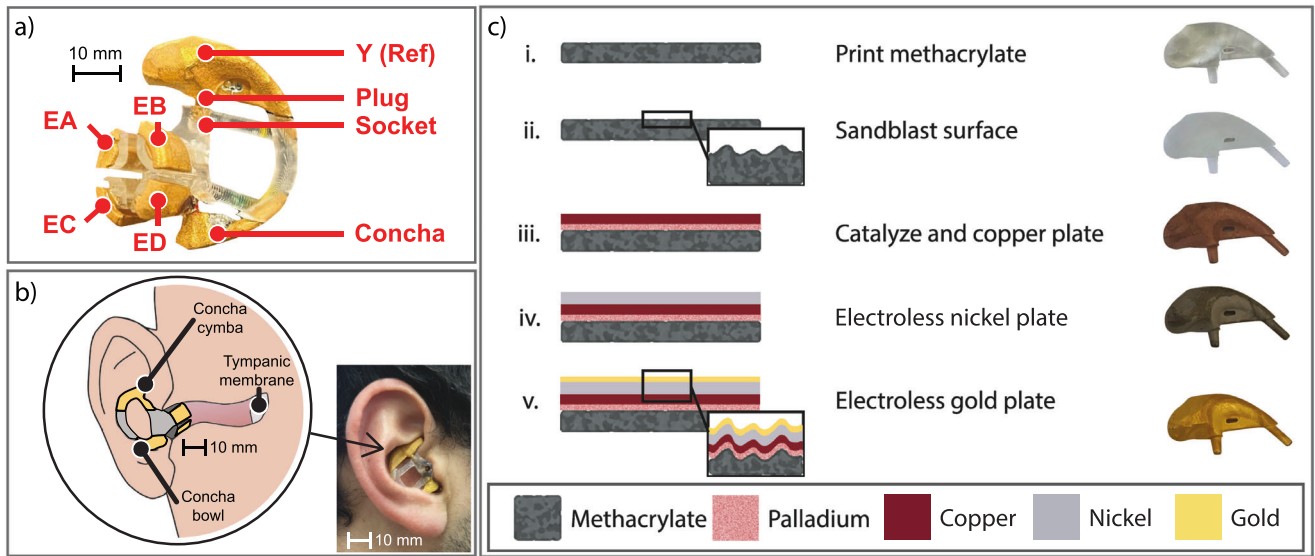

**Fig. 2 | Earpiece assembly, fit, and manufacturing process. a** The final earpieces are composed of four in-ear electrodes and two out-ear electrodes. Manifold 3D-printed earpieces are assembled by plugging rigid, gold-plated earpieces into a soft, flexible skeleton. **b** The out-ear electrodes press against the ear's concha cymba and concha bowl, while the in-ear electrodes contact the ear canal's aperture. In-ear electrodes only enter the first 10 mm of the ear canal. **c** Diagram and photographs of electrode fabrication: i) Electrodes are 3D printed or molded. ii) The bare electrodes are sandblasted and cleaned. iii) The electrodes are electroless copper plated via exposure to surfactant, catalyst, and copper sulfate solutions in sequence. iv) A nickel layer is electroless plated. v) A final gold layer is electroless deposited.

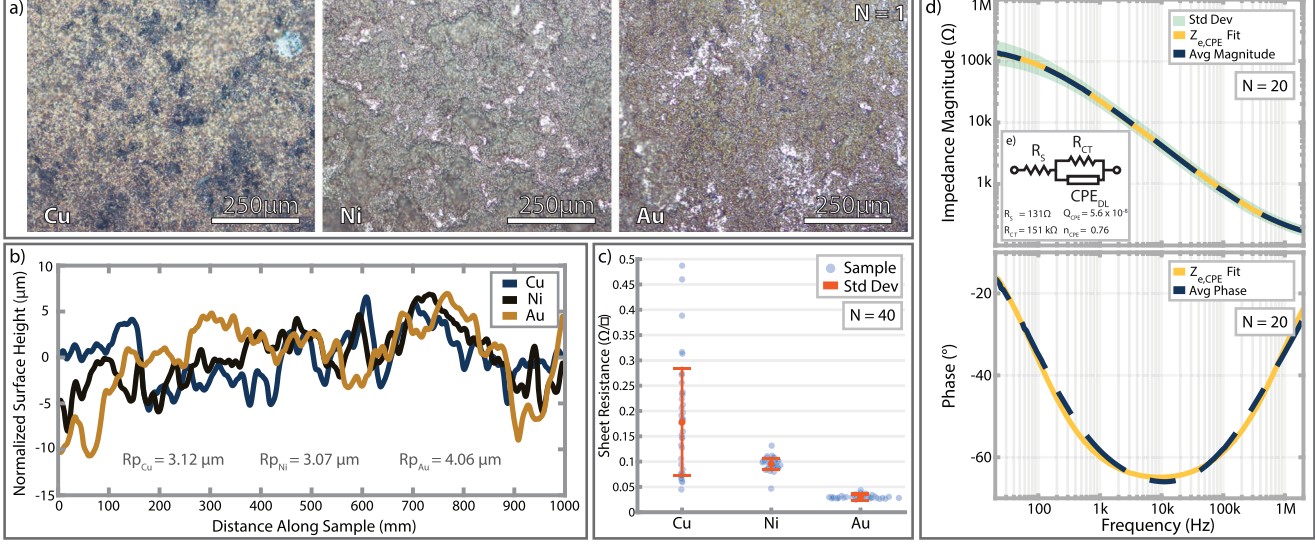

**Fig. 3 | Plated surface characterization. a** Representative light microscopy images of plated surfaces showcasing the roughness resulting from sandblasting. **b** Stylus Profilometer measurements of a flat sample after each plating step. **c** Absolute sheet resistance measurements, mean (red circle), and standard deviation (error bars) immediately after plating. **d** In-ear electrode-skin impedance magnitude, phase, and magnitude fit. Standard deviation of electrode magnitude shown in shaded green region. **e** Constant phase element electrode model used for fitting.

roughness. Samples were then submersed in different catalyst baths to develop copper, nickel, and gold metal layers. Lastly, tinned copper wires are soldered directly to the electrode surface for integration with the neural recording front end. This plating process is expanded on[45,81], with the addition of a nickel layer that limits grain-boundary diffusion of copper and significantly extends electrode lifetime[81–83]. Furthermore, the nickel-plating step removes the need for repeated electroless palladium plating and the overall number of fabrication steps. While other in-ear electrodes use expensive materials like $IrO_3$ or hydrogels[39,40], this improved layer stack-up (Cu, Ni, Au) is reminiscent of printed-circuit-board fabrication and enables similar levels of scale for electrode prototyping. The final surface contains at least 0.5 µm of copper, 0.5 µm of nickel, and 0.25 µm of gold and is suitable for dry electrode recording.

**Plating process characterization**
**Material acid dip tests and tape tests.** The final electrode surfaces were physically and chemically robust. Kapton tape was applied around the entire electrode surface and then removed. No visible gold, nickel, or copper was removed with the tape indicating strong adhesion to the methacrylate substrate[81,84]. Electrode samples were also dipped in nitric acid baths to test the porosity and continuity of the gold surface. While concentrated and dilute nitric acid will readily dissolve copper and nickel, respectively, neither will etch gold. No

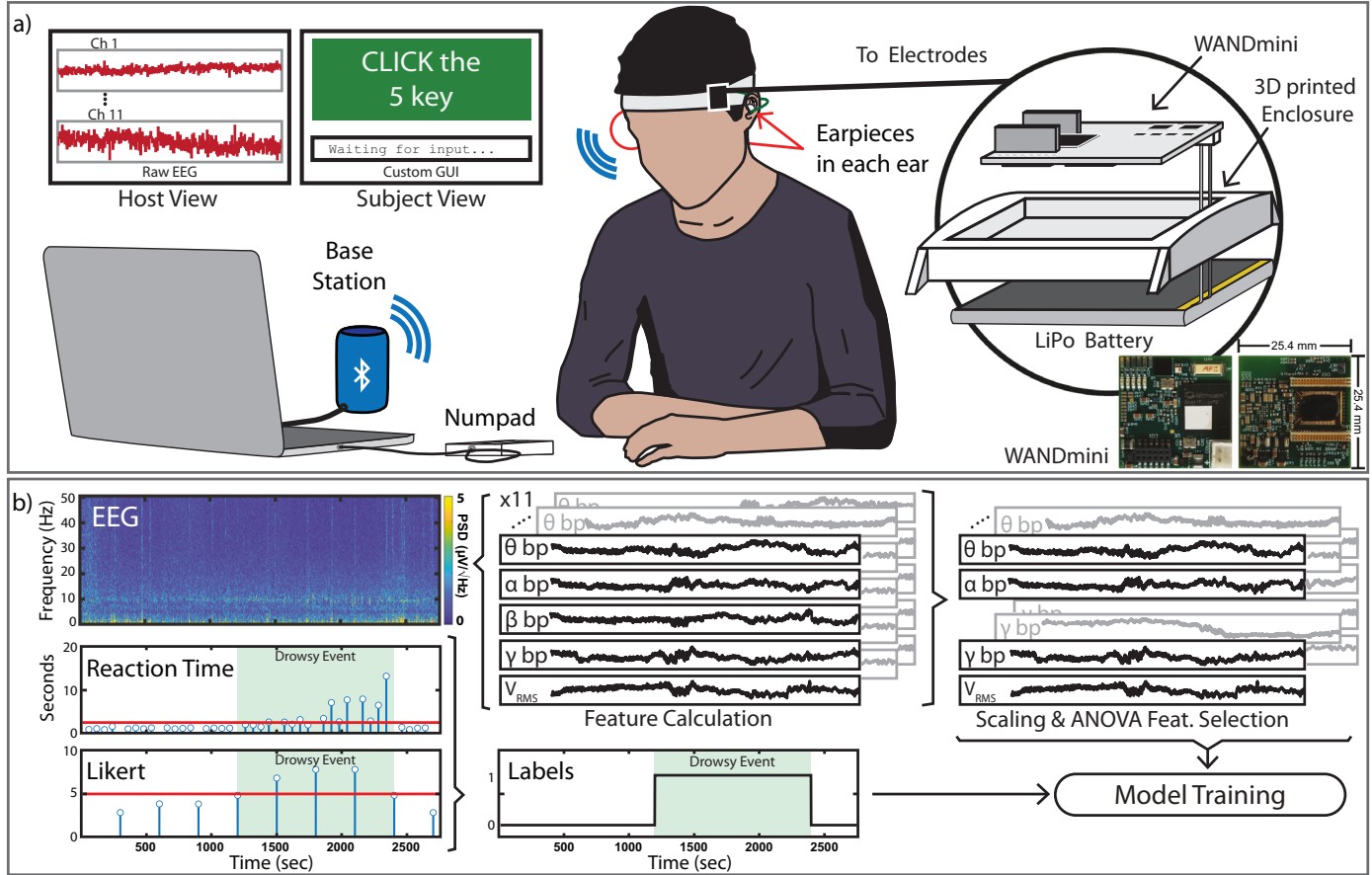

**Fig. 4 | Experimental setup, recordings, and labeling scheme. a** Subjects sit beside a laptop displaying a basic reaction time measuring game. A head-worn WANDmini, secured in a 3D-printed enclosure, records and transmits ExG from contralaterally worn earpieces to a base station via BLE while the subject plays the game. All captured ExG can be live plotted for the trial overseer while the game records subject's reaction times and Likert survey responses. **b** Recorded ExG, reaction times, and Likert items are used to generate features and labels for a brain-state classifier. Drowsy events, shaded in green, are determined when a subject's reaction time and Likert response cross a drowsiness threshold that is determined per subject. Using both the reaction time and Likert scores enables robust label creation that is agnostic to temporary user error.

noticeable differences were observed after dipping gold-plated electrodes into a 1M nitric acid bath. Control samples of copper and nickel, however, were quickly etched down to the bare methacrylate surface. The acid dip tests and subsequent microscope inspections (Fig. 3a) found no micro or nano cracks that may affect the electrode's surface or electrical properties.

**Surface roughness characterization.** Light microscopy photographs and stylus profilometry measurements were used to assess surface roughness between each step of the plating process on a single flat sample. Figure 3b plots the normalized surface topography of the sample during each plating step. The reported Rp values are the standard deviation of the plotted lines. Though surface roughness decreases slightly with each subsequent plating step, the final gold surface is still much rougher than a simple, planar surface. This increases electrode surface area, promotes better film adhesion, and reduces ESI[50,81,84,85].

**Sheet resistance.** Sheet resistance was characterized by a 4-point probe immediately after plating. 40 sheet resistance measurements were taken of each copper-, nickel-, and gold-plated samples. As prepared, copper-plated samples, nickel-plated samples, and gold-plated samples exhibited an average sheet resistance of $177.9 \pm 109$, $95.5 \pm 13$, and $30.3 \pm 3.7 \, m\Omega \, \square^{-1}$, respectively (Fig. 3c). With each subsequent metal layer, the sheet resistance stabilized, and the surfaces became more conductive.

**Bioimpedance of In-ear electrodes across multiple users.** Impedance spectroscopy was used to assess in-ear electrode-skin impedance. Four subjects took impedance measurements (20 total measurements) between the in-ear electrodes and the out-ear cymba electrode. To account for future, real-life conditions with cerumen and oil, no skin preparation was performed before each trial, and measurements were repeated until all four electrodes in the ear canal were measured. Since the ESI measurements include two dry electrodes, the plotted values were divided by two to demonstrate the average ESI of a single dry electrode. All measurements were performed with an LCR meter (E4980 A, Keysight) powered by a wall outlet and arranged as a two-point probe where a single electrode is considered a single probe. The LCR meter was configured with a current limit of 0.5 mA to prevent sensation or injury. While the LCR meter is designed to achieve high accuracy (within 3%) even in the presence of powerline interference, electrode cables were shielded by ground wires to further minimize interference. All impedance results were fitted to an equivalent circuit model (spectra shown in Fig. 3d, circuit model shown in Fig. 3e) to better understand motion artifact settling times associated with the phase elements of the electrode skin interface and provide reference for future analog front-end designs. At 50 Hz, the interface has an average impedance of 120 kΩ and phase of −33°.

### Lightweight ExG recording system
ExG was recorded using an existing compact, wireless recording platform affixed to a headband (Fig. 4a). The platform, known as

WANDmini, is a wireless neural recording frontend built for and already deployed in previous in-ear EEG studies[46]. It is adapted from a system originally designed for electrocorticography and comprises a custom neural recording circuit[68,86], (NMIC[86], Cortera Neurotechnologies, Inc.), a microcontroller, and a Bluetooth radio for wireless transmission. The NMIC digitizes up to 64, fully differential channels of electrophysiological activity with a sampling rate of 1 kSps. WANDmini arranges the NMIC's channels in a monopolar montage with a single reference electrode. This arrangement is it suitable for EEG, EOG, and EMG recording and provides enough sampling and channel count headroom to remove any recording electronics related bottlenecks. An onboard microcontroller and radio packetizes and streams digitized neural data to a base station connected to a host machine over Bluetooth Low Energy (BLE) (Fig. 4a). System power is dominated by the microcontroller and Bluetooth transmission (98.3%) thus making unused channels immaterial from a power perspective. With the NMIC and WANDmini power consumptions, 700 µW and 46 mW, respectively, a 3.7 V 550 mA battery can provide ~44 h of runtime. In summary, the NMIC's significantly lower power than common commercial neural frontends (e.g., ADS1298/1299), high channel count, and sufficiently low noise floor makes it ideal for use in modular in-ear EEG prototypes. NMIC and WANDmini specifications are listed in Table 1 and further detailed in Supplement section II.h. The host machine uses a custom graphical user interface (GUI) that plots and saves all incoming data and cues for the trail overseer. This custom GUI is unique to this work and provides the test subject with a reaction time game, auditory cues, and visual alerts during experiments. More information about the GUI is available in section 2h of the supplement.

## EEG characterization and user-generic drowsiness detection

**Drowsiness Study.** To characterize the full system performance, 35 h of Ear ExG data was recorded during a nine subject drowsiness study. Subjects wore two earpieces with the electrodes organized in a contralateral monopolar montage. Previous works have demonstrated that electrodes on a single earpiece are sufficiently distant from each other to measure ExG[37,41], but greater signal amplitude can be recorded with electrodes placed across both ears[39,45]. To induce drowsiness, subjects played a repetitive reaction time game. Every 60 s, a user was prompted to press a random number between 0 and 9 and their reaction time was recorded (Fig. 4a). Every 5 min, the user was prompted to enter a Likert item according to the Karolinska Sleepiness Scale (KSS). This scale is frequently used to evaluate subjective sleepiness and ranges from 0 = "extremely alert", to 10 = "extremely sleepy, fighting to stay awake"[87]. Queue intervals (60 s and 5 min) were selected based on initial experimentation and previous works that demonstrated a balance between minimizing disturbances and frequent datapoints[45,88]. All recorded ExG, cue timing, reaction times, and Likert items are saved by a custom GUI for post-processing and machine learning model training (Fig. 4b). Immediately after each trial, reaction time and Likert items were thresholded per subject to automatically generate alert/drowsy labels for each trial since behavior and response time metrics are heavily correlated with drowsiness[6,87,88]. By taking both an objective and a subjective drowsiness measurement, high-confidence data labels could be generated in face of user-error and user-bias (memory of previous KSS scores affecting subsequent scores). Both objective and subjective measures must agree to classify

## Table 1 | Relevant system, WANDmini, and NMIC specifications

| | |
|---|---|
| Maximum Recording Channels | 64 |
| Recording Channels Used | 11 |
| Reference Location | Right Cymba |
| Ground Location | Right Mastoid |
| Input Range | 100 mVpp |
| ADC Resolution | 15 bits |
| ADC Sample Rate | 1 kSps |
| Noise Floor | 70 nV/√Hz |
| Wireless Data Rate | 2 Mbps |
| NMIC Power | 700 µW |
| WANDmini Power | 46 mW |
| Battery Life | 44 h |

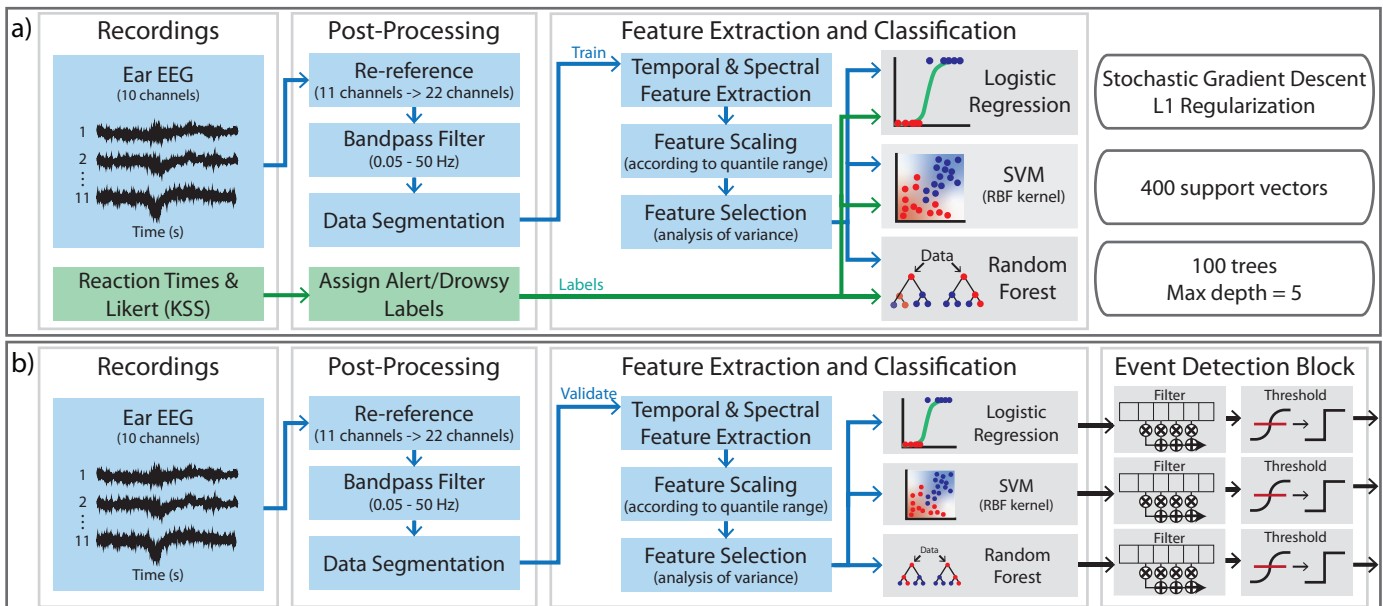

**Fig. 5 | Drowsiness Detector Training and Validation Diagram. a** Ear ExG experimental recordings are re-referenced, filtered, cleaned of motion-contaminated epochs, and then undergo feature extraction and model training. **b** Cross-validation is performed similarly, featurized ear ExG epochs are fed to all three classification models. Model outputs are then fed to an event detector that performs a moving average and then thresholds the resulting classifications to estimate alert and drowsy states.

an event as drowsy. Furthermore, as noted in previous works, reaction times and likert scores are variable on a subject-to-subject basis. As a result, each trial was thresholded on a per subject basis. Each trial contained at least one drowsy event, and 65 drowsiness events were recorded across 34 trials.

**Drowsiness classification pipeline.** The training pipeline for ExG data consisted of post-processing, feature extraction, and model training steps (Fig. 5a). ExG recordings were referenced to maximize spatial covering, band pass filtered, and segmented into 50 s or 10 s windows. If a window of data exhibited an artifact greater than 10 mV (from motion) it would be discarded. This was happened very infrequently as most artifacts were less than 1 mV above the baseline rms voltage. Temporal and spectral features relevant for ExG-based drowsiness detection were implemented to target ocular artifacts and activity in standard EEG frequency bands relevant to drowsiness detection: delta ($\delta$, 0.05–4 Hz), theta ($\theta$, 4–8 Hz), alpha ($\alpha$, 8–13 Hz), beta ($\beta$, 13–30 Hz), and gamma ($\gamma$, 30–50 Hz). Binary (alert/drowsy) classification was performed with low-complexity logistic regression, support vector machine (SVM), and random forest classifier models.

Three cross-validation techniques were used to estimate model performance across varying usage scenarios: user-specific, leave-one-trial-out, and leave-one-user-out. User-specific cross-validation trained models on $n-1$ trials for the subject, tested on their remaining trial, and averaged the results after n independent iterations to determine drowsiness detection accuracy for a single subject. Leave-one-trial-out cross-validation trained models on 33 of the recorded trails, tested on the remaining trial, and averaged results after all 34 independent iterations to determine the study's overall drowsiness detection accuracy. Leave-one-user-out cross-validation trained on recordings from eight subjects, tested on the remaining subject's recordings, and averaged results after all nine independent iterations. This evaluated detection accuracy when using population training and deploying on a never-before-seen subject. Due to the inherent imbalance between drowsy and alert classes, each classification model employed a balancing scheme where over-represented classes are given a smaller class weight than under-represented classes. In the case of drowsy vs. alert, alert epochs are given a class weight inversely proportional to the number of epochs. This allows classes to be treated more fairly across all training/cross-validation regimes (since they will all have different

class balances). During validation, class probabilities returned from the classifier models were filtered with a 3-tap Hamming window FIR filter and thresholded to achieve final binary outputs (Fig. 5b).

**Drowsiness classification results**
**Alpha modulation ratio.** Alpha waves (8–12 Hz) are a spontaneous neural signal that can reflect a person's state of relaxation, which makes them an important spectral feature in ExG-based drowsiness classification[15]. A sample recording from a single user demonstrating alpha wave modulation is presented in Fig. 6a. This modulation is clear in the time–frequency spectrogram (Fig. 6a). To assess the modulation ratio more quantitatively, Fig. 6a also plots the average power across the entire alpha band while the subject opens and closes their eyes every 30 s. The presented sample data's modulation ratio was 2.001.

**Classifier comparison across validation schemes.** The overall average of the user-specific classification results ranged from 77.9% to 92.2% across all models and feature window sizes. In the user-generic leave-one-trial-out case, average classification accuracy was higher and ranged from 91.4% to 93.2% when cross-validating across the 34 trials. This is most likely due to the increased amount of data available for training. Lastly, the leave-one-user-out validation scheme achieved average classification accuracies from 88.1% − 93.3% across all users, window sizes, and models. Figure 6b–g showcases average model accuracy and standard deviation where appropriate.

**10 s vs 50 s windows.** Two feature windowing schemes were investigated, 10 s (Figs. 6b–d) and 50 s (Fig. 6e–g) windows. All training steps, including feature selection, are performed independently. The 10 s feature windows result in significant performance loss in the user-specific validation scheme. For example, the average user-specific logistic regression-based classifier performance increased from 77.9% to 90.8% when increasing feature window sizes to 50 s. Minimal accuracy loss, however, was observed when using leave-one-trial-out and leave-one-user-out validation schemes with features from 10 s windows. This minimal accuracy loss is most likely due to the increased amount of training data available (~30 trials) to the models relative to the user-specific cases where individual models only train on a 1−4 of trials.

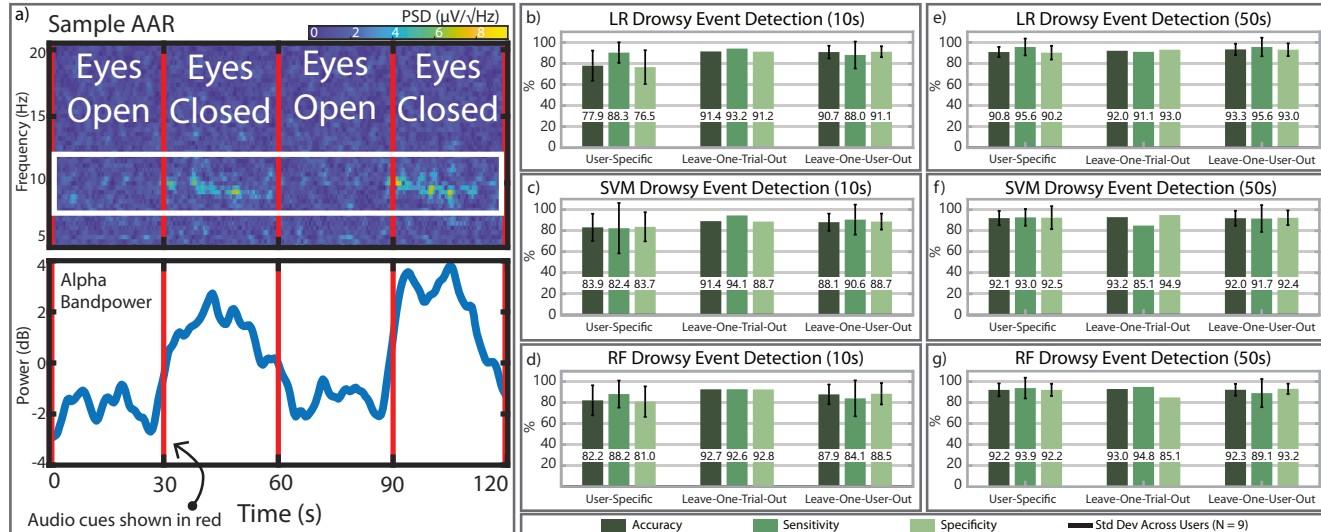

**Fig. 6 | EEG measurement and classifier performance. a** Spectrogram demonstrating alpha modulation when the subject closes their eyes. Alpha bandpower (8–12 Hz put through a 2 s rolling average filter for clarity) is modulated by 4× in amplitude when eyes are closed. **b** Logistic regression event detection with 10 s

feature windows. **c** Support vector machine event detection with 10 s feature windows. **d** Random Forest event detection with 10 s feature windows. (**e, f, g**) Drowsiness event detection using 50 s feature windows. Standard Deviation (Std Dev) shown across results from all nine users.

**Classifier architecture comparison.** Three low-complexity machine learning models were used to promote the scalability and usability of the drowsiness detection platform. All models were implemented in Python 3.8 using scikit-learn packages. Logistic regression models were implemented with a stochastic average gradient descent solver. L1 regularization was used to add a penalty equal to the absolute value of the magnitude of the feature coefficients. Support vector machines were implemented with a radial basis function (RBF) kernel to account for data that may not be lineally separable. The trained models utilized a maximum of 400 support vectors and a regularization parameter, $C = 1$. Random forest models were implemented with 100 trees and a maximum depth of five to prevent overfitting. These implementations resulted in memory footprints that were estimated using python's pympler package. The logistic regression, SVM, and RF models required 2.8 kB, 144.2 kB, and 63.8 kB respectively. These memory requirements are well within the capacity of modern microcontroller's embedded memories (e.g., 32-bit ARM Cortex-M).

Since all three models achieve high accuracy, it is clear that drowsiness is classifiable with in-ear eeg recording. No model shows markedly greater performance or another. The logistic regression model is more computationally efficient, requires significantly less memory, and can be more easily trained/deployed with smaller data-sets. It is important to verify that logistic regression continues to perform as well across larger demographics, a topic for future studies.

## Discussion

We have reported the design and fabrication of in-ear dry electrodes along with the assembly and evaluation of a wireless, wearable, in-ear ExG platform for offline drowsiness detection on never-before-seen users. All aspects of this platform can be adapted to different use-cases. The 3D printed and electroless Au-plated electrodes can be rapidly augmented for any anatomically optimized wearable and used/re-used for long periods of time, WANDmini can support multi-day electrophysiological monitoring, and the presented offline classifiers demonstrate the potential for future dry-electrode based brain-state classification. In contrast to other state-of-the-art in-ear recording platforms (Table 2), the electrodes, wireless electronics, and light-weight algorithms presented lay the groundwork for future large-scale deployment of user-generic, wireless ear ExG brain-computer interfaces that use multiple machine learning algorithms.

Our results are promising for the development of the next generation of standalone wearables that can monitor brain and muscle activity in work environments and in everyday, public scenarios. To realize these standalone, wireless systems, future work requires integrating these classifiers on-chip for real-time brain-state classification and miniaturizing all the hardware into a pair of earbuds. Furthermore, the hardware would need to support online classification to allow for full-day, itinerant use. Lastly, it would be important to take this miniaturized hardware and implement a user-study with a wider demographic. By monitoring in-ear EEG across individuals aged 18–65+, further age specific models can be investigated. If a monolithic model is unable to classify drowsiness stereotypes across such a large age range, it would be interesting to provide models with context such as age, gender, known sleep disorders, and previous night's sleep quality. Furthermore, the feature selection performed in this work suggests that simpler calculations such as bandpower ratios are sufficient for drowsiness classification. If this remains the case across larger demographics, then feature extractors can ignore computationally expensive features such as standard deviation, different entropy measures, etc. to reduce power in embedded classification scenarios. With aforementioned integration, a pair of ear ExG buds would significantly enable long term, daily recording ExG without interrupting a user's day or stigma. These measurements would enable an entirely new era of research for tracking long-term cognitive changes from disorders such as depression, Alzheimer's, narcolepsy, or stress.

## Methods
### Study approval and ethical consent

The user study, subject recruitment, and all data analysis was approved by UC Berkeley's Institutional Review Board (CPHS protocol ID: 2018-09-11395). Informed consent was received by all participants in the study for their results to be included in presented figures/data.

**Table 2 | Comparison of this work with other in-ear drowsiness monitoring works**

|  |  | Hwang '16 | Nakamura '17 | Hong '18 | Barua '19 | Gangadharan '22 | This Work |
|---|---|---|---|---|---|---|---|
| Setup | # Users | 13 | 4 | 16 | 30 | 18 | 9 |
|  | # Recordings | 13 | 4 | 16 | 312 | 18 | 34 |
|  | Recording length (min) | 60–90 | 45 | 55–75 | 30 | 40 | 40–50 |
| Electrodes | Format | In-ear | In-ear | In-ear | Scalp | Muse Headband | In-ear |
|  | Single/both ears | single | Single | Single | - | - | Both |
|  | Sense/Ref Electrodes | Wet | Wet | Wet | Wet | Dry | Dry |
|  | Ground Electrode | Wet | Wet | Wet | Wet | Dry | Wet |
|  | # Channels | 1 | 2 | 1 | 30 | 4 | 11 |
|  | Generic | Yes | Yes | Yes | Yes | Yes | Yes |
|  | Assembly material | Metallic | Foam | Silicone | Metallic | Plastic | 3D printed polymer |
|  | Electrode | -- | Ag/AgCl wire | Ag + Cu | -- | Au | Au |
| System | Wired/wireless | Wired | Wired | Wired | Wired | Wireless | Wireless |
|  | Data rate | - | - | - | - | 1 Mbps[+] | 1.96 Mbps |
|  | Power | - | - | - | - | - | 46 mW |
|  | Battery life | - | - | - | - | 5 h | 44 h |
| Algorithm | Model | SVM | SVM | SVM | SVM | SVM | LR |
|  | Window size | 5s | 30s | 60s | 60s | 4s | 50s |
|  | Sensitivity | - | - | - | 94%[+] | 78.95% | 95.60% |
|  | Specificity | - | - | - | 92%[+] | 77.64% | 93.00% |
|  | Accuracy | 88.30% | 82.90% | 93.50%[a] | 93% | 78.30% | 93.30% |

[a]99% when evaluating on 230s epoch of EEG, ECG, and PPG features

## Electrode fabrication

Both the electrodes and earpiece were printed with a stereolithography (SLA) 3D printer (Formlabs Form 3 printer) with a standard, clear methacrylate photopolymer (Fig. 2c). An SLA printer was used due to its increased precision over standard filament deposition modeling (FDM) based printers. In SLA printers, thin layers of photosensitive polymer are cured by a laser. The resulting printed surfaces must be washed and cured in UV to achieve the final 3D part.

The original 3D printed surface is highly anisotropic due to the structure's uniformly printed layers. To create a more heterogenous surface, electrode structures were sandblasted with 100 grit white fused aluminum oxide blasting media (Industrial Supply, Twin Falls, ID) to remove the regular surface pattern leftover from the printing process while also increasing the effective surface area. The sandblasted samples were then sonicated in a bath of Alconox cleaning solution for ~10 min and rinsed with DI water. Lastly, the electrode structures were treated in a bath of 1% benzalkonium chloride (Sigma Aldrich 12060-100G) surfactant solution for 10 min. These surface treatment steps ensure a clean plating surface with high surface energy and lead to improved catalyst/metal layer adhesion.

The samples are then submersed in catalyst and plating baths. First, the electrodes are submerged in a beaker of palladium-tin catalyst for 10 min followed by a copper plating solution for a minimum of six hours. This initial plating step results in a thick copper layer that will oxidize if left out in ambient atmosphere. As a result, samples would then be quickly rinsed, dried, and placed in a nickel-plating bath for ~10 min (Sigma Aldrich 901630). Afterwards, the electrodes are placed in an electroless gold plating solution for approximately 15 min. In between plating steps, the samples were rinsed with DI water and dried thoroughly.

## WANDmini: ExG recording hardware

The WANDmini board contains a neural recording frontend (NMIC), a SoC FPGA with a 166 MHz Advanced RISC Machine (ARM) Cortex-M3 processor (SmartFusion2 M2S060T from Microsemi), and low-energy radio (nRF51822 from Nordic Semiconductor). The SoC FPGA forms a custom-designed 2Mb/s digital signal and clock interface with a single NMIC, aggregates all data and commands into packets, then streams all the packets to the 2Mb/s 2.4 GHz low-energy radio.

WANDmini also contains a 20 MHz crystal oscillator as a clock source, on-board buck converters (TPS6226x from Texas Instruments), a battery charger circuit (LTC4065 from Linear Technology), and a 6-axis accelerometer and gyroscope (MPU-6050 from InvenSense). While WANDmini can record up to 64 channels of electrophysiological data and motion information from the accelerometer, the drowsiness detection application only uses 11 channels for ExG monitoring. Future applications may integrate real-time motion artifact cancellation and classification directly into the WANDmini's SoC FPGA.

## Subject selection and earpiece application

Nine subjects (7 male, 2 female, ages 18–27) volunteered for this study. Subjects were requested not to exercise or drink caffeine before any trial. Prior to the first experiment, subjects tried out small, medium, and large earpieces and selected the pair they felt were most comfortable and secure in ear. During this onboarding session, subjects also familiarized themselves with the GUI.

At the start of the drowsiness trials, subjects were given their preferred ear EEG earbuds to wear, as well as an electronics headband with a fully charged Li-Po battery and the WANDmini recording hardware. To maintain a realistic daily use scenario, the subjects did not clean or prepare their skin and no hydrogel or saline was applied to the earpiece dry electrodes. The trial hosts also did not help subjects don/doff the headband or earpieces unless explicitly requested. After the experiments, the earpieces were cleaned with 70% isopropyl alcohol since they would be later used by other subjects.

## Electrophysiological recording setup

Each earpiece has six electrodes, four inside the ear canal and two outside the ear canal. The default recording arrangement employs two contralaterally worn earpieces to maximize spatial coverage and recorded signal power[27,39]. These two earpieces provide up to 11 ExG channels with a common reference. Either of the concha cymba electrodes can be used as a reference (the un-used one can be used as an additional sense electrode). After initial experimentation, it was determined that the right concha cymba electrode was sufficient as a reference electrode across all subjects. As a result, each ExG channel is referenced against the right concha cymba electrode in a monopolar montage (electrode Y in Fig. 2a). A single wet Ag/AgCl electrode was applied to the subject's right mastoid and connected to battery ground for interference reduction.

## Drowsiness trial overview

Subjects participated in multiple drowsiness trials to enable both user-specific and user-generic training. Subjects were not familiar with the ear EEG work when selected. No more than five trials were recorded per subject to maintain a diverse data pool. Prior to the trials, subjects were informed of the study purpose and requested to have a 'normal night's rest' (subjectively) and not drink caffeine prior to the trial. Trials took place in a quiet, indoor office space between 8 a.m. and 5 p.m. when the lights were on. After donning the ear eeg system, the subject was left alone in the trial space until the end of the recording session. During the trial, the subject would sit at a desk in front of a laptop with a custom GUI. Subjects were instructed to only perform the reaction game task and not look at personal devices for the extent of the trial. Subjects were allowed to move their heads, readjust in their seat, and move their arms, but were asked to stay seated during the entire session (to minimize motion artifacts). Each trial was 40–50 min in length and was self-ended by the subject to prevent the interruption of a drowsy event. At the end of the trial, the subjects removed the headband and earpieces themselves. They were instructed to wait at least 24 h before participating in subsequent drowsiness trials to maximize variation between trials.

## Label generation

Recording both objective and subjective drowsiness measures made the label generation process robust to user-error momentary distractions (when an alert user looks away from the laptop). Ear ExG samples were labeled as "drowsy" if the user reported a drowsiness Likert item >5 and if their reaction time was more than double the average from the first 5 min of recording. The labels were then passed through a 3-sample rolling average filter and thresholded to achieve a binary label.

## Re-referencing and filtering

ExG re-referencing was used to maximize spatial covering across contralateral earpieces. Each in-ear electrode was re-referenced to the left concha cymba electrode and processed with the 11 EEG channels recorded with the right concha cymba electrode. To remove power-line interference (60 Hz in North America) while maintaining as much EEG activity as possible, both the recorded and re-referenced EEG channels were bandpass filtered from 0.05–50 Hz. Filters were implemented with a 5th order butterworth high pass filter (corner of 0.05 Hz) and a 5th order Butterworth low pass filter (corner of 50 Hz). Both filters were implemented in python but can also be implemented with infinite impulse response (IIR) filters with 16 bit registers for use in FPGA/embedded applications.

## Data segmentation

Filtered ExG was segmented to remove ExG artifacts related to decision-making and motor planning in response to GUI cues. Each epoch began 10 s after a reaction time cue and ended when the next reaction time cue was provided. When using the maximum window

**Table 3 | Per channel extracted features**

| |
|---|
| Maximum peak-to-peak voltage |
| Standard deviation of voltage |
| Maximum PSD (δ, θ, α, β, γ bands) |
| Peak frequency (δ, θ, α, β, γ bands) |
| PSD variance (δ, θ, α, β, γ bands) |
| Absolute power (δ, θ, α, β, γ, α/β, θ/β, (α + θ)/β, (α + θ)/(α + β)) |
| Relative power (δ, θ, α, β, γ, α/β, θ/β, (α + θ)/β, (α + θ)/(α + β)) |

**Table 4 | Top features selected for training and validation**

| |
|---|
| α relative power |
| β relative power |
| δ relative power |
| Previous epoch's α relative power |
| Previous epoch's β relative power |
| Previous epoch's δ relative power |
| θ/β absolute power |
| (α + θ)/β absolute power |
| (α + θ)/(α + β) absolute power |

size, features were calculated for these 50 s epochs. When using a reduced window size, each 50 s epoch and its corresponding label were divided into five 10 s windows. To focus classification on drowsiness onset, epochs were considered "sleep" if a subject's rection time exceeded 10 s. These epochs were excluded from the study.

**Feature extraction and selection**

Temporal and spectral features were extracted in Python 3.8 from the segmented ExG data. Low-complexity features were calculated for each window of ExG data and across all the recorded and re-referenced channels. Voltage standard deviation and maximum peak-to-peak voltage amplitude were calculated in the time-domain to target eye blink artifacts and motion. Welch's method (using a 1000-point Fourier transform, 500 sample overlap, and Hamming window) was used to calculate the power spectral density (PSD) and attain frequency characteristics that relate attention and relaxation. The following spectral features were calculated prior to training: maximum PSD, peak frequency, and PSD variance were calculated for δ, θ, α, β, γ EEG bands. Absolute and relative band powers were also calculated for the following bands and ratios: δ, θ, α, β, γ, α/β, θ/β, (α + θ)/β, and (α + θ)/(α + β). Relative bandpower is the specific band relative to the total PSD from 0.5–50 Hz. Furthermore, features of the previous epoch were included to account for changes in ExG activity, since temporal and spectral features relate to characteristics that changes during the onset of drowsiness such as attention and eye movement. A complete table of features used in offline training (prior to feature selection) is in Table 3.

All features were scaled by subtracting the median and scaling according to their interquartile range. To reduce input feature count, feature selection using an analysis of variance (scikit-learn Python 3.8) was performed to determine the top 20 features (total) that minimize redundancy and maximize class variation during training. Only these 20 features are included during model training and validation. This feature selection also implicitly selected best performing electrodes across users (most likely due to some electrodes fitting better than others). The same feature type was also selected for multiple channels (e.g., the top 20 features would include alpha band power from channels 1, 5, and 10). Contralateral channels (where sense and

reference electrodes are in different ears) were always weighted higher than ipsilateral channels. The most used features (in order of importance) are shown in Table 4.

Spectral features associated with eye movement, relaxation, and drowsiness were the most important for model training. Furthermore, the previous epoch's features were also generally important. This is corroborated by results from other works on scalp data in refs. 14,15,79. All feature extraction was performed in Python using numpy. For implementation into an embedded/FPGA environment, these features can be calculating using a coarse fast-Fourier transform, look-up-tables, and the CORDIC algorithm.

**Statistics and reproducibility**

No statistical method was used to predetermine sample size. No data were excluded from the analyses. The experiments were not randomized. The investigators were not blinded to allocation during experiments and outcome assessment.

**Reporting summary**

Further information on research design is available in the Nature Portfolio Reporting Summary linked to this article.

## Data availability

The Experimental Ear EEG data collected in this study is available at https://github.com/MullerGroup/EarEEG_Drowsiness. Due to IRB restrictions, access may be restricted to any raw EEG data. If there are any issues accessing the repository, please contact ryanka-veh@berkeley.edu, cschwendeman@berkeley.edu or rikky@berkeley.edu. Example code and a deployable notebook can be found in the GitHub repository. Source data used in all figures are provided with this paper. Source data are provided with this paper.

## Code availability

The source code used for offline model validation and analysis of results is available at https://github.com/MullerGroup/EarEEG_Drowsiness.

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

## Acknowledgements

The authors thank the Ford University Research Program, Bakar Spark Award, Cortera Neurotechnologies (acq. Nia Therapeutics), and the Berkeley Wireless Research Center sponsors. The authors would also like to thank Prof. Jan Rabaey and team, Miguel Montalban, Andy Yau, Adelson Chua, Justin Doong, Aviral Pandey, and Natalie Tetreault for technical support.

## Author contributions

R.K. developed the earpieces and fabrication techniques. R.K., C.S., and L.P. fabricated all test sensors. C.S. developed and implemented the machine learning algorithms. R.K and C.S. performed the experiments and analysis. R.M. oversaw all aspects of the research project. A.C.A. and R.M. oversaw the fabrication process development. R.K., C.S., A.C.A., and R.M. wrote and edited the manuscript.

## Competing interests

The authors declare no competing interests.
