## [Peer Review File · Nature Communications]

REVIEWER COMMENTS

Reviewer #1 (Remarks to the Author):

The authors reported an end-to-end design of an ambulatory drowsiness monitoring platform, incorporating both the earpiece design and associated drowsiness detection algorithms. A nine-subject study demonstrates the accuracy and reliability of the monitoring system. The reviewer suggests that this manuscript be accepted after the following concerns are addressed.

Major:

1. Recent advances in in-ear electrophysiological sensors, such as the work by Xu et al. in “In-ear integrated sensor array for the continuous monitoring of brain activity and lactate in sweat” (<https://doi.org/10.1038/s41551-023-01095-1>), demonstrate the growing popularity of this technology. The reviewer suggests the authors elaborate on the new findings of this study over theirs, especially regarding continuous EEG monitoring, to emphasize the distinctive contributions of this work.
2. It's crucial to understand how the earpiece's performance aligns with commercial EEG sensors. The reviewer recommends that the authors provide a comparative analysis between the earpiece's electrode impedance and the signal-to-noise ratio, vis-a-vis a commercial EEG sensor.
3. The choice of specific machine learning algorithms warrants more explanation. Are there any specific advantages or limitations of these algorithms in the context of drowsiness detection?
4. On Page 7, the claim about increased electrode surface area, enhanced film adhesion, and reduced ESI requires further empirical support. It would be beneficial to provide more detailed characterizations of surface roughness impedance to corroborate this statement.
5. The generalizability of the findings from the nine-subject study requires more discussion. What are the possible challenges to applying the same earpiece design and algorithms to a large and diverse population beyond the nine subjects included in the study? Are there any demographic or physiological factors that may influence the performance of the drowsiness detection system?
6. The statement that no skin preparation was done before tests raises concerns about the influence of cerumen on the earpiece's functionality. As cerumen can significantly alter the contact impedance, the reviewer suggests the authors characterize the electrode performance with and without the presence of cerumen. Furthermore, how does the cerumen affect the accuracy of drowsiness classification?

Minor:

7. The figure arrangement in the manuscript needs optimization. For example, Fig.2a and 2b appear to contain redundant information, efforts to make the figure design concise are appreciated. Moreover, the fabrication steps depicted in Figures 2d(i-v) can be consolidated into one, perhaps Figure 2d(v),

supplemented with pertinent material labels. This simplification will enhance the manuscript's readability and overall presentation.

Reviewer #2 (Remarks to the Author):

That's an interesting work, covering all the aspects of the design, from the electrode design and fabrication, to the development of ML models. I appreciate the efforts of the authors towards such end-to-end development. Still, I have the following comments and concerns:

General comments:

- the mechanical and electrical design process of the earpieces and its description is solid, nice work
- The acquisition electronics still appears bulky and attached to a headband; there is still a need for miniaturization before reaching the claim for an earbud system. I suggest to adapt the claims of the paper to account for this limitation
- multiple times, the discussion goes around ambulatory and real-life conditions. Conditions are different and requirements are different in these two settings (e.g., ambulatory conditions do not have the same strong requirements for non-stigmatization as real-life conditions). Please consider improving the overall storyline and the targeted use scenario
- a bench-top PC is needed for operation. No analysis is provided regarding the deployability of the algorithms/models on edge devices (computations needed? memory requirements? power measurements of deployed models?)
- I suggest to improve the comparison to state of the art. The main claims of the paper cover both electrodes design, electronics/readout platform design, ML models. For each one of these contributions, a discussion about how they compare to SoA is recommended. The key novelty and novel contributions of the paper should also be better stated and highlighted (e.g. at the end of the introduction), it's not clear if the authors are claiming that they improved state of the art for each individual element of the chain, or if the novelty lies "only" in the merging of all these elements (without claiming them to be state-of-the-art on their own)

Abstract:

- the discussion spans from ambulatory to use in real-life conditions (e.g. cabin sensors for operators). What is the real goal of the paper? consumer/real-life applications or medical/ambulatory assessment? It should be better clarified

- I suggest to include a bit more details in the abstract. E.g. simply saying "customized wireless electronics", "low-complexity ML" does not give the idea, to potential readers, about what is the added value and contribution beyond SoA of this paper. The abstract reports some quantitative numbers, but does not give an indication of the overall comparison to SoA.

Introduction:

- minor: I suggest to add a reference related to the first quantitative statement (number of driver drowsiness events resulting in accidents)

- this section presents the limitations of other approaches for drowsiness detection, most notably the camera-based eye tracking. The issue of sunglasses preventing the tracking is discussed. However, glasses featuring cameras in the glasses frames do also exist, and they can be used for drowsiness detection. Please incorporate in the bibliography and discussion also these solutions (e.g., but not only: <https://www.mdpi.com/1424-8220/23/7/3475>)

- EEG/EMG/ECG are introduced for the comparison. However, electrodermal activity is also a biomarker for drowsiness and it is not mentioned in the paper. I suggest the authors to revise the bibliography to account also for EDA

- second paragraph: EEG is a general term, not just for over-the-scalp measurements, since it can be applied on the scalp (surface EEG) or below (intracranial EEG). I suggest to amend this first sentence.

- second paragraph: the application of EEG for epilepsy is not only based on large 10-20 setups. Wearables are also becoming more popular

- the literature for dry-based ear-EEG can be improved. Some relevant references and products that are missing: for custom-made earpieces (<https://mjn.cat/en/>), for generic earpieces (<https://ieeexplore.ieee.org/abstract/document/9871874>, <https://pubmed.ncbi.nlm.nih.gov/24109744/>, <https://www.naox.tech/>)

- the literature for the electronics/readout platforms also needs to be improved. Examples include but should not be limited to: <https://ieeexplore.ieee.org/abstract/document/10189286>, <https://www.naox.tech/>

- "As these interfaces become more complicated and employ onboard machine learning algorithms [35]–[38] ..." not all of the papers cited here are really for on-board ML. Please focus the references on ML approaches that demonstrated deployment on edge-platforms (e.g. <https://ieeexplore.ieee.org/abstract/document/9995107>, https://openaccess.thecvf.com/content_cvpr_2017_workshops/w4/html/Reddy_Real-Time_Driver_Drowsiness_CVPR_2017_paper.html, <https://arxiv.org/abs/2111.03177>, ...)

- 5th paragraph: the need for classical ML is not clear. Why do the authors argue that these approaches are needed, as compared to DL solutions? Is it for transparency? for interpretability? why do you need them in the first place? I suggest to justify it better

- The authors mention "an ambulatory EEG system must continue to provide feedback when wireless connectivity is poor and there is unreliable access to large processing power for machine learning" however this study relies on a bench-top PC, with no embedded implementation. That's a major limitation that is not matched to what appear to be the main claims of the paper (which should be better stated)

Sect 2a

- it seems that the main novelty with respect to [45],[46] is the addition of the soft earpiece body to make electrodes more comfortable? please clarify the main novelty

- Good approach of having a partially flexible earpiece that adjusts to different ear shapes. Well done. Plating process with copper, nickel, gold seems good. The claim that the addition of the nickel layer (compared to [45]) significantly extends the lifetime of the electrode should be shown experimentally (or with appropriate references).

- it is mentioned that the new design allows to reduce ESI. Please be quantitative, reporting comparative numbers for both the older version (also taking them from the previously published paper) and new version, to demonstrate the quantitative impact of the proposed changes on the ESI reduction

- I suggest to include a summary of the key novelty and key contributions of this work as regards the electrode fabrication, with a comparison to state of the art (including but not limited to references 45,46,47)

- why are the authors introducing a lumped element model of the electrode-skin impedance? to me it is not entirely clear how this equivalent model is used and if it aids the design optimization

Sect 2b

- I did not find it entirely clear if the authors took a pre-existing readout platform or did a custom design based on pre-existing systems (the abstract gives the impression of something custom, where they say "customized wireless electronics")

- does the platform operate with a fully-differential or monopolar montage?

- I did not find entirely clear what are the key novelty statements of this part of the work, i.e. the readout platform design? How does this platform compare to SoA platforms for EEG acquisition/processing? (note: the comparison should not be made only towards platforms for drawsiness, since the acquisition platform is for acquiring generic EEG signals and there is no customization specific for ear-EEG signals)

- please provide more details about the data collection on the host machine. Was a custom GUI designed? (the answer to this question also relates to the question above about how much novel is this data acquisition platform)

Figures

- Fig 2: No justification regarding the placement of the different channels or the reference.
- Fig 4: "Drowsy events, shaded in green, are determined when a subject's reaction time and Likert response cross a drowsiness threshold that is determined per subject." Why is the labeling corrected on a per subject basis?
- Fig 6: Alpha band-power seems to rise before the subject closes the eyes. I suggest to add a comment about it in the figure caption. Why does it happen?

Methods

- since the authors target embedded applications, the description of the whole pipeline (including feature extraction and classification) needs to be made deeper, including the assessment of computational complexity and memory requirements
- the authors mention "...determine the 20 features that minimize redundancy and maximize class variation during training". It would be nice to report what are these selected features

Reviewer #3 (Remarks to the Author):

This is a very interesting paper reporting the development and evaluation of an in-ear EEG drowsiness detector system. In addition to important technological developments, which seem to be largely based on previous publications (e.g. in-ear dry electrodes design and fabrication, WANDmini), the paper reports classification results for 9 participants performing a monotonous attention task for a total of 35 hours. The key result is that computationally efficient classifiers (e.g. logistic regression) performed very well in detecting drowsiness states (performance > 90% accuracy). While overall very impressive, the paper lacks important details. Specifically, the evaluation study not very convincing. The following concerns should be addressed:

- the introduction part ignores around-the-ear EEG technology, a development known to fill the gap between cap-based and in-ear based EEG acquisition. It is for instance not correct to state that "existing wet electrode arrays tend to be large and delicate for everyday use". The cEEGrid flex-printed electrodes array is a wet electrodes system which has been successfully evaluated repeatedly (e.g. Debener et al., 2015, Scientific Reports, and subsequent publications from several independent research groups). Likewise, several statements throughout the introduction are not backed up by evidence and appear to express the opinion of the authors instead of summarizing recent progress in the field (e.g. "periodic maintenance", software packages requiring "extensive training"). What I miss is a more balanced introduction of in-ear dry EEG electrodes. Many researchers working in the field are aware that dry electrodes perform poorly in mobile conditions (i.e. when the participant moves). This drawback holds also for in-ear positions, because the outer ear canal deforms when individuals talk, chew, eat or drink and in-ear EEG systems provide very little spatial information in contrast to around-the-ear and cap-EEG systems (see Kappel et al., 2019, Front Neuroscience; and Meiser et al., 2020, Brain Topography), making EEG analysis much harder (note that space is a crucial dimension for EEG artifact attenuation). A disadvantage of currently available dry electrodes is that people may not tolerate dry in-ear electrodes over long periods of time, since even modest pressure can cause pain after a little while (same problem as for dry cap EEG systems). While I agree that the solution proposed in this paper is really interesting and clearly has its merits, it should be made clear that all currently available ear-EEG systems are far from perfect and far from ready for chronic (i.e. 9 to 5) use in real-life conditions.

- please state more clearly which parts of the technology are really new and what was already reported in previous studies. Since earpiece design and electrodes fabrication seem to be largely based on previous work, I suggest to remove corresponding details and report primarily essential and new information.

- The paper promises a complete monitoring platform but many details are missing. Please describe/illustrate for instance the electrodes-to-amplifier connection in detail (connector type, length of wires, position of the solder touching the skin). Regarding usability and signal quality, connectors and wiring are essential parts of a complete system. Likewise, please describe the host system used. Apparently, the amplifier was used for data acquisition and wireless transmission only, not for online signal processing, and therefore the offline monitor system requires some computing hardware (smartphone or PC for data acquisition and for offline computing).

- section IV b, there is no such thing as "artifact-free neuromodulation", the name is simply inappropriate for a research platform.

- section IV, if ear-EEG fabrication should remain an essential part of this manuscript I would like to see long-term impedance recordings, if possible along with actigraphy. A proposed monitoring system may not have to work during gross movement, but it should tolerate gross movements (e.g. conversations, short walks etc) to an extent that signal quality recovers after those events. Without such evidence, and

online system processing validation, the claim of a drowsiness monitoring system near ready for daily life application should not be made.

- section C I, the drowsiness study needs to be described in much more detail. How were the participants included into the study recruited, were they already familiar with in-ear EEG and the study purpose? Was ethical approval obtained for this work? What was the recording context, did participants sit at a table throughout the recording? Were they alone in the room, and was the recording environment a lab, an office or a home environment? What did they do in addition to the task, how much had they slept on the night before data acquisition, what time of day were the data acquired, were participants monitored by an experimenter all the time, what was their movement pattern during data acquisition., etc. Many context factors are not mentioned but could have influenced the results.

- regarding drowsiness detection, it is not described how reaction time and likert scale thresholds were derived. Were thresholds adapted to optimize system performance based on all data or determined by independent data not used for the final evaluation (as it should be)? It is difficult to evaluate system performance as long as the gold standard is not convincingly presented.

- signal validation. EEG alpha eyes open vs eyes closed spectra are shown, but it is not stated which EEG features contributed to classification. I miss a complete description of the calculation of the EEG features (FFT?, resolution?, overlap?, windowing?, bandpass filter characteristics, etc), the feature selection process and the feature selection validation process (Figure 4 mentions ANOVA feature selection) and finally descriptive statistics which features contributed (e.g. was alpha power really discriminative?). At present all "neural" data presented in the report are anecdotal evidence and their contribution to system performance is not clear. It is also not explained how EEG artifacts like eye blinks, movement artifacts etc. were attenuated. Please show representative raw data (maybe in supplemental material), not just processed data. Even better would be to provide the code for the offline analysis along with the paper.

- it is known that computationally efficient classifiers perform very well with EEG data (see Blankertz et al., 2011, Neuroimage). This should be made clear. It is however very surprising that a classifier performs better on never-before-seen users. How could this be? I do not remember coming across any EEG/BCI classification paper reporting such a finding. May it be that one, or a few users, contributed non-neural signals in an unbalanced way? Maybe some participants had different eye blink patterns for drowsy versus non-drowsy events?

- Along those lines, it would be more adequate to use EXG throughout the paper and avoid "neural signals" and EEG as long as the neural contribution to classification is not clearly identified and validated.

Reviewer #4 (Remarks to the Author):

This paper presents an in-ear EEG device and study using it for drowsiness detection. In general ear EEG is a very active and timely research area, and this paper is an interesting 'full stack' approach, covering everything from the electrodes to the instrumentation to the machine learning based analysis. However, the work is generally very brief and lacking in details in many areas which does not make it suitable for publication in its current form.

Points for considering:

- More motivation is required for the chosen machine learning models, which are in general far from current state of the art approaches. A motivation of transparent machine learning is given, and this is an important motivation, but there is no detail behind this. For example, to me, generally an SVM isn't considered a "transparent" method unless it's paired with something to explain the feature selection made and the weighting given to these. Just being less of a black box compared to deep learning approaches doesn't mean the approach is transparent.
- A full review isn't needed, but for fairness more commercial EEG systems than CGX, Muse and Open BCI should be listed. Well known references such as G. Niso, E. Romero, J. T. Moreau, A. Araujo, L. R. Krol, "Wireless EEG: A survey of systems and studies," *NeuroImage*, vol. 269, no. 119774, pp. 1-17, 2023 should be included to guide the reader to other systems.
- Similar to the above, a reasonable review of electrodes is included, but references to review papers to guide readers to more comprehensive work are lacking. Potential examples include: G. Di Flumeri, P. Aricò, G. Borghini, N. Sciaraffa, A. Di Florio, F. Babiloni, The dry revolution: evaluation of three different EEG dry electrode types in terms of signal spectral features, mental states classification and usability, *Sensors* 19 (2019) 1365. Y. Fu, J. Zhao, Y. Dong, X. Wang, Dry electrodes for human bioelectrical signal monitoring, *Sensors* 20 (2020) 3651. C. Im, J.-M. Seo, A review of electrodes for the electrical brain signal recording, *Biomedical Engineering Letters* 6 (2016) 104–112.
- Information on how the E4980 was isolated during bioimpedance measurements should be added. There is no section on bioimpedance in the methods.
- References supporting the validity of doing repeated KSS scores every 5 minutes should be added. A priori I would assume that the scores are highly correlated, and potentially bias one another as they are close enough together in time to easily remember the previous score. It is not clear how this and similar effects were accounted for.
- EEG frequency bands were determined using Welch's method. There is no information on the settings used for this; the window size, the overlap, the number of FFT points, the window function used, and so on.
- It is only vaguely stated which features are used in the machine learning analysis. The text states "Time-domain features including..." and "Spectral features including..." and so doesn't actually give a list

of what was used, and whether there are more (or less as not all were applied to all channels) than those mentioned explicitly.

- The manuscript is incomplete in a number of places. There is a hanging sentence "Sample alpha attenuation response". A "Nikon XX microscope" was used. "Sensitivity represents the true positive rate or the number of correctly classified drowsy events (REF)" and a number of similar REF statements rather than an actual reference.
- It is stated that "classification accuracy was significantly higher and ranged from 91.4% to 93.2%" and "The 10 second feature windows result in significant performance loss" with no details on the statistical tests used.
- The contribution of the WANDmini device was not clear. The abstract and introduction implied that a key contribution was considering all parts of the system together, but with 64 channels this EEG unit appears poorly matched for this application. Is it a new contribution here, or just what was selected to use? If the latter, why not use a lower power, lower channel count device?
- The abstract states "low-complexity machine learning algorithms". From this, I was perhaps expecting the algorithms to be running online in the hardware, but this is minor. More importantly, there is no exploration of this in the article. No measures of run-time or complexity are included. Whilst probably correct, most AI accelerators are focusing on deep neural networks, and so I don't think it is necessarily true that the methods chosen will actually be lower power when running online compared to more modern approaches which can make use of dedicated hardware and software accelerators.
- No information on the data balancing is given. How equally sized are the drowsy and non-drowsy datasets?
- The article states "Both the recorded and re-referenced EEG channels were bandpass filtered from 0.05-50Hz since these frequencies are associated with drowsy and attentive EEG activity." What filter (order, type) was used? Moreover, 0.05 - 50 Hz basically covers the whole EEG range, and so the justification is not suitable.

Small, medium and large electrodes are mentioned in the supplementary results, but there are no details on them apart from for the medium. What is the change in electrode size? What is the expected change in signal quality (if any meaningful) as a result?

First, the authors would like to express their gratitude to the editor and reviewers for providing insightful and instructive feedback to the authors. Below is the list of questions and comments raised by the reviewers alongside our responses. For the convenience of the reviewers both the modifications made to the manuscript and our responses below are shown in green.

Reviewer 1:

1. Recent advances in in-ear electrophysiological sensors, such as the work by Xu et al. in “In-ear integrated sensor array for the continuous monitoring of brain activity and lactate in sweat” (<https://doi.org/10.1038/s41551-023-01095-1>), demonstrate the growing popularity of this technology. The reviewer suggests the authors elaborate on the new findings of this study over theirs, especially regarding continuous EEG monitoring, to emphasize the distinctive contributions of this work.

Thank you for highlighting this important area of work. Xu et al.’s paper was not published when we initially submitted our work for review and we’re incredibly excited by its results. We have updated the 3rd and 4th paragraphs of our introduction to reference (reference 44) this and other works employing multi-modal earpieces with flexible arrays.

Discreet, multi-channel EEG recordings from inside the ear canal have been demonstrated [26]–[28] with recent advancements focusing on earpiece design, electrode materials, and multi-sensor arrays. The ear canal is an ideal sensor location due to its inherent mechanical stability and wealth of potential recording modalities. In-ear sensors and electrodes are well situated to record temporal lobe activity, blood oxygen saturation, head movement, and masseter muscle activity making it ideal for multi-modal sensing if high spatial coverage is not required [29], [30]. While some applications may treat muscle activity or ear canal deformation as interference signals, these signals can be useful for other general ExG workloads. It is also important to note that in and around-the-ear EEG is inherently limited in gathering spatially encoded brain-activity relative to broader scalp arrays[27], [31]. With this in mind, many successful designs have leveraged hydrogel coated on flex-pcb arrays or user-customized earpieces to record ExG features such as EOG, low-frequency EEG (1 – 30 Hz), and evoked potentials (40 – 80 Hz) [26]–[28], [32], [33]...

...
...

Recent user-generic earpieces equipped with wet electrodes, dry electrodes[39]–[42], PPG, and/or chemical sensors have achieved high degrees of accuracy for brain-state and activity classification [39], [40], [43]–[46]. Additionally, Dry-electrode based in-ear ExG have recorded low frequency neural rhythms, evoked potentials, and EOG comparable to wet-electrode. While potentially more susceptible to noise due to higher electrode-skin impedance (ESI) interfaces [47], dry electrodes eliminate the use of hydrogel, simplify the earpiece application process, and can improve user comfort. To achieve a middle ground between comfort and low ESI, state-of-the-art dry electrodes employ a wide range of solutions ranging from exotic materials, conductive composites, capacitive interfaces, solid-gels, and high-surface area 3D electrodes (microneedles, fingers, and nanowires) [20], [40], [41], [48]–[56]. PEDOT:PSS and IrO₃ are commonly used in the small-scale production of rigid electrodes due to their superior conductivity

and faradaic interfaces [57]–[59]. Both materials promote charge transfer by leveraging doped surfaces and high effective surface areas. Conductive, flexible composites, such as silvered-glass silicone and carbon-infused silicone, are not as conductive as PEDOT:PSS and IrO₃ but offer significantly greater comfort. Conductive composites are made from polymers or elastomers that can be molded into arbitrary shapes for anatomically fit electrodes and use added conductive particles to achieve a desirable ESI. The more conductive particles that are added will ultimately limit polymer cross-linking and may lead to cracking over time [60]. The clinical and industry standard materials are silver/silver chloride (Ag/AgCl) and gold due to their cost, biocompatibility, and electrical properties. Ag/AgCl can be painted on 3D electrodes to form consistent, faradaic, low-impedance interface through hair and grime. Furthermore, Ag/AgCl is also popular for consumable electrodes since the conductive particles deplete over time [61]. Gold electrodes are more inert, can be repeatedly reused, and form a capacitive interface that is not reliant on added conductive ions. While potentially more susceptible to motion artifacts and interference, gold’s lifetime and chemical properties make it ideal for long-lasting ExG recording systems. Most commercial wearables and existing in-ear ExG systems use Ag/AgCl, Au, or conductive composite electrodes has made it the choice material for electrophysiological recording hardware and commercial wearables [24], [62]–[64]

Lastly, we updated the final paragraph of the introduction to focus on our unique specific system integration, electrode technology, and classification, demonstrating a long-lasting gold-plated earpiece capable of brain-state classification across multiple users.

This project is the first integration and demonstration of wireless, dry-electrode in-ear ExG sensors used for drowsiness classification. This body of work presents a complete system inclusive of novel sensor manufacturing, adapted electronic design, use of open-source machine learning classification, and a 9-subject study. A fabrication process for dry, gold-plated electrodes suitable for repeated, comfortable, low impedance earpieces is introduced and tested over the course of months of electrode use. This electrode technology provides a unique method for the rapid prototyping of reusable, Au electrodes that remain stable over 12 months of use. These electrodes can replace existing solutions that rely on shorter-lifespan Ag/AgCl electrodes or expensive materials such as platinum or IrO₃. The earpieces are then coupled with wireless, discreet electronics capable of taking uninterrupted, low-noise neural measurements for over 40 hours to form a wearable, in-ear ExG system. The resulting Ear ExG BCI is then demonstrated with a nine subject drowsiness monitoring study. Low-complexity temporal and spectral features are extracted from the recorded ExG data and used to train multiple, offline machine learning models for automated drowsiness detection. The best performing model utilizing a support vector machine achieved an average drowsy-event detection accuracy of 93.2 % when evaluating on users it has seen before and 93.3% when evaluating never-before-seen users. This system and its use of offline classifiers lays the groundwork for future, discreet, fully wireless, long term, longitudinal brain monitoring.

2. It's crucial to understand how the earpiece's performance aligns with commercial EEG sensors. The reviewer recommends that the authors provide a comparative analysis between the earpiece's electrode impedance and the signal-to-noise ratio, vis-a-vis a commercial EEG sensor.

This is an excellent point. Comparing the earpiece to commercial EEG sensors would be enlightening. While we cannot re-run the EEG experiments, we have gone back to retake electrode impedance measurements of both our dry electrodes and standard gold cup EEG sensors (with 10-20 hydrogel). As expected, the gold cup electrodes with hydrogel have lower impedance magnitude than our dry electrodes (even with a smaller electrode surface area), but phase performance is relatively similar (they are both gold electrodes after all). We have added an impedance comparison with a standard, commercial EEG gold cup electrode (figure 2s in the supplement) as well as a description in supplement section 1.b.

Figure 2s: Commercial gold cup electrode impedance (with hydrogel) In ear electrode impedance

Dry electrode impedance is slightly larger than that of commercial gold cup electrodes (Fig 2s). At 50 Hz, the in-ear electrodes achieve an impedance of 109 kΩ whereas commercial wet electrodes (without skin abrasion) have an impedance of 31.7 kΩ. While this difference of impedance may not seriously affect measured signal amplitude, dry electrode interfaces are more susceptible to motion artifacts.

3. The choice of specific machine learning algorithms warrants more explanation. Are there any specific advantages or limitations of these algorithms in the context of drowsiness detection?

Thank you for bringing up this question. There are trade-offs to any choice of machine learning algorithms and it is very important we discuss this. To address this, we have added significantly more discussion to the 6th paragraph of the introduction.

In addition to system-optimization, the choice of machine learning algorithm determines system functionality from the perspective of training, data, and processing requirements. Every-day ExG EEG systems would ideally work out of the box, improve over time, and continue to provide feedback when wireless connectivity is poor and there is unreliable access to large processing power (construction sites, planes, and trucks). Classical algorithms that can evaluate spontaneous EEG on-chip such as logistic regression, SVMs, and random forest have demonstrated impressive success in classifying neural signals with limited datasets [25], [74], [81], [82]. Neural network based algorithms have also achieved impressive results [83]–[86] and are good candidates for further research. Generally speaking, neural network based algorithms require more compute power that is power-hungry for common embedded microcontrollers. Algorithms such as SVMs, logistic regression, and random forest generally require less processing power than similarly performing neural net or perceptron-based architectures, making them ideal for low-power, edge-based deployments on existing microcontrollers. Additionally, while existing in-ear ExG BCIs have achieved high classification accuracies with user-specific training and validation [43], [84], [87], [35], [88], ideal in-ear ExG EEG wearables would leverage pre-trained algorithms so never-before-seen users can use these devices without time-consuming training. This user-generic classification has been explored in scalp-based drowsiness monitoring with great success but not yet with in-ear ExG [15].

4. On Page 7, the claim about increased electrode surface area, enhanced film adhesion, and reduced ESI requires further empirical support. It would be beneficial to provide more detailed characterizations of surface roughness impedance to corroborate this statement.

The initial surface treatment that increases electrode surface area, enhanced film adhesion, and ultimately reduced ESI is based off previous works that have demonstrated these claims. To further verify these claims, we added clarification to the novelty of this work as well as further explanation to each section describing electrode results.

In *Electrode Fabrication* (Section II.a.ii) now contains:

This plating process is expanded on [45], [90] with the addition of a nickel layer that limits grain-boundary diffusion of copper and significantly extends electrode lifetime [90]–[92]. Furthermore, the nickel-plating step removes the need for repeated electroless palladium plating and the overall number of fabrication steps. While other in-ear electrodes use expensive materials like IrO₃ or hydrogels [39], [40], this improved layer stack-up (Cu, Ni, Au) is reminiscent of printed-circuit-board fabrication and enables similar levels of scale for electrode prototyping. The final surface contains at least 0.5 μm of copper, 0.5 μm of nickel, and 0.25 μm of gold and is suitable for dry electrode recording.

Material acid dip tests & tape tests (Section II.a.iii.1) now includes references backing up claims on adhesion:

No visible gold, nickel, or copper was removed with the tape indicating strong adhesion to the methacrylate substrate [90], [93].

Surface roughness & characterization (Section II.a.iii.2) now ties back to profilometer measurements and includes references justifying claims:

Though surface roughness decreases slightly with each subsequent plating step, the final gold surface is still much rougher than a simple, planar surface. This increases electrode surface area, promotes better film adhesion, and reduces ESI [50], [90], [93], [94].

5. The generalizability of the findings from the nine-subject study requires more discussion. What are the possible challenges to applying the same earpiece design and algorithms to a large and diverse population beyond the nine subjects included in the study? Are there any demographic or physiological factors that may influence the performance of the drowsiness detection system?

The reviewer makes an important point that must be clarified in the paper. Firstly, subject selection criteria has been moved from the supplement into the main paper's body.

Nine subjects (7 male, 2 female, ages 18-27) volunteered for this study. Subjects were requested not to exercise or drink caffeine before any trial. Prior to the first experiment, subjects tried out small, medium, and large earpieces and selected the pair they felt were most comfortable and secure in ear. During this onboarding session, subjects also familiarized themselves with the GUI.

At the start of the drowsiness trials, subjects were given their preferred ear EEG earbuds to wear, as well as an electronics headband with a fully charged Li-Po battery and the WANDmini recording hardware. To maintain a realistic daily use scenario, the subjects did not clean or prepare their skin and no hydrogel or saline was applied to the earpiece dry electrodes. The trial hosts also did not help subjects don/doff the headband or earpieces unless explicitly requested. After the experiments, the earpieces were cleaned with 70% isopropyl alcohol since they would be later used by other subjects.

In addition, we did not notice any differences between male and female subject results. A current limitation of our existing subject selection is that everyone was from within a narrow age range and had no history of sleep disorders. It is likely that including older subjects and subjects with abnormal sleep patterns may affect performance without a larger dataset.

6. The statement that no skin preparation was done before tests raises concerns about the influence of cerumen on the earpiece's functionality. As cerumen can significantly alter the contact impedance, the reviewer suggests the authors characterize the electrode performance with and without the presence of cerumen. Furthermore, how does the cerumen affect the accuracy of drowsiness classification?

To assess the impact of cerumen and dust on the earpiece functionality, further impedance measurements were performed with "unclean" and "clean" ears and earpieces. Two subjects took measurements before cleaning their ears and the earpieces with isopropyl alcohol. After

cleaning they took another set of measurements. While the 'cleaned' electrodes did perform slightly better, there was no large difference in electrode impedance between the two experiments. We added these new results to supplement section 1.b.

Electrode impedance is unchanged in the presence of naturally occurring cerumen (ear wax and oil). To better understand how electrode behavior may change in the presence of cerumen in real human ears, electrode impedance was measured initially without any skin or earpiece cleaning and again with a simple ear and earpiece cleaning step (Fig 1s). Ears and electrodes were cleaned with isopropyl alcohol and kim wipes to remove any physical detritus. Final measurement results (N=8) indicated no meaningful difference between the 'uncleaned' and 'cleaned' scenarios. This is likely due to the large surface area, capacitive electrodes, which are not relying on a direct faradaic charge transfer.

We have also included figure 1s in the supplement to show this comparison of electrode impedance.

Figure 1s: In ear electrode impedance magnitude and phase before and after skin/electrode cleaning (N=8 for both measurements).

7. The figure arrangement in the manuscript needs optimization. For example, Fig.2a and 2b appear to contain redundant information, efforts to make the figure design concise are appreciated. Moreover, the fabrication steps depicted in Figures 2d(i-v) can be consolidated into one, perhaps Figure 2d(v), supplemented with pertinent material labels. This simplification will enhance the manuscript's readability and overall presentation.

Thank you for these comments! The authors agree and have removed figure 2b since (as you mentioned) it is similar to figure 2a. We also rearranged figure 2d to consolidate and to make it clearer that each electrode rendering connects to a fabrication step. We hope these changes improve the readability of the figure.

Figure 2 (with updates)

Reviewer 2:

1. General Comments: The acquisition electronics still appears bulky and attached to a headband; there is still a need for miniaturization before reaching the claim for an earbud system. I suggest to adapt the claims of the paper to account for this limitation.

Thank you for this comment, the reviewer brings up a good point that should be better clarified in the text. We have adjusted our claims and discussed the required improvements needed for an earbud system in the *summary* (section 3).

Our results are promising for the development of the next generation of standalone wearables that can monitor brain and muscle activity in work environments and in everyday, public scenarios. To realize these standalone, wireless systems, future work requires integrating these classifiers on-chip for real-time brain-state classification and miniaturizing all the hardware into a pair of earbuds. Furthermore, the hardware would need to support online classification to actually allow for full-day, itinerant use. With further integration, a pair of ear ExG buds would significantly enable long term, daily recording ExG without interrupting a user's day or stigma. These measurements would enable an entirely new era of research for tracking long-term cognitive changes from disorders such as depression, Alzheimer's, narcolepsy, or stress.

2. General Comments: Multiple times, the discussion goes around ambulatory and real-life conditions. Conditions are different and requirements are different in these two settings (e.g., ambulatory conditions do not have the same strong requirements for non-stigmatization as real-life conditions). Please consider improving the overall storyline and the targeted use scenario.

Thank you for raising this important point about the distinction between ambulatory and real-life conditions. We were using ambulatory in terms of ambulation and not in reference to emergency scenarios. To make this clearer, we removed the use of ambulatory from the manuscript and focused on specifying 'every day, real-life conditions'. The *summary* (section 3) also now clarifies 'itinerant use' instead of ambulatory.

Our results are promising for the development of the next generation of standalone wearables that can monitor brain and muscle activity in work environments and in everyday, public scenarios. To realize these standalone, wireless systems, future work requires integrating these classifiers on-chip for real-time brain-state classification and miniaturizing all the hardware into a pair of earbuds. Furthermore, the hardware would need to support online classification to actually allow for full-day, itinerant use. With further integration, a pair of ear ExG buds would significantly enable long term, daily recording ExG without interrupting a user's day or stigma. These measurements would enable an entirely new era of research for tracking long-term cognitive changes from disorders such as depression, Alzheimer's, narcolepsy, or stress.

3. General Comments: A bench-top PC is needed for operation. No analysis is provided regarding the deployability of the algorithms/models on edge devices (computations needed? memory requirements? power measurements of deployed models?).

So far, our work to date has focused on a demonstration of offline drowsiness detection that can eventually be deployed on-chip. The reviewer brings up an important point that even still, our algorithms and models must have reasonable computation requirements to be deployed. Memory requirements can provide a reasonable estimation of complexity and deployability. We have added estimates of memory requirements calculated with python's pympler package to section 4.i. All of which are well within the capacity of commonly used microcontrollers.

Memory requirements were estimated for each model using python's pympler package. The logistic regression, SVM, and RF models required 2.8 kB, 144.2 kB, and 63.8 kB respectively. These memory requirements are well within the capacity of modern microcontroller's embedded memories (e.g. 32 bit ARM Cortex-M).

Furthermore, the summary has been updated to clarify that on-chip integration is a future step for the presented work.

4. General Comments: I suggest to improve the comparison to state of the art. The main claims of the paper cover both electrodes design, electronics/readout platform design, ML models. For each one of these contributions, a discussion about how they compare to SoA is recommended. The key novelty and novel contributions of the paper should also be better stated and highlighted (e.g. at the end of the introduction), it's not clear if the authors are claiming that they improved state of the art for each individual element of the chain, or if the novelty lies "only" in the merging of all these elements (without claiming them to be state-of-the-art on their own).

Thank you for raising this excellent point. Firstly, we have improved the introduction to have a much more complete discussion of the SoA. We have brought in significantly more references and discussed how all of these works present important improvements for different aspects (electrode technology, hardware integration, etc.). The comparison table in the summary has also been modified to have an improved comparison.

Additionally, the authors believe that the novelty of this work lies in improved dry-electrode fabrication, the merging of existing elements in a new application, and a demonstration of the system's functionality through a user-study. To clarify this point, we have changed the language in the last paragraph of the introduction to better explain the novelty of this work.

This project is the first integration and demonstration of wireless, dry-electrode in-ear ExG sensors being used for drowsiness classification. This body of work presents a complete system inclusive of novel sensor manufacturing, adapted electronic design, use of open-source machine learning classification, and a 9-subject study.

5. Abstract: The discussion spans from ambulatory to use in real-life conditions (e.g. cabin sensors for operators). What is the real goal of the paper? consumer/real-life applications or medical/ambulatory assessment? It should be better clarified.

Thank you for pointing this out. The authors agree that the abstract and manuscript would benefit from more precise language. The goal of the paper is to motivate and improve the state of the art for discreet neural wearables for use in consumer/real-life applications. 'Ambulatory' was meant to describe walking and itinerant environments. We have removed the use of the word 'ambulatory' throughout the title, abstract, and manuscript to make this clearer.

6. Abstract: I suggest to include a bit more details in the abstract. E.g. simply saying "customized wireless electronics", "low-complexity ML" does not give the idea, to potential readers, about what is the added value and contribution beyond SoA of this paper. The abstract reports some quantitative numbers, but does not give an indication of the overall comparison to SoA.

Thank you for the suggestion. The reviewer is correct in that the abstract should include a clearer description of the work performed, the novelty, and comparison to the state of the art. We have rewritten the corresponding parts of the abstract to be much clearer and explain the goals of this paper. The new, updated, abstract is below:

Wireless, neural wearables can enable life-saving drowsiness, cognitive, and health monitoring for heavy machinery operators, pilots, and drivers. While existing systems use in-cabin sensors to alert operators before accidents, wearables may enable monitoring across many user environments. Current neural wearables are promising but limited by consumable electrodes and bulky, wired electronics. To improve neural wearable usability, scalability, and enable discreet use in daily and itinerant environments, this work showcases the end-to-end design of the first wireless, in-ear, dry-electrode drowsiness monitoring platform. The proposed platform integrates additive manufacturing processes for gold-plated dry electrodes, user-generic earpiece designs, wireless electronics, and low-complexity machine learning algorithms. To evaluate the platform, thirty-five hours of ExG data were recorded across nine subjects performing repetitive drowsiness-inducing tasks. The data was used to train three, offline classifier models (logistic regression, support vector machine, and random forest) and evaluated with three training regimes (user-specific, leave-one-trial-out, and leave-one-user-out). The support vector machine classifier achieved an average accuracy of 93.2% while evaluating users it has seen before and 93.3% when evaluating a never-before-seen user. These results demonstrate, for the first time, that dry, 3D printed, user-generic electrodes can be used with wireless electronics to rapidly prototype wearable systems and achieve comparable (>90% average accuracy) to existing state-of-the-art in-ear and scalp ExG systems that utilize wet electrodes and wired, benchtop electronics. Further, this work demonstrates the feasibility of using population-trained machine learning models in future, wearable ear ExG applications focused on cognitive health and wellness tracking.

7. Introduction: I suggest to add a reference related to the first quantitative statement (number of driver drowsiness events resulting in accidents).

Thank you for the reminder. We have added three references to support the quantitative statements in our first sentence which now reads:

Drowsy driving and fatigue while operating heavy machinery can be life-threatening. It is estimated that over 16.5% of fatal vehicle accidents in the United States include a drowsy driver resulting in over 8,000 deaths and \$109 billion in damages [1]–[3].

References:

[1] C. P. Landrigan, "Driving Drowsy Commentary," 2008. [Online]. Available: http://www.vtti.vt.edu/PDF/100-Car_Fact-Sheet.pdf.

[2] N. Highway Traffic Safety Administration and U. Department of Transportation, "Crash Stats: Drowsy Driving 2015," no. October, 2017, [Online]. Available: <https://crashstats.nhtsa.dot.gov/Api/Public/ViewPublication/812446>

[3] B. C. Tefft, "The Prevalence and Impact of Drowsy Driving," Washington, DC, 2010.

8. Introduction: This section presents the limitations of other approaches for drowsiness detection, most notably the camera-based eye tracking. The issue of sunglasses preventing the tracking is discussed. However, glasses featuring cameras in the glasses frames do also exist, and they can be used for drowsiness detection. Please incorporate in the bibliography and discussion also these solutions (e.g., but not only: <https://www.mdpi.com/1424-8220/23/7/3475>).

Thank you for bringing up this related drowsiness detection solution. The reviewer is correct in that there are more elaborate vision techniques as well as other multi-modal, on-body solutions for drowsiness detection. We have updated the first paragraph of our introduction to include a more holistic discussion on different solutions.

Drowsy driving and fatigue while operating heavy machinery can be life-threatening. It is estimated that over 16.5% of fatal vehicle accidents in the United States include a drowsy driver resulting in over 8,000 deaths and \$109 billion in damages [1]–[3]. In addition to private and commercial (trucking) accidents, the National Safety Council has also cited drowsiness as the most critical hazard in construction and mining. While these deaths may be prevented with common risk assessments, fatigued individuals are often unable to recognize the full extent of their impairment before it is too late [4]. Drowsiness monitoring solutions use camera-based eye-tracking, steering trajectory sensors, or electrophysiological recording devices [5]–[7]. While they can be a good fit in automotive scenarios, eye tracking is obscured by sunglasses and other obstructions while steering sensors can be susceptible to false alarms on rough roads. User-centered recording modalities such as body-worn cameras, photoplethysmography (PPG), electrodermal activity (EDA), electrocardiography (ECG), electrooculography (EOG), and electroencephalography (EEG) are becoming increasingly popular because they are highly portable and adaptable to professional work environments [8]–[11]. These modalities have been incorporated into multiple different form-factors such as eye-tracking glasses [12], PPG/ExG tracking helmets [7], and in-ear ExG sensors [13], [14]. Of these methods, ExG generally achieves the highest drowsiness detection accuracies [15].

9. Introduction: EEG/EMG/ECG are introduced for the comparison. However, electrodermal activity is also a biomarker for drowsiness and it is not mentioned in the paper. I suggest the authors to revise the bibliography to account also for EDA.

This is related to Reviewer 2's earlier comment as well. EDA is indeed an important measurement modality that has been demonstrated as useful for drowsiness. Thank you for the suggestion and we have added EDA to our introduction (copied in the above comment's response).

10. Introduction (2nd paragraph): EEG is a general term, not just for over-the-scalp measurements, since it can be applied on the scalp (surface EEG) or below (intracranial EEG). I suggest to amend this first sentence.

The reviewer brings up an important point that requires clarification. We have now updated the first sentence of the 2nd paragraph to make it clear what we are referring to when we use 'EEG'.

Surface EEG is a safe, non-invasive method of monitoring the brain's electrical activity from the scalp.

11. Introduction (2nd paragraph): The application of EEG for epilepsy is not only based on large 10-20 setups. Wearables are also becoming more popular.

This is true and this comment has been incorporated to the betterment of our introduction. We have refined the language in the 2nd paragraph of our introduction to make this clearer. The beginning of this paragraph focuses on what EEG is and the most prevalent use (in-clinic monitoring). These systems generally use large, scalp-based electrode arrays.

Clinically, the most prevalent use of EEG is the monitoring and diagnosis of stereotyped neurological disorders related to sleep and epilepsy. These clinical systems generally use large, scalp-based, gold (Au) and silver/silverchloride (Ag/AgCl) electrode arrays.

Later in this paragraph, we focus on wearable EEG systems to promote use outside of the lab and to simplify clinical measurements.

To promote use outside the lab and simplify clinical measurements, recent wearable EEG monitoring systems [8] have focused on using dry electrodes that eliminate the use of hydrogels, integrating electronics and electrodes into a headset form factor, and software packages that allow for use in more everyday applications. These research (e.g. CGX systems and Emotiv), commercial (e.g. Muse headband and Neurosity), and hobbyist (e.g. OpenBCI and Brainbit) systems have not only demonstrated impressive EEG recordings of spontaneous and evoked neural signals but also enabled disease monitoring, brain-computer interfaces (BCIs) and meditation guidance.

12. Introduction: The literature for dry-based ear-EEG can be improved. Some relevant references and products that are missing: for custom-made earpieces (<https://mjin.cat/en/>), for generic earpieces (<https://ieeexplore.ieee.org/abstract/document/9871874>, <https://pubmed.ncbi.nlm.nih.gov/24109744/>, <https://www.naox.tech/>)

Thank you for the suggestion. We have updated the introduction to include a more encompassing discussion of not just dry-based ear-EEG, but also around the ear-EEG, and wet electrode ear-EEG as well. The related paragraphs in the introduction are much expanded and included below:

Discreet, multi-channel EEG recordings from inside the ear canal have been demonstrated [26]–[28] with recent advancements focusing on earpiece design, electrode materials, and multi-sensor arrays. The ear canal is an ideal sensor location due to its inherent mechanical stability and wealth of potential recording modalities. In-ear sensors and electrodes are well situated to record temporal lobe activity, blood oxygen saturation, head movement, and masseter muscle activity making it ideal for multi-modal sensing if high spatial coverage is not required [29], [30]. While some applications may treat muscle activity or ear canal deformation as interference signals, these signals can be useful for other general ExG workloads. It is also important to note that in and around-the-ear EEG is inherently limited in gathering spatially encoded brain-activity relative to broader scalp arrays[27], [31]. With this in mind, many successful designs have leveraged hydrogel coated on flex-pcb arrays or user-customized earpieces to record ExG features such as EOG, low-frequency EEG (1 – 30 Hz), and evoked potentials (40 – 80 Hz) [26]–[28], [32], [33]. These wet-electrode based, custom earpiece systems established the feasibility of in-ear monitoring for attention monitoring, seizure monitoring, whole night sleep monitoring, and sleep stage classification [34]–[37]. Due to their user customized approach, earpieces require a case-by-case integration schemes to minimize earpiece volume resulting in variable electrode positioning. The required skin-preparation and hydrogel also can lead the conductive bridging between electrodes, limit-user-comfort, and reduced electrode lifetime [38]. The next step to more scalable deployment of in-ear ExG recordings would be the utilization of one-size-fits-most (user-generic) earpiece designs, dry electrodes, wireless electronics, and electrode materials that do not require maintenance.

Recent user-generic earpieces equipped with wet electrodes, dry electrodes[39]–[42], PPG, and/or chemical sensors have achieved high degrees of accuracy for brain-state and activity classification [39], [40], [43]–[46]. Additionally, dry-electrode based in-ear ExG have recorded low frequency neural rhythms, evoked potentials, and EOG comparable to wet-electrode. While potentially more susceptible to noise due to higher electrode-skin impedance (ESI) interfaces [47], dry electrodes eliminate the use of hydrogel, simplify the earpiece application process, and can improve user comfort. To achieve a middle ground between comfort and low ESI, state-of-the-art dry electrodes employ a wide range of solutions ranging from exotic materials, conductive composites, capacitive interfaces, solid-gels, and high-surface area 3D electrodes (microneedles, fingers, and nanowires) [20], [40], [41], [48]–[56]. PEDOT:PSS and IrO₃ are commonly

used in the small-scale production of rigid electrodes due to their superior conductivity and faradaic interfaces [57]–[59]. Both materials promote charge transfer by leveraging doped surfaces and high effective surface areas. Conductive, flexible composites, such as silvered-glass silicone and carbon-infused silicone, are not as conductive as PEDOT:PSS and IrO₃ but offer significantly greater comfort. Conductive composites are made from polymers or elastomers that can be molded into arbitrary shapes for anatomically fit electrodes and use added conductive particles to achieve a desirable ESI. The more conductive particles that are added will ultimately limit polymer cross-linking and may lead to cracking over time [60]. The clinical and industry standard materials are silver/silver chloride (Ag/AgCl) and gold due to their cost, biocompatibility, and electrical properties. Ag/AgCl can be painted on 3D electrodes to form consistent, faradaic, low-impedance interface through hair and grime. Furthermore, Ag/AgCl is also popular for consumable electrodes since the conductive particles deplete over time [61]. Gold electrodes are more inert, can be repeatedly reused, and form a capacitive interface that is not reliant on added conductive ions. While potentially more susceptible to motion artifacts and interference, gold's lifetime and chemical properties make it ideal for long-lasting ExG recording systems. Most commercial wearables and existing in-ear ExG systems use Ag/AgCl, Au, or conductive composite electrodes has made it the choice material for electrophysiological recording hardware and commercial wearables [24], [62]–[64].

13. Introduction: The literature for the electronics/readout platforms also needs to be improved. Examples include but should not be limited to:
<https://ieeexplore.ieee.org/abstract/document/10189286>, <https://www.naox.tech/>

We have significantly increased review and discussion of the electronics in the 5th paragraph of the introduction. Furthermore, the suggested references and many more were incorporated for a more complete discussion.

Electrodes are just one piece of signal acquisition. Neural recording hardware is required to digitize neural signals and transmit them to a processing unit/base-station for offline processing. Neural recording hardware for more consumer-facing products, tend to be tailor-made with low bandwidth, noise, and power specifications [65]–[67]. These devices tend to have bandwidths around 100 Hz and can achieve ultra-lower power operation (<100 μ W [67]). Research focused devices, however, utilizing high resolution and bandwidth hardware enables greater investigation outside the original project description. Such versatile systems generally support higher channel counts (16 – 64+), commercial wireless protocols (bluetooth or Wi-Fi), higher sampling rates (500 – 1000 Hz), and can take advantage of different signal modalities (e.g. EMG) at the cost of higher power (>50 mW) [46], [68], [69]. Low-noise and high-resolution systems allows for greater flexibility, repeated interpretable signal processing (frequency analysis, time-domain averaging, etc.) and algorithm development to illuminate different feature classes, mitigate interference, and discover new potential applications. Such systems have been used to build brain-machine interfaces with P300 responses and steady-state evoked potentials [27], [29], [34], [70], [71]. When adapting existing electronics for use

with wearable dry electrodes, increased ESI, system noise, and interference susceptibility bear important considerations for power requirements and any downstream machine learning algorithm [79], [80]. Employing versatile, higher power electronics with more transparent, light weight classical algorithms (e.g. logistic regression, support vector machines, random forest) is an important first step for future sensor and power optimizations. To this effect, this work uses an existing, high channel count, high bandwidth system to enable studying the relationship between the employed ExG electrode technology and drowsiness detection.

14. Introduction: "As these interfaces become more complicated and employ onboard machine learning algorithms [35]–[38] ..." not all of the papers cited here are really for on-board ML. Please focus the references on ML approaches that demonstrated deployment on edge-platforms (e.g. <https://ieeexplore.ieee.org/abstract/document/9995107>, https://openaccess.thecvf.com/content_cvpr_2017_workshops/w4/html/Reddy_Real-Time_Driver_Drowsiness_CVPR_2017_paper.html, <https://arxiv.org/abs/2111.03177>, ...)

Thank you for catching this oversight. The reviewer brings up an excellent point and we have updated the corresponding paragraph (copied below) of the introduction to highlight projects meant for on-board ML. We have also made the motivation and algorithm selection much clearer.

In addition to system-optimization, the choice of machine learning algorithm determines system functionality from the perspective of training, data, and processing requirements. Every-day ExG systems would ideally work out of the box, improve over time, and continue to provide feedback when wireless connectivity is poor and there is unreliable access to large processing power (construction sites, planes, and trucks). Classical algorithms such as logistic regression, SVMs, and random forest have demonstrated impressive success in classifying neural signals with limited datasets [25], [74], [81], [82]. Neural network-based algorithms have also achieved impressive results [83] – [86] and are good candidates for further research. Neural network-based algorithms, on average, require more training data than SVMs, logistic regression, and random forest, making them difficult to work with on smaller data sets. Furthermore, interpretable algorithms such as logistic regression and SVMs enable greater visibility into which types of features have sufficient SNR for classification and could potentially be applied to different applications. Lastly, algorithms such as SVMs, logistic regression, and random forest generally require less processing power than similarly performing neural net or perceptron-based architectures, making them ideal for low-power, edge-based deployments on existing microcontrollers. Additionally, while existing in-ear ExG BCIs have achieved high classification accuracies with user-specific training and validation [43], [84], [87]-[35], [88], ideal in-ear ExG wearables would leverage pre-trained algorithms so never-before-seen users can use these devices without time-consuming training. This user-generic classification has been explored in scalp-based drowsiness monitoring with great success but not yet with in-ear ExG [15].

15. Introduction (5th paragraph): The need for classical ML is not clear. Why do the authors argue that these approaches are needed, as compared to DL solutions? Is it for transparency? for interpretability? why do you need them in the first place? I suggest to justify it better.

The reviewer brings up a good point that warrants clarification in the manuscript. The motivation is two-fold, primarily interpretability and their reduced training data requirements. This is a more practical requirement for our own studies since gathering hours of drowsiness data for multiple subjects is difficult thus making large datasets expensive to build. Furthermore, since human-driven, time-domain analysis is difficult with ear-EEG data (lower SNR recordings relative to scalp EEG), employing an interpretable, classical models like logistic regression or SVM (with further analysis) enable greater visibility into Ear EEG features that would go unnoticed to a human technician. To address this, we have added significantly more discussion to the 6th paragraph of the introduction (copied above in response to the previous comment).

16. Introduction: The authors mention "an ambulatory EEG system must continue to provide feedback when wireless connectivity is poor and there is unreliable access to large processing power for machine learning" however this study relies on a bench-top PC, with no embedded implementation. That's a major limitation that is not matched to what appear to be the main claims of the paper (which should be better stated).

Thank you for raising this excellent point that requires clarification in the main text. The goal of this paper is to evaluate a system that could eventually be used in such purely wireless conditions. We did not mean to imply we perform embedded classification and have changed the language in the last paragraph of the introduction to better explain the novelty of this work.

This project is the first integration and demonstration of wireless, dry-electrode in-ear ExG sensors used for drowsiness classification. This body of work presents a complete system inclusive of novel sensor manufacturing, adapted electronic design, use of open-source machine learning classification, and a 9-subject study.

Furthermore, we have updated the *summary* (section 3) to specify what else is required for building a more integrated system with embedded classification.

Our results are promising for the development of the next generation of standalone wearables that can monitor brain and muscle activity in work environments and in everyday, public scenarios. To realize these standalone, wireless systems, future work requires integrating these classifiers on-chip for real-time brain-state classification and miniaturizing all the hardware into a pair of earbuds. Furthermore, the hardware would need to support online classification to actually allow for full-day, itinerant use. With further integration, a pair of ear ExG buds would significantly enable long term, daily recording ExG without interrupting a user's day or stigma.

17. Section 2a: It seems that the main novelty with respect to [45],[46] is the addition of the soft earpiece body to make electrodes more comfortable? please clarify the main novelty.

The main novelty is twofold: 1) the addition of the soft earpiece body makes the earpiece more comfortable, 2) the updated electroless plating process (addition of nickel) increases the

electrode lifetime and stability. To clarify this in the text, we have updated the *earpiece design* (section II.a.i) and *electrode fabrication* (section II.a.ii) sections to better summarize and describe the electrode novelty over previous works.

i. *Earpiece design*

Easy-to-use neural wearables require a user-generic earpiece and electrode scheme designed for recording across multiple individuals. The in-ear electrodes must make consistent contact with the skin regardless of the individual's age and be comfortable to wear for multiple hours at a time. To achieve these requirements, electrode and earpiece designs were derived from [46] and [89] and resulted in a small, medium, and large size of a single design with modular electrodes. Previous studies [30], [41] have highlighted high value electrode locations that minimize channel-to-channel correlation while maximizing mechanical stability. To also maximize electrode surface area across different individuals, small, medium, and large sized earpieces were designed with slightly differing electrode sizes. The final 'medium-sized' earpiece is comprised of four 60 mm² electrodes inside the ear canal and two 3 cm² electrodes on the ear's concha cymba and concha cavity (Figure 2a). The in-ear electrodes are cantilevers that apply gentle outward pressure to achieve lower ESI over previous iterations (370 k Ω to 120 k Ω at 50 Hz [46]) and improve mechanical stability. The out-ear electrodes act as fiducial guideposts to ensure the electrodes contact the same surface with each wear. Furthermore, electrodes outside the ear are good reference and ground candidates due to their increased distance from the brain or any muscle. To improve the earpiece assembly and further increase comfort over [46], a soft earpiece body with a manifold in-ear design was 3D printed with a clear methacrylate photopolymer (Figure 2a). Each rigid electrode is attached to this soft, elastic substrate and moves independently from the other electrodes to fit in a subject's ear. This new, modular assembly properly demonstrates the capabilities of the manifold earpiece fabrication process.

ii. *Electrode fabrication*

A low-cost, fully electroless plating process was developed to enable rapid prototyping of arbitrary shaped electrophysiological sensors. Electrodes were 3D printed with a clear methacrylate polymer (Figure 2d) and sandblasted to increase surface roughness. Samples were then submersed in different catalyst baths to develop copper, nickel, and gold metal layers. Lastly, tinned copper wires are soldered directly to the electrode surface for integration with the neural recording front end. This plating process is expanded on [45], [90] with the addition of a nickel layer that limits grain-boundary diffusion of copper and significantly extends electrode lifetime [90]–[92]. Furthermore, the nickel-plating step removes the need for repeated electroless palladium plating and the overall number of fabrication steps. While other in-ear electrodes use expensive materials like IrO₃ or hydrogels [39], [40], this improved layer stack-up (Cu, Ni, Au) is reminiscent of printed-circuit-board fabrication and enables similar levels of scale for electrode prototyping. The

final surface contains at least 0.5 μm of copper, 0.5 μm of nickel, and 0.25 μm of gold and is suitable for dry electrode recording.

18. Section 2a: Good approach of having a partially flexible earpiece that adjusts to different ear shapes. Well done. Plating process with copper, nickel, gold seems good. The claim that the addition of the nickel layer (compared to [45]) significantly extends the lifetime of the electrode should be shown experimentally (or with appropriate references).

Thank you for this comment and the kind words. We have reworded section 2.a.ii. and included appropriate references justifying how nickel can limit grain boundary from copper through gold. The updated text is below.

Electrodes were 3D printed with a clear methacrylate polymer (Figure 2d) and sandblasted to increase surface roughness. Samples were then submersed in different catalyst baths to develop copper, nickel, and gold metal layers. Lastly, tinned copper wires are soldered directly to the electrode surface for integration with the neural recording front end. This plating process is expanded on [45], [90] with the addition of a nickel layer that limits grain-boundary diffusion of copper and significantly extends electrode lifetime [90]–[92]. Furthermore, the nickel-plating step removes the need for repeated electroless palladium plating and the overall number of fabrication steps. While other in-ear electrodes use expensive materials like IrO₃ or hydrogels [39], [40], this improved layer stack-up (Cu, Ni, Au) is reminiscent of printed-circuit-board fabrication and enables similar levels of scale for electrode prototyping. The final surface contains at least 0.5 μm of copper, 0.5 μm of nickel, and 0.25 μm of gold and is suitable for dry electrode recording.

19. Section 2a: It is mentioned that the new design allows to reduce ESI. Please be quantitative, reporting comparative numbers for both the older version (also taking them from the previously published paper) and new version, to demonstrate the quantitative impact of the proposed changes on the ESI reduction.

The reviewer highlights a hole in the original manuscript that should be addressed. We have added a comparison point between the old and new designs ESI in *earpiece design* (section 2.a.i.) to better describe the effect of the improvements.

The in-ear electrodes are cantilevers that apply gentle outward pressure to achieve lower ESI over previous iterations (370 k Ω to 120 k Ω at 50 Hz [46]) and improve mechanical stability. The out-ear electrodes act as fiducial guideposts to ensure the electrodes contact the same surface with each wear. Furthermore, electrodes outside the ear are good reference and ground candidates due to their increased distance from the brain or any muscle. To improve the earpiece assembly and further increase comfort over [46], a soft earpiece body with a manifold in-ear design was 3D printed with a clear methacrylate photopolymer (Figure 2a). Each rigid electrode is attached to this soft, elastic substrate and moves independently from the other electrodes to fit in a subject's

ear. This new, modular assembly properly demonstrates the capabilities of the manifold earpiece fabrication process.

20. Section 2a: I suggest to include a summary of the key novelty and key contributions of this work as regards the electrode fabrication, with a comparison to state of the art (including but not limited to references 45,46,47).

This is a good point. To better clarify key contributions of this work's electrode fabrication process, there are qualitative and quantitative comparisons made across the introduction, *earpiece design* (section II.a.i), *electrode fabrication* (section II.a.ii), and summary. The updated text is copied above in response to Reviewer 2's comments (17 - 19).

21. Section 2a: Why are the authors introducing a lumped element model of the electrode-skin impedance? To me it is not entirely clear how this equivalent model is used and if it aids the design optimization.

This is an excellent question, and more explanation is clearly needed in the manuscript. We revised the in-ear electrode impedance section 2.a.iv. (now called "*Bioimpedance of in-ear electrodes across multiple users*") to better explain our measurements and their purpose. These models are used to better understand motion artifact settling times (associated with phase elements of the electrode skin interface) and provide important details for future analog front-end designs.

Impedance spectroscopy was used to assess in-ear electrode-skin impedance. Four subjects took impedance measurements (20 total measurements) between the in-ear electrodes and the out-ear cymba electrode. To account for future, real-life conditions with cerumen and oil, no skin preparation was performed before each trial, and measurements were repeated until all four electrodes in the ear canal were measured. Since the ESI measurements include two dry electrodes, the plotted values were divided by two to demonstrate the average ESI of a single dry electrode. All measurements were performed with an LCR meter (E4980 A, Keysight) arranged as a two-point probe where a single electrode is considered a single probe. All electrode cables were shielded by ground wires in order to minimize power-line interference. All impedance results were fitted to an equivalent circuit model (spectra shown in Fig. 3d, circuit model shown in Fig. 3e) to better understand motion artifact settling times associated with the phase elements of the electrode skin interface and provide reference for future analog front-end designs. At 50 Hz, the interface has an average impedance of 120 k Ω and phase of -33° .

22. Section 2b: I did not find it entirely clear if the authors took a pre-existing readout platform or did a custom design based on pre-existing systems (the abstract gives the impression of something custom, where they say "customized wireless electronics").

Reviewer 2 brings up a point that Reviewers 3 noted as well. This work took a pre-existing readout hardware platform and developed a custom graphical user-interface. The readout

hardware platform does utilize a ‘custom’ integrated circuit that is not commercially available, but that is not a part of this paper. We have made multiple revisions across the entire paper to clarify this (including the word ‘custom’ from the abstract). Most importantly we revised section 2.b. to clarify that this is an existing wireless recording platform (with a custom neural recording circuit).

ExG was recorded using an existing compact, wireless recording platform affixed to a headband (Figure 4a). The platform, known as the miniature, wireless, artifact-free neuromodulation device (WANDmini) [69], was originally designed for electrocorticography and comprises a custom neural recording circuit [95] (NMIC, Cortera Neurotechnologies, Inc.), a microcontroller, and a Bluetooth radio for wireless transmission.

23. Section 2b: Does the platform operate with a fully-differential or monopolar montage?

This is an excellent question and two clarifications have been added to the text. The platform operates with a monopolar montage, and section 2.b. now includes this detail.

The NMIC digitizes up to 64, fully differential channels of electrophysiological activity with a sampling rate of 1 kSps. WANDmini arranges the NMIC’s channels in a monopolar montage with a single reference electrode. This arrangement is suitable for EEG, EOG, and EMG recording and provides enough sampling headroom to remove any recording electronics related bottlenecks.

Additionally, section 2.c.i. has also been updated to clearly specify that we use a monopolar montage.

Subjects wore two earpieces with the electrodes organized in a contralateral monopolar montage. Previous works have demonstrated that electrodes on a single earpiece are sufficiently distant from each other to measure ExG [37], [41], but greater signal amplitude can be recorded with electrodes placed across both ears [39], [45].

24. Section 2b: I did not find entirely clear what are the key novelty statements of this part of the work, i.e. the readout platform design? How does this platform compare to SoA platforms for EEG acquisition/processing? (note: the comparison should not be made only towards platforms for drowsiness, since the acquisition platform is for acquiring generic EEG signals and there is no customization specific for ear-EEG signals).

The key novelty is not the readout hardware (WANDmini), but rather the overall system integration (earpiece fabrication, earpiece design, hardware, and user-study). Reviewer 3 brought up a similar concern and we have revised the manuscript’s main text to clarify why this specific hardware was used and how it was adapted for this work. In the 5th paragraph of the introduction, we discuss different hardware systems and compare their capabilities and uses:

Neural recording hardware for more consumer-facing products, tend to be tailor-made with low bandwidth, noise, and power specifications [65]–[67]. These devices tend to have bandwidths around 100 Hz and can achieve ultra-lower power operation (<100 μ W

[67]). Research focused devices, however, utilizing high resolution and bandwidth hardware enables greater investigation outside the original project description. Such versatile systems generally support higher channel counts (16 – 64+), commercial wireless protocols (bluetooth or Wi-Fi), higher sampling rates (500 – 1000 Hz), and can take advantage of different signal modalities (e.g. EMG) at the cost of higher power (>50 mW) [46], [68], [69]. Low-noise and high-resolution systems allows for greater flexibility, repeated interpretable signal processing (frequency analysis, time-domain averaging, etc.) and algorithm development to illuminate different feature classes, mitigate interference, and discover new potential applications. Such systems have been used to build brain-machine interfaces with P300 responses and steady-state evoked potentials [27], [29], [34], [70], [71]. When adapting existing electronics for use with wearable dry electrodes, increased ESI, system noise, and interference susceptibility bear important considerations for power requirements and any downstream machine learning algorithm [79], [80]...

The last paragraph of the introduction now includes mention of an ‘adapted electronic design’ to make it clear WANDmini is not a key novelty of this work:

This project is the first integration and demonstration of wireless, dry-electrode in-ear ExG sensors used for drowsiness classification. This body of work presents a complete system inclusive of novel sensor manufacturing, adapted electronic design, use of open-source machine learning classification, and a 9-subject study.

Lastly, we clarify the use of an existing system in *Lightweight ExG Recording System* (section 2.b.) as well as the original paper that demonstrated WANDmini.

ExG was recorded using an existing compact, wireless recording platform affixed to a headband (Figure 4a). The platform, known as the miniature, wireless, artifact-free neuromodulation device (WANDmini) [69], was originally designed for electrocorticography and comprises a custom neural recording circuit [95] (NMIC, Cortera Neurotechnologies, Inc.), a microcontroller, and a Bluetooth radio for wireless transmission.

25. Section 2b: Please provide more details about the data collection on the host machine. Was a custom GUI designed? (The answer to this question also relates to the question above about how much novel is this data acquisition platform).

A custom GUI was designed specifically for this application. The GUI not only records, plots, and saves all ExG data, but also contains a ‘subject’ view that presents subjects with a reaction time game and Likert queries. We have added a reference to this section in the manuscript for clarity. The *Lightweight ExG Recording System* (section 2b) now includes the following:

The host machine uses a custom graphical user interface (GUI) that plots and saves all incoming data and cues for the trail overseer. This custom GUI is unique to this work and provides the test subject with a reaction time game, auditory cues, and visual alerts during experiments. More information about the GUI is available in section 2h of the supplement.

Additionally, there are more details on the gui in the supplement, *Custom Graphical User Interface* (section 2g), copied below:

The graphical user interface (GUI) was developed in Python 3.8 in PyQt and run on the base station during the drowsiness studies. This GUI received and logged the streamed data, plotted all received data for the trial hosts, and provided subjects with the reaction time game and Likert item queries. To prevent users from preparing and “pre-clicking” ahead of time, the reaction time game prompted subjects to enter a specific randomly generated number between 0 and 9. Cues were provided every 60 seconds to minimize interruptions while maximizing drowsiness label granularity. All recorded neural data was saved in a MATLAB array format while the reaction times and Likert items were saved in a CSV format.

26. Fig 2: No justification regarding the placement of the different channels or the reference.

Thank you for highlighting this oversight! We have updated section 2.a.i in the main paper body to describe the electrode sizing, locations, and referencing options in more detail.

To achieve these requirements, electrode and earpiece designs were derived from [46] and [89] and resulted in a small, medium, and large size of a single design with modular electrodes. Previous studies [30], [41] have highlighted high value electrode locations that minimize channel-to-channel correlation while maximizing mechanical stability. To also maximize electrode surface area across different individuals, small, medium, and large sized earpieces were designed with slightly differing electrode sizes. The final ‘medium-sized’ earpiece is comprised of four 60 mm² electrodes inside the ear canal and two 3 cm² electrodes on the ear’s concha cymba and concha cavity (Figure 2a). The in-ear electrodes are cantilevers that apply gentle outward pressure to achieve lower ESI over previous iterations (370 kΩ to 120 kΩ at 50 Hz [46]) and improve mechanical stability. The out-ear electrodes act as fiducial guideposts to ensure the electrodes contact the same surface with each wear. Furthermore, electrodes outside the ear are good reference and ground candidates due to their increased distance from the brain or any muscle.

Furthermore, we have updated section 2.c.i. to include more discussion of our contralateral recording set up.

Subjects wore two earpieces with the electrodes organized in a contralateral monopolar montage. Previous works have demonstrated that electrodes on a single earpiece are sufficiently distant from each other to measure ExG [37], [41], but greater signal amplitude can be recorded with electrodes placed across both ears [39], [45].

27. Fig 4: "Drowsy events, shaded in green, are determined when a subject’s reaction time and Likert response cross a drowsiness threshold that is determined per subject." Why is the labeling corrected on a per subject basis?

This is an excellent question which was not adequately answered in the original manuscript. Unlike sleep staging, drowsiness data is not hand labelled by trained technician (a process which would inherently have some per subject bias anyway). Previous works (Barua et al. 2019, Hwang et al. 2016, etc.) all have used some reaction time or eye movement related measure that is confirmed by subjective measures (e.g. Likert score). Due to the inherent user-to-user variation of reaction time (e.g. one person's natural reaction time may simply be slower than another's) and lack of a 'ground-truth' drowsiness measure, these methods, when corrected with subjective measures, have been established as accepted methods for drowsiness labelling (Brown et al. 2014). It is important to clarify that these labels were generated before any classification code was executed. After each measurement, drowsiness labels were immediately generated and then left untouched. To clarify this, we have added explanation to section 2.c.i.

Immediately after each trial, reaction time and Likert items were thresholded per subject to automatically generate alert/drowsy labels for each trial since behavior and response time metrics are heavily correlated with drowsiness[6], [96], [97]. By taking both an objective and a subjective drowsiness measurement, high confidence data labels could be generated in face of user-error and user-bias (memory of previous KSS scores affecting subsequent scores). Both objective and subjective measures must agree to classify an event as drowsy. Furthermore, as noted in previous works, reaction times and likert scores are variable on a subject-to-subject basis. As a result, each trial was thresholded on a per subject basis.

28. Fig 6: Alpha band-power seems to rise before the subject closes the eyes. I suggest to add a comment about it in the figure caption. Why does it happen?

Thank you for pointing this out. There was a problem with the figure axis alignment and a filter artifact that made it appear as if the alpha band-power increased before subject closed the eyes. This has been corrected in the figure 6. Originally, a 5 second rolling average filter was applied to the alpha band power for clarity. It has been reduced to a 2 second rolling average filter. Any "early" activation is due to the rolling average filter and the inherent averaging of the 1000 point fourier transform in the time-frequency spectrogram.

29. Methods: Since the authors target embedded applications, the description of the whole pipeline (including feature extraction and classification) needs to be made deeper, including the assessment of computational complexity and memory requirements

Thank you for this comment. We agree and have significantly expanded on these details related to filtering, feature extraction, feature selection, and computational complexity.

First, we have described more details on our bandpass filters and pre-processing steps in *Re-referencing and Filtering* (section 4.g.).

To remove powerline interference while maintaining as much EEG activity as possible, both the recorded and re-referenced EEG channels were bandpass filtered from 0.05 - 50Hz. Filters were implemented with a 5th order butterworth high pass filter (corner of 0.05 Hz) and a 5th order Butterworth low pass filter (corner of 50 Hz). Both filters were

implemented in python but can also be implemented with infinite impulse response (IIR) filters with 16 bit registers for use in FPGA/embedded applications.

Furthermore, we have clarified what features are calculated for each channel and updated *Feature Extraction and Selection* (section 4.i) to include significantly more detail on the 20 features that are commonly selected across all models and training schemes (Table 4). As well as implementation details for common embedded applications.

Table 3: Per channel extracted features

Maximum peak-to-peak voltage
Standard deviation of voltage
Maximum PSD ($\delta, \vartheta, \alpha, \beta, \gamma$ bands)
Peak frequency ($\delta, \vartheta, \alpha, \beta, \gamma$ bands)
PSD variance ($\delta, \vartheta, \alpha, \beta, \gamma$ bands)
Absolute power ($\delta, \vartheta, \alpha, \beta, \gamma, \alpha/\beta, \vartheta/\beta, (\alpha + \vartheta)/\beta, (\alpha + \vartheta)/(\alpha + \beta)$)
Relative power ($\delta, \vartheta, \alpha, \beta, \gamma, \alpha/\beta, \vartheta/\beta, (\alpha + \vartheta)/\beta, (\alpha + \vartheta)/(\alpha + \beta)$)

All features were scaled by subtracting the median and scaling according to their interquartile range. To reduce input feature count, feature selection using an analysis of variance (scikit-learn Python 3.8) was performed to determine the top 20 features (total) that minimize redundancy and maximize class variation during training. Only these 20 features are included during model training and validation. This feature selection also implicitly selected best performing electrodes across users (most likely due to some electrodes fitting better than others). The same feature type was also selected for multiple channels (e.g. the top 20 features would include alpha band power from channel 1, 5, and 10). Contralateral channels (where sense and reference electrodes are in different ears) were always weighted higher than ipsilateral channels. The most used features are shown in Table 4.

Table 4: Top features selected for training and validation

α relative power
β relative power
δ relative power
Previous epoch's α relative power
Previous epoch's β relative power
Previous epoch's δ relative power
ϑ/β absolute power
$(\alpha + \vartheta)/\beta$ absolute power
$(\alpha + \vartheta)/(\alpha + \beta)$ absolute power

Spectral features associated with eye movement, relaxation, and drowsiness were the most important for model training. Furthermore, the previous epoch's features were also

generally important. This is corroborated by results from other works on scalp data in [14], [78], [84]. All feature extraction was performed in Python using numpy. For implementation into an embedded/FPGA environment, these features can be calculating using a coarse fast-Fourier transform, look-up-tables, and the CORDIC algorithm.

Lastly the *Machine learning models* section in Methods (section 4j) now includes memory requirements for our machine learning models.

Memory requirements were estimated for each model using python's pympler package. The logistic regression, SVM, and RF models required 2.8 kB, 144.2 kB, and 63.8 kB respectively. These memory requirements are well within the capacity of modern microcontroller's embedded memories (e.g. 32 bit ARM Cortex-M).

30. Methods: The authors mention "...determine the 20 features that minimize redundancy and maximize class variation during training". It would be nice to report what are these selected features.

The reviewer makes an excellent point that these features would be incredibly useful for any subsequent work. We have updated section 4.i. to included significantly more detail on the 20 features that are commonly selected across all models and training schemes (Table 4) and list which ones are most important (updated text is copied above in response to comment 29).

Reviewer 3:

1. The introduction part ignores around-the-ear EEG technology, a development known to fill the gap between cap-based and in-ear based EEG acquisition. It is for instance not correct to state that "existing wet electrode arrays tend to be large and delicate for everyday use". The cEEGrid flex-printed electrodes array is a wet electrodes system which has been successfully evaluated repeatedly (e.g. Debener et al., 2015, Scientific Reports, and subsequent publications from several independent research groups). Likewise, several statements throughout the introduction are not backed up by evidence and appear to express the opinion of the authors instead of summarizing recent progress in the field (e.g. "periodic maintenance", software packages requiring "extensive training"). What I miss is a more balanced introduction of in-ear dry EEG electrodes. Many researchers working in the field are aware that dry electrodes perform poorly in mobile conditions (i.e. when the participant moves). This drawback holds also for in-ear positions, because the outer ear canal deforms when individuals talk, chew, eat or drink and in-ear EEG systems provide very little spatial information in contrast to around-the-ear and cap-EEG systems (see Kappel et al., 2019, Front Neuroscience; and Meiser et al., 2020, Brain Topography), making EEG analysis much harder (note that space is a crucial dimension for EEG artifact attenuation). A disadvantage of currently available dry electrodes is that people may not tolerate dry in-ear electrodes over long periods of time, since even modest pressure can cause pain after a little while (same problem as for dry cap EEG systems). While I agree that the solution proposed in this paper is really interesting and clearly has its merits, it should be made clear that all currently available ear-EEG systems are far from perfect and far from ready for chronic (i.e. 9 to 5) use in real-life conditions.

Thank you for these notes. We agree that the introduction and comparisons between clinical, commercial, wet, and dry EEG setups would benefit from a more holistic discussion. To that effect all of these references are very helpful. We agree that there are significantly smaller form-factor wet electrode systems. Between commercial headsets (Emotiv, Muse, etc.) and arrays like the cEEGrid (which is by far the most discreet wet-electrode system we have come across) there have been a lot of advancements in the field. In fact, we would even argue that devices like the cEEGrid can achieve higher SNR recordings with fewer motion related artifacts than in-ear dry electrodes. The goal of our work is to demonstrate an in-ear dry electrode technique that is still capable of performing brain-state classification despite higher impedance electrodes and lower SNR. To address your comments and clarify our motivation, we have made a number of changes to our introduction, results, and summary.

First, we have softened the language at the beginning of introduction paragraph 2. The purpose of these sentences is simply to provide a reader with an introduction to clinical EEG.

These clinical systems generally use large, scalp-based, gold (Au) and silver/silver chloride (Ag/AgCl) electrode arrays.

We then go on to explain the working mechanism for wet electrodes and where some limitations arise. We have added additional references to support this.

Au forms a capacitive interface due to its inert nature, while Ag/AgCl forms a faradaic interface between the Ag and skin. The AgCl is a slightly soluble salt that quickly saturates the skin and forms a stable electrode-skin interface. To maintain a low-

impedance electrode-skin interface, contact is improved with tedious skin preparation and multiple technicians. While suitable for occasional, short-term monitoring, existing wet electrode arrays tend to be large and delicate for everyday use. Additionally, prolonged use of devices that require skin abrasion can result in skin irritation and lesions, further limiting their long-term use [19], [20].

Next, we expanded our discussion of EEG monitoring systems.

To promote use outside the lab and simplify clinical measurements, recent wearable EEG monitoring systems have focused on using smaller form-factor wet electrode arrays (e.g. cEEGgrid) [21] and dry electrodes that eliminate the use of hydrogels, integrating electronics and electrodes into a headset form factor, and software packages that allow for use in more everyday applications. The improved wet electrode systems (e.g. the cEEG grid) can provide unobtrusive EEG monitoring for 7+ hours, but still requires hydrogel application (limiting day-to-day use). Dry electrode systems for research (e.g. CGX systems and Emotiv), commercial (e.g. Muse headband and Neurosity), and hobbyist (e.g. OpenBCI and Brainbit) have similarly demonstrated impressive EEG recordings of spontaneous and evoked neural signals and enabled disease monitoring, brain-computer interfaces (BCIs) and meditation guidance. As these commercial systems' popularity increases, more and more wireless EEG systems are being developed and deployed across different environments [22] – [25]. The least cumbersome systems employ dry electrodes that minimize set-up time but generally still require skin cleaning and electrode surface treatments. Furthermore, the associated software packages require training to use [23], [24]. Lastly, the headset electronics are better suited for research and clinical environments as opposed to public, everyday use.

In the third paragraph of the introduction, we have expanded our discussion of in-ear EEG sensors to better highlight use in applications where high spatial coverage is not necessarily required. In addition, we added much more objective discussions on the limitation of sensors placed inside the ear canal.

Discreet, multi-channel EEG recordings from inside the ear canal have been demonstrated [26] – [28] with recent advancements focusing on earpiece design, electrode materials, and multi-sensor arrays. The ear canal is an ideal sensor location due to its inherent mechanical stability and wealth of potential recording modalities. In-ear sensors and electrodes are well situated to record temporal lobe activity, blood oxygen saturation, head movement, and masseter muscle activity making it ideal for multi-modal sensing if high spatial coverage is not required [29], [30]. While some applications may treat muscle activity or ear canal deformation as interference signals, these signals can be useful for other general ExG workloads. It is also important to note that in and around-the-ear EEG is inherently limited in gathering spatially encoded brain-activity relative to broader scalp arrays [27], [31].

2. Please state more clearly which parts of the technology are really new and what was already reported in previous studies. Since earpiece design and electrodes fabrication seem to be largely based on previous work, I suggest to remove corresponding details and report primarily essential and new information.

Thank you for raising this excellent point. We have changed the language in the last paragraph of the introduction to better explain the novelty of this work and how components are integrated together.

This project is the first integration and demonstration of wireless, dry-electrode in-ear ExG sensors used for drowsiness classification. This body of work presents a complete system inclusive of novel sensor manufacturing, adapted electronic design, use of open-source machine learning classification, and a 9-subject study.

In addition, we have added further clarification on improvements to the earpiece design and electrode fabrication process in section 2.a.i. and 2.a.ii. (copied below). These changes better summarize the novelty of the earpiece and electrodes.

To achieve these requirements, electrode and earpiece designs were derived from [42] and [85] and resulted in a small, medium, and large size of a single design with modular electrodes. Previous studies [29], [37] have highlighted high value electrode locations that minimize channel-to-channel correlation while maximizing mechanical stability. To also maximize electrode surface area across different individuals, small, medium, and large sized earpieces were designed with slightly differing electrode sizes. The final 'medium-sized' earpiece is comprised of four 60 mm² electrodes inside the ear canal and two 3 cm² electrodes on the ear's concha cymba and concha cavity (Figure 2a). The in-ear electrodes are cantilevers that apply gentle outward pressure to achieve lower ESI over previous iterations (370 kΩ to 120 kΩ at 50 Hz [42]) and improve mechanical stability. The out-ear electrodes act as fiducial guide posts to ensure the electrodes contact the same surface with each wear. Furthermore, electrodes outside the ear are good reference and ground candidates due to their increased distance from the brain or any muscle. To improve the earpiece assembly and further increase comfort over [42], a soft earpiece body with a manifold in-ear design was 3D printed with a clear methacrylate photopolymer (Figure 2a). Each rigid electrode is attached to this soft, elastic substrate and moves independently from the other electrodes to fit in a subject's ear. This new, modular assembly properly demonstrates the capabilities of the manifold earpiece fabrication process.

...

A low-cost, fully electroless plating process was developed to enable rapid prototyping of arbitrary shaped electrophysiological sensors. Electrodes were 3D printed with a clear methacrylate polymer (Figure 2d) and sandblasted to increase surface roughness. Samples were then submersed in different catalyst baths to develop copper, nickel, and gold metal layers. Lastly, tinned copper wires are soldered directly to the electrode surface for integration with the neural recording front end. This plating process is expanded on [41], [86] with the addition of a nickel layer that limits grain-boundary diffusion of copper and significantly extends electrode lifetime [86]–[88]. Furthermore,

the nickel-plating step removes the need for repeated electroless palladium plating and the overall number of fabrication steps. While other in-ear electrodes use expensive materials like IrO₃ or hydrogels [35], [36], this improved layer stack-up (Cu, Ni, Au) is reminiscent of printed-circuit-board fabrication and enables similar levels of scale for electrode prototyping. The final surface contains at least 0.5 μm of copper, 0.5 μm of nickel, and 0.25 μm of gold and is suitable for dry electrode recording.

3. The paper promises a complete monitoring platform but many details are missing. Please describe/illustrate for instance the electrodes-to-amplifier connection in detail (connector type, length of wires, position of the solder touching the skin). Regarding usability and signal quality, connectors and wiring are essential parts of a complete system. Likewise, please describe the host system used. Apparently, the amplifier was used for data acquisition and wireless transmission only, not for online signal processing, and therefore the offline monitor system requires some computing hardware (smartphone or PC for data acquisition and for offline computing).

Thank you for highlighting these missing details. We have added significantly more detail to the earpiece design and fabrication sections (copied above). Furthermore, we have an *Earpiece interconnects and wiring* description in the supplement (section 2.g.) to provide the hardware details requested.

Electrodes were connected to the recording electronics through conventional 30 AWG solid core, copper wire. Due to the quality of the copper, nickel, and gold plating, each electrode is hot soldered to a wire in a location that will not contact the subject's skin (Fig. 4s). For added mechanical stability and insulation, all solder joints were covered with UV cured epoxy. Furthermore, electrode wires were bundled with adjacent wires and placed in heat shrink tubing. Each wire was roughly 7 inches such that it could reach a subject's forehead or the corresponding point on the sagittal plane (rear of the head). Each 30 AWG wire was then soldered to a commercially available .1" jumper cable header so that it can be connected to the WANDmini's electrode inputs (female .1" headers).

Figure 4s: Earpiece close ups showcasing multiple different perspectives and mechanical feature highlights.

In addition, we have added more information on our offline signal processing throughout the *EEG Characterization and User-Generic Drowsiness Detection* results discussion (section IIc). Lastly, we added a plot of raw data in our supplement to demonstrate signal quality from both time-domain and frequency-domain perspectives (supplement section 1d).

Figure 4s: Raw ExG data in time-domain and time-frequency spectrogram from a single drowsiness trial.

4. Section IV b, there is no such thing as "artifact-free neuromodulation", the name is simply inappropriate for a research platform.

Truly artifact-free neuromodulation is indeed challenging, and we do not intend to make claims related to it in this manuscript. This phrase is part of a sentence citing previous work published in *Nature Biomedical* in 2018 ('*A wireless and artefact-free 128-channel neuromodulation device for closed-loop stimulation and recording in non-human primates*') and acknowledging the platform (known as the miniature wireless, artifact-free, neuromodulation device or WANDmini) which we used for the studies detailed in this manuscript.

5. Section IV, if ear-EEG fabrication should remain an essential part of this manuscript I would like to see long-term impedance recordings, if possible along with actigraphy. A proposed monitoring system may not have to work during gross movement, but it should tolerate gross movements (e.g. conversations, short walks etc) to an extent that signal quality recovers after those events. Without such evidence, and online system processing validation, the claim of a drowsiness monitoring system near ready for daily life application should not be made.

Thank you for the suggestion. We agree that this would be an important measurement and have prepared a figure for the supplement under *Electrode Impedance Settling* (section 2c in the supplement). A subject sat down for ~ 35 minutes with a break in the middle to walk around, jump, and make large mouth movements. The new section and figure are copied below:

In-ear dry electrodes tend to have settling times, but initial settling should be on the order of 2 minutes (based off the electrode model in figure 3e). Longer settling activity (on the order of 10's of minutes) is more likely due to sweat formation at the

electrode-skin interface. Figure 3s plots In-ear electrode impedance at 50 Hz over the course of 34 minutes of wear. The subject was seated comfortably by the LCR meter (Keysight E4980), stood up and walked around in the middle of the measurement, and then sat back down to continue measuring long term electrode impedance. Initially, ESI at 50 Hz was $\sim 610\text{ k}\Omega$ and by the end of the measurement, ESI had dropped to $\sim 510\text{ k}\Omega$. The measurement setup did not allow for simultaneous walking and impedance measurement so future work would involve monitoring impedance and EEG at the same time to track motion artifacts associated with impedance changes.

Figure 3s: In-ear electrode impedance at 50 Hz over the course of 34 minutes of wear. The subject was seated for the first half of the measurement, stood up at the 16-minute mark, walked around for a minute, and then sat back down. Impedance continues to decrease as the user sweats and the electrode interface settles.

- Section C I, the drowsiness study needs to be described in much more detail. How were the participants included into the study recruited, were they already familiar with in-ear EEG and the study purpose? Was ethical approval obtained for this work? What was the recording context, did participants sit at a table throughout the recording? Were they alone in the room, and was the recording environment a lab, an office or a home environment? What did they do in addition to the task, how much had they slept on the night before data acquisition, what time of day were the data acquired, where participants monitored by an experimenter all the time, what was their movement pattern during data acquisition., etc. Many context factors are not mentioned but could have influenced the results.

The reviewer makes an excellent point about the significance of trial conditions in these types of studies. To address these concerns, we have included significantly more detail in section 4c, d, and 4e (copied below) to explain our experimental set up, user selection, and trail conditions.

c. Subject selection and earpiece application

Nine subjects (7 male, 2 female, ages 18-27) volunteered for this study. Subjects were requested not to exercise or drink caffeine before any trial. Prior to the first experiment, subjects tried out small, medium, and large earpieces and selected the pair

they felt were most comfortable and secure in ear. During this onboarding session, subjects also familiarized themselves with the GUI.

At the start of the drowsiness trials, subjects were given their preferred ear EEG earbuds to wear, as well as an electronics headband with a fully charged Li-Po battery and the WANDmini recording hardware. To maintain a realistic daily use scenario, the subjects did not clean or prepare their skin and no hydrogel or saline was applied to the earpiece dry electrodes. The trial hosts also did not help subjects don/doff the headband or earpieces unless explicitly requested. After the experiments, the earpieces were cleaned with 70% isopropyl alcohol since they would be later used by other subjects.

d. Electrophysiological recording setup

Each earpiece has six electrodes, four inside the ear canal and two outside the ear canal. The default recording arrangement employs two contralaterally worn earpieces to maximize spatial coverage and recorded signal power [27], [39]. These two earpieces provide up to 11 ExG channels with a common reference. Either of the concha cymba electrodes can be used as a reference (the un-used one can be used as an additional sense electrode). After initial experimentation, it was determined that the right concha cymba electrode was sufficient as a reference electrode across all subjects. As a result, each ExG channel is referenced against the right concha cymba electrode in a monopolar montage (electrode Y in Figure 2A). A single wet Ag/AgCl electrode was applied to the subject's right mastoid and connected to battery ground for interference reduction.

e. Drowsiness trial overview

Subjects participated in multiple drowsiness trials to enable both user-specific and user-generic training. Subjects were not familiar with the ear EEG work when selected. No more than five trials were recorded per subject to maintain a diverse data pool. Prior to the trials, subjects were informed of the study purpose and requested to have a 'normal night's rest' (subjectively) and not drink caffeine prior to the trial. Trials took place in a quiet, indoor office space between 8am – 5pm when the lights were on. After donning the ear eeg system, the subject was left alone in the trial space until the end of the recording session. During the trial, the subject would sit at a desk in front of a laptop with a custom GUI. Subjects were instructed to only perform the reaction game task and not look at personal devices for the extent of the trial. Subjects were allowed to move their heads, readjust in their seat, and move their arms, but were asked to stay seated during the entire session (to minimize motion artifacts). Each trial was 40 – 50 minutes in length and was self-ended by the subject to prevent the interruption of a drowsy event. At the end of the trial, the subjects removed the headband and earpieces themselves. They were instructed to wait at least 24 hours before participating in subsequent drowsiness trials to maximize variation between trials.

Lastly, we also included our protocol approval in section 4.j.

The user study, subject recruitment, and all data analysis was approved by UC Berkeley's Institutional Review Board (CPHS protocol ID: 2018-09-11395).

7. Regarding drowsiness detection, it is not described how reaction time and likert scale thresholds were derived. Were thresholds adapted to optimize system performance based on all data or determined by independent data not used for the final evaluation (as it should be)? It is difficult to evaluate system performance as long as the gold standard is not convincingly presented.

Thank you for the note. Reviewer 2 had mentioned a similar note about label generation. Unlike sleep staging, drowsiness data is not hand labelled by trained technician (a process which would inherently have some per subject bias anyway). Previous works (Barua et al. 2019, Hwang et al. 2016, etc.) all have used some reaction time or eye movement related measure that is confirmed by subjective measures (e.g. Likert score). Due to the inherent user-to-user variation of reaction time (e.g. one person's natural reaction time may simply be slower than another's) and lack of a 'ground-truth' drowsiness measure, these methods, when corrected with subjective measures, have been established as accepted methods for drowsiness labelling (Brown et al. 2014). It is important to clarify that these labels were generated before any classification code was executed. After each measurement, drowsiness labels were immediately generated and then left untouched. To clarify this, we have added explanation to section 2ci

Immediately after each trial, reaction time and Likert items were thresholded per subject to automatically generate alert/drowsy labels for each trial since behavior and response time metrics are heavily correlated with drowsiness[6], [96], [97]. By taking both an objective and a subjective drowsiness measurement, high confidence data labels could be generated in face of user-error and user-bias (memory of previous KSS scores affecting subsequent scores). Both objective and subjective measures must agree to classify an event as drowsy. Furthermore, as noted in previous works, reaction times and likert scores are variable on a subject-to-subject basis. As a result, each trial was thresholded on a per subject basis.

Lastly, we have updated the *label generation* section in methods (section 4f – copied below):

Recording both objective and subjective drowsiness measures made the label generation process robust to user-error momentary distractions (when an alert user looks away from the laptop). Ear ExG samples were labelled as 'drowsy' if the user reported a drowsiness Likert item >5 and if their reaction time was more than double the average from the first 5 minutes of recording. The labels were then put through a 3-sample rolling average filter and thresholded to achieve a binary label.

8. Signal validation. EEG alpha eyes open vs eyes closed spectra are shown, but it is not stated which EEG features contributed to classification. I miss a complete description of the calculation of the EEG features (FFT?, resolution?, overlap?, windowing?, bandpass filter characteristics, etc), the feature selection process and the feature selection validation process (Figure 4 mentions ANOVA feature selection) and finally descriptive statistics which features contributed (e.g. was alpha power really discriminative?). At present all "neural" data presented in the

report are anecdotal evidence and their contribution to system performance is not clear. It is also not explained how EEG artifacts like eye blinks, movement artifacts etc. were attenuated. Please show representative raw data (maybe in supplemental material), not just processed data. Even better would be to provide the code for the offline analysis along with the paper.

Thank you for this comment. We agree and have significantly expanded on everything from filtering, to feature extraction, feature selection, and computational complexity. Ultimately, we performed as little pre-processing on the raw data as possible and relied on implicit filtering of the feature extraction process. First, we have described more details on our bandpass filters and pre-processing steps in *Re-referencing and Filtering* (section 4g).

To remove powerline interference while maintaining as much EEG activity as possible, both the recorded and re-referenced EEG channels were bandpass filtered from 0.05 - 50Hz. Filters were implemented with a 5th order butterworth high pass filter (corner of 0.05 Hz) and a 5th order Butterworth low pass filter (corner of 50 Hz). Both filters were implemented in python but can also be implemented with infinite impulse response (IIR) filters with 16 bit registers for use in FPGA/embedded applications.

Furthermore, we have clarified what features are calculated for each channel and updated *Feature Extraction and Selection* (section 4i) to included significantly more detail on the 20 features that are commonly selected across all models and training schemes (Table 4). As well as implementation details for common embedded applications.

Temporal and spectral features were extracted in Python 3.8 from the segmented ExG data. Low-complexity features were calculated for each window of ExG data and across all the recorded and re-referenced channels. Voltage standard deviation and maximum peak-to-peak voltage amplitude were calculated in the time-domain to target eye blink artifacts and motion. Welch’s method (using a 1000-point Fourier transform, 500 sample overlap, and Hamming window) was used to calculate the power spectral density (PSD) and attain frequency characteristics that relate attention and relaxation. The following spectral features were calculated prior to training: maximum PSD, peak frequency, and PSD variance were calculated for δ , ϑ , α , β , γ EEG bands. Absolute and relative band powers were also calculated for the following bands and ratios: δ , ϑ , α , β , γ , α/β , ϑ/β , $(\alpha + \vartheta)/\beta$, and $(\alpha + \vartheta)/(\alpha + \beta)$. Relative bandpower is the specific band relative to the total PSD from 0.5 – 50 Hz. Furthermore, features of the previous epoch were included to account for changes in ExG activity, since temporal and spectral features relate to characteristics that changes during the onset of drowsiness such as attention and eye movement. A complete table of features used in offline training (prior to feature selection) are below.

Table 3: Per channel extracted features

Maximum peak-to-peak voltage
Standard deviation of voltage
Maximum PSD (δ, ϑ, α, β, γ bands)

Peak frequency ($\delta, \vartheta, \alpha, \beta, \gamma$ bands)
PSD variance ($\delta, \vartheta, \alpha, \beta, \gamma$ bands)
Absolute power ($\delta, \vartheta, \alpha, \beta, \gamma, \alpha/\beta, \vartheta/\beta, (\alpha + \vartheta)/\beta, (\alpha + \vartheta)/(\alpha + \beta)$)
Relative power ($\delta, \vartheta, \alpha, \beta, \gamma, \alpha/\beta, \vartheta/\beta, (\alpha + \vartheta)/\beta, (\alpha + \vartheta)/(\alpha + \beta)$)

All features were scaled by subtracting the median and scaling according to their interquartile range. To reduce input feature count, feature selection using an analysis of variance (scikit-learn Python 3.8) was performed to determine the top 20 features (total) that minimize redundancy and maximize class variation during training. Only these 20 features are included during model training and validation. This feature selection also implicitly selected best performing electrodes across users (most likely due to some electrodes fitting better than others). The same feature type was also selected for multiple channels (e.g. the top 20 features would include alpha band power from channel 1, 5, and 10). Contralateral channels (where sense and reference electrodes are in different ears) were always weighted higher than ipsilateral channels. The most used features are shown in Table 4.

Table 4: Top features selected for training and validation

α relative power
β relative power
δ relative power
Previous epoch's α relative power
Previous epoch's β relative power
Previous epoch's δ relative power
ϑ/β absolute power
$(\alpha + \vartheta)/\beta$ absolute power
$(\alpha + \vartheta)/(\alpha + \beta)$ absolute power

Spectral features associated with eye movement, relaxation, and drowsiness were the most important for model training. Furthermore, the previous epoch's features were also generally important. This is corroborated by results from other works on scalp data in [14], [78], [84]. All feature extraction was performed in Python using numpy. For implementation into an embedded/FPGA environment, these features can be calculating using a coarse fast-Fourier transform, look-up-tables, and the CORDIC algorithm.

Lastly, we have added section 1c to the supplement to discuss raw data and show a representative example (Figure 3s).

Figure 3s showcases a whole trial's worth of ExG data in both time and frequency domain. No filters were applied to the time-domain representation while the time-frequency spectrogram was generated using a 3000 point FFT using a Hamming window with 2500 sample overlap. Frequency bins above 50 Hz were discarded for visual clarity. The time domain plot exhibits a long settling time most likely due to the large capacitance of the dry electrodes. The spectrogram showcases clear modulation in the alpha band as well as broad spectrum artifacts (most likely caused by motion). There is

significant lower frequency (<5 Hz) activity, this is most likely caused by ECG, EOG, and EEG (delta and theta band).

Figure 3s: Raw ExG data in time-domain and time-frequency spectrogram from a single drowsiness trial.

9. It is known that computationally efficient classifiers perform very well with EEG data (see Blankertz et al., 2011, Neuroimage). This should be made clear. It is however very surprising that a classifier performs better on never-before-seen users. How could this be? I do not remember coming across any EEG/BCI classification paper reporting such a finding. May it be that one, or a few users, contributed non-neural signals in an unbalanced way? Maybe some participants had different eye blink patterns for drowsy versus non-drowsy events?

Thank you for the note and reference, we have added it to our introduction. Furthermore, we agree that this is not an intuitive result at first but is ultimately related to the data partitioning/balancing as your intuition suggested. In the 'user-specific' case, models were trained and validated on a single user. In the largest data-set case that would mean that 4 hours of data would be used to train and 1 hour of data would be used to validate (and then repeated 4 times as possible to estimate average model performance with n-fold cross validation). In the 'never-before-seen user' case, 30 hours of data could be used for training and the remaining 5 hours of data (from that single user) would be used for validation. To better describe this cross-validation based partitioning scheme, which is common for small data sets, we have added further details to the main paper body as well as the supplement (both copied below).

(Section II.c.ii.) Transparent Drowsiness Classification Pipeline

...

Three cross validation techniques were used to estimate model performance across varying usage scenarios: user specific, leave-one-trial-out, and leave-one-user-out. User-specific cross validation trained models on $n-1$ trials for the subject, tested on their remaining trial, and averaged the results after n independent iterations to determine drowsiness detection accuracy for a single subject. Leave-one-trial-out cross

validation trained models on 33 of the recorded trails, tested on the remaining trial, and averaged results after all 34 independent iterations to determine the study's overall drowsiness detection accuracy. Leave-one-user-out cross validation trained on recordings from 8 subjects, tested on the remaining subject's recordings, and averaged results after all 9 independent iterations. This evaluated detection accuracy when using population-training and deploying on a never-before-seen subject. Due to the inherent imbalance between drowsy and alert classes, each classification model employed a balancing scheme where over-represented classes are given a smaller class-weight than under-represented classes. In the case of drowsy vs. alert, alert epochs are given a class weight inversely proportional to the number of epochs. This allows classes to be treated more fairly across all training/cross-validation regimes (since they will all have different class balances). During validation, class probabilities returned from the classifier models were filtered with a 3-tap Hamming window FIR filter and thresholded to achieve final binary outputs (Figure 5b).

Supplemental methods (section II.j.) Data partitioning across different training and cross-validation schemes

To understand data set requirements with our minimal drowsiness data, three different training and cross-validation regimes were devised. Each scheme had its own partitioning schemes (Fig. 6s). User-specific cross validation trained models on $n-1$ trials for the subject, tested on their remaining trial, and averaged the results after n independent iterations to determine drowsiness detection accuracy for a single subject. Leave-one-trial-out cross validation trained models on 33 of the recorded trails, tested on the remaining trial, and averaged results after all 34 independent iterations to determine the study's overall drowsiness detection accuracy. Leave-one-user-out cross validation trained on recordings from 8 subjects, tested on the remaining subject's recordings, and averaged results after all 9 independent iterations. This evaluated detection accuracy when using population-training and deploying on a never-before-seen subject. In practice, this means that 'leave-one-trial-out' and 'leave-one-user-out' trained models had access to significantly more training data than the user-specific case.

Lastly, to further confirm this partitioning hypothesis, we ran a new experiment in which we artificially limited the amount of data available to the 'leave-one-user-out' trained models (i.e. we only provided 1 trial from each subject for training and validation). In this limited data-set regime, all models performed significantly worse than the user-specific trained models with

average accuracies hovering around ~80% (compared to >91% for the user-specific models trained with similar amounts of data).

10. Along those lines, it would be more adequate to use EXG throughout the paper and avoid "neural signals" and EEG as long as the neural contribution to classification is not clearly identified and validated.

Thank you for bringing up this excellent distinction between EEG and ExG data. The reviewer is right, and we agree that ExG is a more appropriate term to use in this manuscript. We have gone through the manuscript and changed any general reference to electrophysiological monitoring to ExG.

Reviewer 4:

1. More motivation is required for the chosen machine learning models, which are in general far from current state of the art approaches. A motivation of transparent machine learning is given, and this is an important motivation, but there is no detail behind this. For example, to me, generally an SVM isn't considered a "transparent" method unless it's paired with something to explain the feature selection made and the weighting given to these. Just being less of a black box compared to deep learning approaches doesn't mean the approach is transparent.

Thank you for the note. The reviewer brings up a good point which requires further clarification in the manuscript. The initial wording of the introduction was unclear and did not provide enough context to explain our choice of algorithms. We agree that an SVM by itself is not interpretable. The logistic regression model is the main pathway of interpretability for our purposes. We were interested in SVMs and Random Forests as well because they're computationally efficient models that have already shown success with neural data and can be employed on common embedded microcontrollers in future work. To this effect, we have added significantly more discussion of this to the 6th paragraph of the introduction to motivate our model selection process.

In addition to system-optimization, the choice of machine learning algorithm determines system functionality from the perspective of training, data, and processing requirements. Every-day ExG EEG systems would ideally work out of the box, improve over time, and continue to provide feedback when wireless connectivity is poor and there is unreliable access to large processing power (construction sites, planes, and trucks). Classical algorithms that can evaluate spontaneous EEG on-chip such as logistic regression, SVMs, and random forest have demonstrated impressive success in classifying neural signals with limited datasets [25], [74], [81], [82]. Neural network based algorithms have also achieved impressive results [83]–[86] and are good candidates for further research. Generally speaking, neural network based algorithms require more compute power that is power-hungry for common embedded microcontrollers. Algorithms such as SVMs, logistic regression, and random forest generally require less processing power than similarly performing neural net or perceptron-based architectures, making them ideal for low-power, edge-based deployments on existing microcontrollers. Additionally, while existing in-ear ExG BCIs have achieved high classification accuracies with user-specific training and validation [43], [84], [87], [35], [88], ideal in-ear ExG EEG wearables would leverage pre-trained algorithms so never-before-seen users can use these devices without time-consuming training. This user-generic classification has been explored in scalp-based drowsiness monitoring with great success but not yet with in-ear ExG [15].

2. A full review isn't needed, but for fairness more commercial EEG systems than CGX, Muse and Open BCI should be listed. Well known references such as G. Niso, E. Romero, J. T. Moreau, A.

Araujo, L. R. Krol, "Wireless EEG: A survey of systems and studies," *NeuroImage*, vol. 269, no. 119774, pp. 1-17, 2023 should be included to guide the reader to other systems.

Thank you for bringing up these important references. These works are all incredibly relevant and have been added to our introduction (in addition to others). Specifically, we have expanded on the sentences related to commercial EEG systems in the 2nd paragraph of the introduction.

... To promote use outside the lab and simplify clinical measurements, recent wearable EEG monitoring systems have focused on using smaller form-factor wet electrode arrays (e.g. cEEGgrid) [21] and dry electrodes that eliminate the use of hydrogels, integrating electronics and electrodes into a headset form factor, and software packages that allow for use in more everyday applications. The improved wet electrode systems (e.g. the cEEG grid) can provide unobtrusive EEG monitoring for 7+ hours, but still requires hydrogel application (limiting day-to-day use). Dry electrode systems for research (e.g. CGX systems and Emotiv), commercial (e.g. Muse headband and Neurosity), and hobbyist (e.g. OpenBCI and Brainbit) have similarly demonstrated impressive EEG recordings of spontaneous and evoked neural signals and enabled disease monitoring, brain-computer interfaces (BCIs) and meditation guidance. As these commercial systems' popularity increases, more and more wireless EEG systems are being developed and deployed across different environments [22] – [25]. The least cumbersome systems employ dry electrodes that minimize set-up time but generally still require skin cleaning and electrode surface treatments. Furthermore, the associated software packages require training to use [23], [24]. Lastly, the headset electronics are better suited for research and clinical environments as opposed to public, everyday use. ...

3. Similar to the above, a reasonable review of electrodes is included, but references to review papers to guide readers to more comprehensive work are lacking. Potential examples include: G. Di Flumeri, P. Aricò, G. Borghini, N. Sciaraffa, A. Di Florio, F. Babiloni, The dry revolution: evaluation of three different EEG dry electrode types in terms of signal spectral features, mental states classification and usability, *Sensors* 19 (2019) 1365. Y. Fu, J. Zhao, Y. Dong, X. Wang, Dry electrodes for human bioelectrical signal monitoring, *Sensors* 20 (2020) 3651. C. Im, J.-M. Seo, A review of electrodes for the electrical brain signal recording, *Biomedical Engineering Letters* 6 (2016) 104–112.

Thank you for pointing out this oversight and for these suggested references. We have added the above works as well as other relevant papers/reviews to the 3rd and 4th paragraphs of the introduction (copied below for convenience).

Discreet, multi-channel EEG recordings from inside the ear canal have been demonstrated [26] – [28] with recent advancements focusing on earpiece design, electrode materials, and multi-sensor arrays. The ear canal is an ideal sensor location due to its inherent mechanical stability and wealth of potential recording modalities. In-ear sensors and electrodes are well situated to record temporal lobe activity, blood oxygen saturation, head movement, and masseter muscle activity making it ideal for multi-modal sensing if high spatial coverage is not required [29], [30]. While some applications may treat muscle activity or ear canal deformation as interference signals, these signals can be useful for other general ExG workloads. It is also important to note

that in and around-the-ear EEG is inherently limited in gathering spatially encoded brain-activity relative to broader scalp arrays [27], [31]. With this in mind, many successful designs have leveraged hydrogel coated on flex-pcb arrays or user-customized earpieces to record ExG features such as EOG, low-frequency EEG (1 – 30 Hz), and evoked potentials (40 – 80 Hz)[26]–[28], [32], [33]. These wet-electrode based, custom earpiece systems established the feasibility of in-ear monitoring for attention monitoring, seizure monitoring, whole night sleep monitoring, and sleep stage classification [34]–[37]. Due to their user customized approach, earpieces require a case-by-case integration schemes to minimize earpiece volume resulting in variable electrode positioning. The required skin-preparation and hydrogel also can lead the conductive bridging between electrodes, limit-user-comfort, and reduced electrode lifetime [38]. The next step to more scalable deployment of in-ear ExG recordings would be the utilization of one-size-fits-most (user-generic) earpiece designs, dry electrodes, wireless electronics, and electrode materials that do not require maintenance.

Recent user-generic earpieces equipped with wet electrodes, dry electrodes[39]–[42], PPG, and/or chemical sensors have achieved high degrees of accuracy for brain-state and activity classification [39], [40], [43]–[46]. Additionally, dry-electrode based in-ear ExG have recorded low frequency neural rhythms, evoked potentials, and EOG comparable to wet-electrode. While potentially more susceptible to noise due to higher electrode-skin impedance (ESI) interfaces [47], dry electrodes eliminate the use of hydrogel, simplify the earpiece application process, and can improve user comfort. To achieve a middle ground between comfort and low ESI, state-of-the-art dry electrodes employ a wide range of solutions ranging from exotic materials, conductive composites, capacitive interfaces, solid-gels, and high-surface area 3D electrodes (microneedles, fingers, and nanowires) [20], [40], [41], [48]–[56]. PEDOT:PSS and IrO₃ are commonly used in the small-scale production of rigid electrodes due to their superior conductivity and faradaic interfaces [57]–[59]. Both materials promote charge transfer by leveraging doped surfaces and high effective surface areas. Conductive, flexible composites, such as silvered-glass silicone and carbon-infused silicone, are not as conductive as PEDOT:PSS and IrO₃ but offer significantly greater comfort. Conductive composites are made from polymers or elastomers that can be molded into arbitrary shapes for anatomically fit electrodes and use added conductive particles to achieve a desirable ESI. The more conductive particles that are added will ultimately limit polymer cross-linking and may lead to cracking over time [60]. The clinical and industry standard materials are silver/silver chloride (Ag/AgCl) and gold due to their cost, biocompatibility, and electrical properties. Ag/AgCl can be painted on 3D electrodes to form consistent, faradaic, low-impedance interface through hair and grime. Furthermore, Ag/AgCl is also popular for consumable electrodes since the conductive particles deplete over time [61]. Gold electrodes are more inert, can be repeatedly reused, and form a capacitive interface that is not reliant on added conductive ions. While potentially more susceptible to motion artifacts and interference, gold's lifetime and chemical properties make it ideal for long-lasting ExG recording systems. Most commercial wearables and existing in-ear ExG systems use Ag/AgCl, Au, or conductive composite electrodes has made it the choice

material for electrophysiological recording hardware and commercial wearables [24], [62]–[64].

4. Information on how the E4980 was isolated during bioimpedance measurements should be added. There is no section on bioimpedance in the methods.

Thank you for the suggestion. We have updated section II.a.iv (now called '*Bioimpedance of in-ear electrodes across multiple users*') to include more important set up and isolation (copied below):

Impedance spectroscopy was used to assess in-ear electrode-skin impedance. Four subjects took impedance measurements (20 total measurements) between the in-ear electrodes and the out-ear cymba electrode. To account for future, real-life conditions with cerumen and oil, no skin preparation was performed before each trial, and measurements were repeated until all four electrodes in the ear canal were measured. Since the ESI measurements include two dry electrodes, the plotted values were divided by two to demonstrate the average ESI of a single dry electrode. All measurements were performed with an LCR meter (E4980 A, Keysight) arranged as a two-point probe where a single electrode is considered a single probe. All electrode cables were shielded by ground wires in order to minimize power-line interference. All impedance results were fitted to an equivalent circuit model (spectra shown in Fig. 3d, circuit model shown in Fig. 3e) to better understand motion artifact settling times associated with the phase elements of the electrode skin interface and provide reference for future analog front-end designs. At 50 Hz, the interface has an average impedance of 120 k Ω and phase of -33° .

5. References supporting the validity of doing repeated KSS scores every 5 minutes should be added. A priori I would assume that the scores are highly correlated, and potentially bias one another as they are close enough together in time to easily remember the previous score. It is not clear how this and similar effects were accounted for.

This is an excellent point which can be better clarified in the manuscript. Unlike sleep staging, drowsiness data is not hand labelled by trained technician (a process which would inherently

have some per subject bias anyway). Previous works (Barua et al. 2019, Hwang et al. 2016, etc.) all have used some reaction time or eye movement related measure that is confirmed by subjective measures (e.g. Likert score). Due to the inherent user-to-user variation of reaction time (e.g. one person's natural reaction time may simply be slower than another's) and lack of a 'ground-truth' drowsiness measure, these methods, when corrected with subjective measures, have been established as accepted methods for drowsiness labelling (Brown et al. 2014). As far as the high correlation between subsequent scores, that is indeed a risk. Prior to running our user-study we ran smaller tests with different query frequencies and found 60 second to 5 minutes resulted in the fewest disruptions (users also reported they did not remember their previous responses). To support this experimental design and labeling method, references have been added to section 2.c.i.

Every five minutes, the user was prompted to enter a Likert item according to the Karolinska Sleepiness Scale (KSS). This scale is frequently used to evaluate subjective sleepiness and ranges from 0 = "extremely alert", to 10 = "extremely sleepy, fighting to stay awake" [92]. Queue intervals (60 seconds and 5 minutes) were selected based off initial experimentation and previous works that demonstrated a balance between minimizing disturbances and frequent datapoints [41], [93].

Additionally, we have added more text in section 2.c.i. to highlight the importance of using subjective and objective scores, since this design helps us account for the errors mentioned in the reviewer's comment.

By taking both an objective and a subjective drowsiness measurement, high confidence data labels could be generated in face of user-error and user-bias (memory of previous KSS scores affecting subsequent scores). Both objective and subjective measures must agree to classify an event as drowsy. Furthermore, as noted in previous works, reaction times and likert scores are variable on a subject-to-subject basis. As a result, each trial was thresholded on a per subject basis.

6. EEG frequency bands were determined using Welch's method. There is no information on the settings used for this; the window size, the overlap, the number of FFT points, the window function used, and so on.

The reviewer makes an excellent point (reviewers 2 and 3 brought up similar points as well) and these details are important to include in the manuscript. The text in section 4.h. (*feature selection and extraction*) has been updated to include this.

Welch's method (using a 1000 point fft, 500 sample overlap, and hanning window) was used to calculate the power spectral density (PSD) and attain frequency characteristics that relate attention and relaxation.

7. It is only vaguely stated which features are used in the machine learning analysis. The text states "Time-domain features including..." and "Spectral features including..." and so doesn't actually give a list of what was used, and whether there are more (or less as not all were applied to all channels) than those mentioned explicitly.

Thank you for pointing out this vague language. The original manuscript had a typo implying more features were included but that was not the case. To better clarify our feature extraction methods, we have revised the text in section 4.h. and added a feature table (Table 3) Additionally, we have included significantly more detail on the 20 features that are commonly selected across all models and training schemes - these 20 features are selected during training and used for validation (Table 4).

Voltage standard deviation and maximum peak-to-peak voltage amplitude were calculated in the time-domain to target eye blink artifacts and motion.

The following spectral features were calculated prior to training: maximum PSD, peak frequency of maximum PSD, and PSD variance were calculated for δ , ϑ , α , β , γ EEG bands. Absolute and relative band powers were also calculated for the following bands and ratios: δ , ϑ , α , β , γ , α/β , ϑ/β , $(\alpha + \vartheta)/\beta$, and $(\alpha + \vartheta)/(\alpha + \beta)$. Relative bandpower is the specific band relative to the total PSD from 0.5 – 540 Hz. Furthermore, features of the previous epoch were included to account for changes in ExG activity, since temporal and spectral features relate to characteristics that changes during the onset of drowsiness such as attention and eye movement. A complete table of features used in offline training (prior to feature selection) are below.

Table 3: Per channel extracted features

Maximum peak-to-peak voltage
Standard deviation of voltage
Maximum PSD (δ, ϑ, α, β, γ bands)
Peak frequency (δ, ϑ, α, β, γ bands)
PSD variance (δ, ϑ, α, β, γ bands)
Absolute power (δ, ϑ, α, β, γ, α/β, ϑ/β, $(\alpha + \vartheta)/\beta$, $(\alpha + \vartheta)/(\alpha + \beta)$)
Relative power (δ, ϑ, α, β, γ, α/β, ϑ/β, $(\alpha + \vartheta)/\beta$, $(\alpha + \vartheta)/(\alpha + \beta)$)

All features were scaled by subtracting the median and scaling according to their interquartile range. To reduce input feature count, feature selection using an analysis of variance (scikit-learn Python 3.8) was performed to determine the top 20 features (total) that minimize redundancy and maximize class variation during training. Only these 20 features are included during model training and validation. This feature selection also implicitly selected best performing electrodes across users (most likely due to some electrodes fitting better than others). The same feature type was also selected for multiple channels (e.g. the top 20 features would include alpha band power from channel 1, 5, and 10). Contralateral channels (where sense and reference electrodes are in different ears) were always weighted higher than ipsilateral channels. The most used features are shown in Table 4.

Table 4: Top features selected for training and validation

α relative power
β relative power
δ relative power
Previous epoch's α relative power

Previous epoch's β relative power
Previous epoch's δ relative power
ϑ/β absolute power
$(\alpha + \vartheta)/\beta$ absolute power
$(\alpha + \vartheta)/(\alpha + \beta)$ absolute power

Spectral features associated with eye movement, relaxation, and drowsiness were the most important for model training. Furthermore, the previous epoch's features were also generally important. This is corroborated by results from other works on scalp data in [14], [78], [84]. All feature extraction was performed in Python using numpy. For implementation into an embedded/FPGA environment, these features can be calculating using a coarse fast-fourier transform, look-up-tables, and the CORDIC algorithm.

8. The manuscript is incomplete in a number of places. There is a hanging sentence "Sample alpha attenuation response". A "Nikon XX microscope" was used. "Sensitivity represents the true positive rate or the number of correctly classified drowsy events (REF)" and a number of similar REF statements rather than an actual reference.

Thank you for catching these errors. A few of our final edits did not save correctly in the supplement and paper! We have corrected the supplement in section 2.b. and 2.j.

Microscopy images were taken of each sample using a Nikon Eclipse 50i microscope under 20x magnification, while stylus profilometry measurements were taken with a Dektak stylus profilometer.

Sensitivity represents the true positive rate or the number of correctly classified drowsy events. Similarly, specificity represents the true negative rate or the number of correctly classified alert events. Accuracy is determined by the total number of correctly classified events. Since there is an unequal number of alert and drowsy epochs in this data set, each of these performance metrics is significant to understanding the effectiveness of the drowsiness detection system [7],[16], [50].

The dangling sentence "Sample alpha attenuation response" has been deleted from section 2.c.ii.

Lastly, we have fixed all the remaining 'REF' statements.

9. It is stated that "classification accuracy was significantly higher and ranged from 91.4% to 93.2%" and "The 10 second feature windows result in significant performance loss" with no details on the statistical tests used.

Thank you for the note. The confusion was due to imprecise language in the manuscript. We merely compare the arithmetic mean of the results from each training/validation scheme. The word, 'significantly' has been removed as to not imply anything outside a straight comparison.

The overall average of the user-specific classification results ranged from 77.9% to 92.2% across all models and feature window sizes. In the user-generic leave-one-trial-

out case, average classification accuracy was higher and ranged from 91.4% to 93.2% when cross-validating across the 34 trials.

10. The contribution of the WANDmini device was not clear. The abstract and introduction implied that a key contribution was considering all parts of the system together, but with 64 channels this EEG unit appears poorly matched for this application. Is it a new contribution here, or just what was selected to use? If the latter, why not use a lower power, lower channel count device?

As reviewer 2 also noted, this was unclear in the original manuscript. This work took a pre-existing readout hardware platform and developed a custom graphical user-interface. The readout hardware platform does utilize a 'custom' integrated circuit that is not commercially available, but that is not a part of this paper. We have made multiple revisions across the entire paper to clarify this (including the word 'custom' from the abstract). Most importantly we revised section 2.b. to clarify that this is an existing wireless recording platform (with a custom neural recording circuit).

ExG was recorded using an existing compact, wireless recording platform affixed to a headband (Figure 4a). The platform, known as the miniature, wireless, artifact-free neuromodulation device (WANDmini) [69], was originally designed for electrocorticography and comprises a custom neural recording circuit [95] (NMIC, Cortera Neurotechnologies, Inc.), a microcontroller, and a Bluetooth radio for wireless transmission.

WANDmini was ultimately a good fit because when the system was originally designed, we did not know how many channels would be required. In the end, the readout power is dominated by Bluetooth transmission and the unused channels do not have a large effect.

11. The abstract states "low-complexity machine learning algorithms". From this, I was perhaps expecting the algorithms to be running online in the hardware, but this is minor. More importantly, there is no exploration of this in the article. No measures of run-time or complexity are included. Whilst probably correct, most AI accelerators are focusing on deep neural networks, and so I don't think it is necessarily true that the methods chosen will actually be lower power when running online compared to more modern approaches which can make use of dedicated hardware and software accelerators.

Thank you for bringing up this concern. It is true that modern accelerators can achieve really low power and beat general purpose embedded systems. The goal of this work is to demonstrate

brain-state classification that could eventually be deployed and altered on embedded systems with more commercially oriented microcontrollers. In addition to clarifying this aim in the introduction we have also added implementation requirements throughout the paper. The feature extraction and selection section includes important implantation details (Fourier transform requirements + potential use of the CORDIC algorithm). We have added some analysis of memory requirements in section 4.i. to address complexity and emphasized this goal.

Memory requirements were estimated for each model using python's pympler package. The logistic regression, SVM, and RF models required 2.8 kB, 144.2 kB, and 63.8 kB respectively. These memory requirements are well within the capacity of modern microcontroller's embedded memories (e.g. 32 bit ARM Cortex-M).

12. No information on the data balancing is given. How equally sized are the drowsy and non-drowsy datasets?

The reviewer brings up an excellent point about the importance of balancing data for these types of studies. There is an inherent imbalance between alert and drowsy data since users spend significantly more time alert than drowsy. Additionally, each training regime will include a different subset of data and therefore a different number of alert and drowsy events. We account for this imbalance in the dataset using two methods. First, each classifier model uses a balancing function. The text in section 2.c.ii. has been updated to describe this.

Due to the inherent imbalance between drowsy and alert classes, each classification model employed a balancing scheme where over-represented classes are given a smaller class-weight than under-represented classes. In the case of drowsy vs. alert, alert epochs are given a class weight inversely proportional to the number of epochs. This allows classes to be treated more fairly across all training/cross-validation regimes (since they will all have different class balances).

We also account for the imbalance in alert and drowsy data sets by reporting sensitivity and specificity of each classifier (in addition to accuracy). Since there are significantly fewer drowsy events, sensitivity is a very important metric that captures the number of drowsy events detected over the total number of drowsy events in the data set. These metrics are described in the supplement section 2.i.

Sensitivity represents the true positive rate or the number of correctly classified drowsy events. Similarly, specificity represents the true negative rate or the number of correctly classified alert events. Accuracy is determined by the total number of correctly classified events. Since there is an unequal number of alert and drowsy epochs in this data set, each of these performance metrics is significant to understanding the effectiveness of the drowsiness detection system [7],[16], [50].

Lastly, we added a *Data partitioning across different training and cross-validation schemes* section in Supplemental methods (section II.j.), complete with a graphic showcasing how trails were partitioned.

To understand data set requirements with our minimal drowsiness data, three different training and cross-validation regimes were devised. Each scheme had its own partitioning schemes (Fig. 6s). User-specific cross validation trained models on $n-1$ trials for the subject, tested on their remaining trial, and averaged the results after n independent iterations to determine drowsiness detection accuracy for a single subject. Leave-one-trial-out cross validation trained models on 33 of the recorded trails, tested on the remaining trial, and averaged results after all 34 independent iterations to determine the study's overall drowsiness detection accuracy. Leave-one-user-out cross validation trained on recordings from 8 subjects, tested on the remaining subject's recordings, and averaged results after all 9 independent iterations. This evaluated detection accuracy when using population-training and deploying on a never-before-seen subject. In practice, this means that 'leave-one-trial-out' and 'leave-one-user-out' trained models had access to significantly more training data than the user-specific case.

Figure 6s: Data partitioning between training and validation sets for different cross-validation schemes.

13. The article states "Both the recorded and re-referenced EEG channels were bandpass filtered from 0.05-50Hz since these frequencies are associated with drowsy and attentive EEG activity." What filter (order, type) was used? Moreover, 0.05 - 50 Hz basically covers the whole EEG range, and so the justification is not suitable. Small, medium and large electrodes are mentioned in the supplementary results, but there are no details on them apart from for the medium. What is the change in electrode size? What is the expected change in signal quality (if any meaningful) as a result?

Thank you for this comment. The original language was unclear and not all filtering details were provided. Signals are bandpass filtered to isolate the whole EEG range. This includes frequencies associated with alert and drowsy activity, but bandpass filtering is certainly not meant to isolate them specifically. The text in section 4.f. has been updated to clarify this and include necessary filtering details.

Both the recorded and re-referenced EEG channels were bandpass filtered from 0.05-50Hz. Filters were implemented with a 5th order butterworth high pass filter (corner of 0.05 Hz) and a 5th order Butterworth low pass filter (corner of 50 Hz). Both filters were implemented in python, but can also be implemented with infinite impulse response (IIR) filters with 16 bit registers for use in FPGA/embedded applications.

The supplement has also been updated to include electrode areas and changes in signal quality in section 1.a.

Small and large earpieces were generated by scaling key earpiece features (ear canal aperture, isthmus, and length) by the standard deviations reported in [45]. The small in-ear and out-ear electrodes were 50 mm² and 2cm², respectively. The large in-ear and out-ear electrodes were 65mm² and 4 cm², respectively. Electrode impedances across earpiece sizes were roughly equivalent, implying the electrodes are not making full contact. This is an acceptable outcome, because it is likely that electrodes will contact different ear canals in different ways. Maximizing electrode surface area is a way to guarantee that at least some part of the electrode will make reasonable contact (<1 MΩ) regardless of the subject.

REVIEWER COMMENTS

Reviewer #1 (Remarks to the Author):

The revised manuscript demonstrates a notable effort in addressing the concerns raised by the reviewer. It is acknowledged that the authors have thoroughly discussed recent developments in the field of in-ear monitoring sensors, complementing their discussion with empirical evidence to substantiate the claims presented. This approach has significantly elevated the quality of the manuscript. To fully align with the rigorous publication standards of Nature Communications, the reviewer recommends that the authors address a few remaining minor issues.

1. The authors should more comprehensively discuss the potential challenges and applicability of the ear sensor in diverse populations. Specifically, the limitations of the current study, which focuses on a narrow age range, need addressing. It would be beneficial if the authors could hypothesize how the device might perform with younger subjects and propose potential solutions or modifications that could make the algorithm more effective for older subjects with abnormal sleep patterns.

2. The authors are encouraged to provide a clearer comparative analysis of the three models—logistic regression, SVM, and random forest—used in the study. Given that all models exhibit high accuracy, the manuscript would benefit from a more detailed discussion on which model might be the most effective or preferable, considering various factors such as computational efficiency, scalability, and real-world applicability.

3. The rationale behind the selection of the WANDmini device, despite its apparent bulkiness, needs to be explicitly stated. The authors should also provide details on the power consumption characteristics of this system, as this is a critical factor for its feasibility in long-term monitoring scenarios.

4. While the manuscript claims that the ear canal location minimizes motion artifacts, the reviewer raises a valid concern about their complete elimination, particularly during head movements. It would be beneficial if the authors could identify a safety threshold or parameters within which EEG recording and accuracy remain reliable, even in the presence of motion artifacts.

Reviewer #2 (Remarks to the Author):

I thank the authors for the corrections, which improved the quality of the manuscript. I still have the following comments and concerns:

1) About the reply to reviewer 1 comment 2: I agree that it would have been interesting to see a comparison of the EEG performance to a reference. At the same time, it's clear that a practical limitation would be if a comparative system is not available in the lab of the researchers. However, the authors state that "we cannot re-run EEG experiments", and I do not understand the reason of this statement. Why is it not possible to make new EEG experiments? Are the reported results not repeatable/reproducible under similar experimental conditions? Can the authors please clarify this point?

2) the authors state that the novelty is "a complete system inclusive of novel sensor manufacturing, adapted electronic design, use of open source ML classification, and a 9-subject study". However, the readout hardware was pre-existing. Please adapt the wording (stating that one of the paper's main contributions is an "adapted electronic design" seems inappropriate). Some of the points where I believe the wording should be adjusted to more fairly represent the contribution are:

- sentence at the end of the introduction: "This body of work presents a complete system inclusive of novel sensor manufacturing, adapted electronic design, use of open-source machine learning classification, and a 9-subject study". This sentence could, for example, be rewritten as "This body of work presents a novel in-ear EEG sensor manufacturing, coupled to a pre-existing wireless data acquisition platform, with an open-source machine learning classification on 9-subjects".

- still at the end of the intro, the sentence "The earpieces are then coupled with wireless, discreet electronics capable of taking uninterrupted, low-noise neural measurements for over 40 hours to form a wearable, in-ear ExG system" needs to include the reference to the paper of the WAND/WANDmini system

- sentence in the abstract: "this work showcases the end-to-end design of the first wireless, in-ear, dry-electrode drowsiness monitoring platform". The work is not showing an end-to-end "design", but rather the design of one part (electrodes) and integration with other pre-existing components (WAND) in a complete system.

- sentence in the summary: "We have reported the design, fabrication, assembly, and evaluation of a wireless, wearable, in-ear ExG platform" should be adapted to reflect which parts of the work are about design/fabrication, and which parts are about integration/assembly

3) an important thing missing is a photo showing the whole system. Now, there is Figure 2 with the photo of the electrode isolated. Then Figure 4 shows only a sketch, with the actual connectivity not really visible. I suggest including a real photo of the whole setup (electrodes, cabling, headband, wandmini...) under actual experimental conditions

4) About the measurements. Given the strong focus of the work on the electrodes, providing long-term measurements (collected over multiple days) is encouraged to support the claims, to understand how fast do they degrade

5) in section 2, when the authors discuss the WANDmini, they say "System power is dominated by the Bluetooth transmission...". Please be quantitative and report here the power consumption of the system.

6) the choice for the WAND system seems to be related to the availability of this system in the author's lab. The system has a resolution of only 15bits, when EEG data are commonly sampled with better resolutions (the ADS1298 and 1299 from TI, most commonly used for EEG recordings, have 24 bit resolutions). The in-ear measurements are especially challenging due to the small inter-electrode distance, hence better resolutions are required.

7) when the authors mention the filtering to remove power line noise: I suggest to include, between brackets, the frequency of the power line noise (60 Hz, since I guess the authors are based in the U.S.). This is helpful for readers located in areas where power lines are at 50 Hz, who might find doubtful the filtering up to 50 Hz

8) About feature extraction. The authors state that only the top 20 features are used in training and validation. I would suggest adding a feature-importance analysis, to understand how much each of these 20 features contributes to the final results

9) About feature extraction. Deployment does not require only the classifier, but also the feature extractor. There seem to lack comments about the deployability of the whole feature extraction pipeline. Some features also involve complex calculations (like standard deviations), who are not trivial to be implemented on low-power MCUs. I suggest to include a discussion about the actual deployability of the whole end-to-end signal processing pipeline

10) About the ML models. I appreciate that the authors provided extra details about the model complexity, and it's nice that they indicate that they would fit on low power microcontrollers. At the same time, if possible, I would suggest the authors to consider including the deployment as part of this work (on a low-power microcontroller of their choice - ideally on the microcontroller featured on the WANDmini) so that the overall paper appears more complete.

11) About the ML models. The performance of the SVM on seen-before users is about 93%, and on never-seen-before users it's again about 93%. More in depth analyses are needed to demonstrate that the model and number of subjects on which the training was performed is sufficient to generalize across subjects. I suggest to perform progressive additions of new subjects to see at which point a saturation of the performance occurs. E.g. start with training with data of one single subject, then do training with data from two subjects, then do training with data from 3 subjects... showing how the testing performance (on unseen subjects) progresses. Essentially it means generating a plot of accuracy vs #subjects used for training.

12) About the ML models. The authors say "In the user-generic leave-one-trial-out case, average classification accuracy was higher and ranged from 91.4% to 93.2% when cross-validating across the 34 trials. This is most likely due to the increased amount of data available for training". This result gives a hint that the data collected on single subjects were not sufficient for an effective subject-specific training, doesn't it? It would be nice, similar to the comment above, to have an analysis of the required number of trials on a per-subject basis to reach best accuracies, i.e., having an accuracy vs #trials plot

Reviewer #3 (Remarks to the Author):

The authors have done a great job addressing my comments and those of the other reviewers. While some minor decisions could be debated, I am overall happy with the quality of the revisions. Two minor issues should be fixed though:

1) Figure 3s axis label should be Time (min), not Time (s).

2) My point 4 about "artifact-free neuromodulation". The authors justify this label by referring to a previous paper title ("A wireless and artifact-free 128-channel neuromodulation device...") but a clearly wrong, false label does not get any better by repeating it. Even worse, continued use may raise expectations in uneducated readers (eg journalists) the authors can never ever fulfill. Please adhere to scientific standards, not advertising. "Artifact-free" should be removed because it is plain wrong whenever real data acquisition, where ground truth is generally not known, is involved. Every even half-decent biomedical signal textbook includes discussion of the concept of noise, and in real recordings the physiologically plausible portion of the recorded signal is generally not known, hence we somehow have to estimate/validate signal (and noise, which includes, but is not limited to "artifacts"). I cannot recommend publication unless the authors discontinue using this false statement.

3) False impression of an online, closed-loop system. Similar to other reviewers, I was initially confused with the manuscript and expected, based on the paper title and figure 1, a portable closed loop (i.e. online) system running on portable hardware. Paper title (“a...monitor”) and figure 1 (showing a smartphone with actual data coming in and life feedback to the participant) are still misleading. I suggest changing the title to better guide the reader and avoid any confusion in the first place (eg “A wireless ear EEG to monitor drowsiness” or “Hardware for a wireless ear EEG drowsiness monitor” “A wireless ear EEG to monitor drowsiness”). In addition, figure 1 should be removed or at least moved to supplemental material, as this figure illustrates the envisioned solution, which is still far beyond the empirically presented work.

Reviewer #4 (Remarks to the Author):

The authors have made a wide number of edits to the paper, and certainly moved it in the right direction. While the reply to reviewers is very lengthy, in several places the actual engagement with the underlying comment is weak and the lengthy comments are on related, but tangential, issues. Thus there are still a number of places where more information is required.

1) Transparent machine learning. The word transparent now only appears twice in the article, once in a heading and once in the summary. Given this, it would probably just be better to remove it all together.

4) Isolation. No information on isolation is included in the revised text. Instead the reply refers to power-line interference which is not a concern. I would expect to see the Keysight unit being battery powered with current limiting resistors added. The E4980A can output up to 20mA in the no fault condition, and so a lot of care is needed, probably double checking that the correct current range was selected. I think it's important to include these details to help ensure others follow suitable procedures.

5) Correlated KSS scores. A lot of the reply focuses on physiological measures, which was not the question. For correlated KSS scores, it is good that a preliminary experiment was carried out. Maybe it would be simplest to include details of this in supplementary material. At present, in the modified text two references are used to support the choice. However [41] makes no reference to KSS, and [93] appears to be a mistake, that paper is on a very different topic. I've not gone to the effort of doing a

detailed search, but I'm surprised no-one in the sleep research community has done an investigation into how KSS scores correlate with different frequencies of asking, that you could cite. At a minimum you could add a brief entry on this potential limitation to the discussion.

First, the authors would like to express their gratitude to the editor and reviewers for providing further suggestions and constructive feedback to the authors. Below is the list of questions and comments raised by the reviewers alongside our responses. For the convenience of the reviewers both the modifications made to the manuscript and our responses below are shown in green.

Reviewer 1:

1. The authors should more comprehensively discuss the potential challenges and applicability of the ear sensor in diverse populations. Specifically, the limitations of the current study, which focuses on a narrow age range, need addressing. It would be beneficial if the authors could hypothesize how the device might perform with younger subjects and propose potential solutions or modifications that could make the algorithm more effective for older subjects with abnormal sleep patterns.

Thank you for pointing out this opportunity for clarification. We have added more detail to several places in the paper to address how the study could be adapted to wider demographics from an earpiece and algorithm perspective.

Section II.a.i (The earpiece design section) now includes the following sentences:

To achieve these requirements, electrode and earpiece designs were derived from [46] and [83] and resulted in a small, medium, and large size of a single design with modular electrodes. Furthermore, as individuals age, their ear canals tend to develop a corkscrew shape with the about the isthmus. Thus, by staying near the ear canal entrance and not passing the isthmus, the implemented design can be worn regardless of age. Previous studies [30], [41] have highlighted high value electrode locations that minimize channel-to-channel correlation while maximizing mechanical stability.

To extend this study further and discuss how the models could be improved, we have also updated the Section III (the summary) with the following text:

Lastly, it would be important to take this miniaturized hardware and implement a user-study with a wider demographic. By monitoring in-ear EEG across individuals aged 18 – 65+, further age specific models can be investigated. If a monolithic model is unable to classify drowsiness stereotypes across such a large age range, it would be interesting to provide models with context such as age, gender, known sleep disorders, and previous night's sleep quality.

This concept of adding user context as a basic feature is certainly feasible from a product perspective and is something we are personally excited to implement in future studies. In fact, previous age and night's sleep quality is often recorded in other drowsiness studies but small data sets have limited its impact. With easier to use wearables, we hope to run much larger studies to investigate the concept of context-aware brain classification models for drowsiness specifically. In practice, it would likely lead to different models being used for different age groups (as is commonly done in fitness tracking and VO2 max models).

2. The authors are encouraged to provide a clearer comparative analysis of the three models—logistic regression, SVM, and random forest—used in the study. Given that all models exhibit high accuracy, the manuscript would benefit from a more detailed discussion on which model might be the most effective or preferable, considering various factors such as computational efficiency, scalability, and real-world applicability.

This is a great point. We moved model details and comparisons from our methods and supplement to our results section. The results now include a section titled “classifier architecture comparison” with the following text:

Three low-complexity machine learning models were used to promote the scalability and usability of the drowsiness detection platform. All models were implemented in Python 3.8 using scikit-learn packages. Logistic regression models were implemented with a stochastic average gradient descent solver. L1 regularization was used to add a penalty equal to the absolute value of the magnitude of the feature coefficients. Support vector machines were implemented with a radial basis function (RBF) kernel to account for data that may not be linearly separable. The trained models utilized a maximum of 400 support vectors and a regularization parameter, $C=1$. Random forest models were implemented with 100 trees and a maximum depth of five to prevent overfitting. These implementations resulted in memory footprints that were estimated using python’s pympler package. The logistic regression, SVM, and RF models required 2.8 kB, 144.2 kB, and 63.8 kB respectively. These memory requirements are well within the capacity of modern microcontroller’s embedded memories (e.g. 32 bit ARM Cortex-M).

Since all three models achieve high accuracy, it is clear that drowsiness is classifiable with in-ear eeg recording. No model shows markedly greater performance or another. The logistic regression model is more computationally efficient, requires significantly less memory, and can be more easily trained/deployed with smaller datasets. It is important to verify that logistic regression continues to perform as well across larger demographics, a topic for future studies.

This also connects to the earlier comment on how further, more concrete determinations can be made after a larger user-study.

3. The rationale behind the selection of the WANDmini device, despite its apparent bulkiness, needs to be explicitly stated. The authors should also provide details on the power consumption characteristics of this system, as this is a critical factor for its feasibility in long-term monitoring scenarios.

We apologize for the confusion regarding the use of WANDmini. We have further updated section II.b. (Lightweight ExG Recording system) with the following text to make it clearer that WANDmini was designed for in-ear eeg related explorations where the heart of the system was adapted from existing work.

The platform, known as WANDmini, is a wireless neural recording frontend built for and already deployed in previous in-ear EEG studies [46]. It is adapted from a system originally designed for electrocorticography and comprises a custom neural recording circuit [69], [89] (NMIC, Cortera Neurotechnologies, Inc.), a microcontroller, and a Bluetooth radio for wireless transmission.

...

With the NMIC and WANDmini power consumptions, 700 μ W and 46 mW, respectively, a 3.7 V 550 mA battery can provide ~44 hours of runtime. In summary, the NMIC is lower power, higher channel count, and lower noise than commonly used commercial neural frontends (e.g. ADS1298/1299) making it ideal for use in channel elastic in-ear EEG prototypes.

We also updated Table 1 with the NMIC power consumption and WANDmini battery life. Lastly, Reviewer 2 also was curious why we did not use TI's ADS1298 or a similar chip from Analog Devices so we provided a general comparison in the paper (with further details in our response to Reviewer 2's comment #2).

4. While the manuscript claims that the ear canal location minimizes motion artifacts, the reviewer raises a valid concern about their complete elimination, particularly during head movements. It would be beneficial if the authors could identify a safety threshold or parameters within which EEG recording and accuracy remain reliable, even in the presence of motion artifacts.

Thank you for pointing out this oversight! It is certainly important to describe our discard criteria. Section II.c.ii (Transparent Drowsiness Classification Pipeline) now includes the following:

The training pipeline for ExG data consisted of post-processing, feature extraction, and model training steps (Figure 5a). ExG recordings were referenced to maximize spatial covering, band pass filtered, and segmented into 50 second or 10 second windows. If a window of data exhibited an artifact greater than 10 mV (from motion) it would be discarded. This was happened very infrequently as most artifacts were less than 1mV above the baseline rms voltage.

Reviewer 2:

1. About the reply to reviewer 1 comment 2: I agree that it would have been interesting to see a comparison of the EEG performance to a reference. At the same time, it's clear that a practical limitation would be if a comparative system is not available in the lab of the researchers. However, the authors state that "we cannot re-run EEG experiments", and I do not understand the reason of this statement. Why is it not possible to make new EEG experiments? Are the reported results not repeatable/reproducible under similar experimental conditions? Can the authors please clarify this point?

We apologize for any confusion our response resulted in. To make a proper comparison, we would have to re-run EEG experiments with all previous subjects (a process that took over 100 hours) to generate a similar amount of data to our Ear EEG experiments. This is possible, but impractical. Our goal in this work was to demonstrate that high accuracy classification is possible with in-ear EEG. To make a comparison to scalp EEG, we reference the following body of work in drowsiness detection from scalp EEG [1][2][3]. Furthermore, our prior work [4] makes a direct and simultaneous comparison of signal characteristics in dry Ear EEG vs. wet scalp EEG.

[1] S. Barua, M. U. Ahmed, C. Ahlström, and S. Begum, "Automatic driver sleepiness detection using EEG, EOG and contextual information," *Expert Syst Appl*, vol. 115, pp. 121–135, Jan. 2019.

[2] T. Hwang, M. Kim, S. Hong, and K. S. Park, "Driver drowsiness detection using the in-ear EEG," in *Proceedings of the Annual International Conference of the IEEE Engineering in Medicine and Biology Society, EMBS*, Institute of Electrical and Electronics Engineers Inc., Oct. 2016.

[3] S. Hong, H. Kwon, S. H. Choi, and K. S. Park, "Intelligent system for drowsiness recognition based on ear canal electroencephalography with photoplethysmography and electrocardiography," *Inf Sci (N Y)*, vol. 453, pp. 302–322, Jul. 2018.

[4] R. Kaveh *et al.*, "Wireless User-Generic Ear EEG," in *Transactions on Biomedical Circuits and Systems*, vol. 14, no. 4, pp. 727–737, Aug. 2020.

2. the authors state that the novelty is "a complete system inclusive of novel sensor manufacturing, adapted electronic design, use of open source ML classification, and a 9-subject study". However, the readout hardware was pre-existing. Please adapt the wording (stating that one of the paper's main contributions is an "adapted electronic design" seems inappropriate). Some of the points where I believe the wording should be adjusted to more fairly represent the contribution are:

Thank you for pointing these opportunities for clarification out and providing constructive feedback. We've addressed all of the highlighted areas and placed the updated text below the specific comment (as well as figures if they were generated).

- sentence at the end of the introduction: "This body of work presents a complete system inclusive of novel sensor manufacturing, adapted electronic design, use of open-source machine learning classification, and a 9-subject study". This sentence could, for example, be rewritten as "This body of work presents a novel in-ear EEG sensor manufacturing, coupled to a pre-existing wireless data acquisition platform, with an open-source machine learning classification on 9-subjects".

Thank you for the suggestion. We have now incorporated it and the second sentence in the last introductory paragraph now reads:

This project is the first integration and demonstration of wireless, dry-electrode in-ear ExG sensors used for drowsiness classification. To this effect, a novel in-ear EEG sensor manufacturing method coupled to a pre-existing wireless data acquisition platform is presented and verified with open-source machine learning classification on 9-subjects.

- still at the end of the intro, the sentence "The earpieces are then coupled with wireless, discreet electronics capable of taking uninterrupted, low-noise neural measurements for over 40 hours to form a wearable, in-ear ExG system" needs to include the reference to the paper of the WAND/WANDmini system

This is a very important point to better illustrate what was adapted for this work. We have added the reference there and the section now reads as follows:

These electrodes can replace existing solutions that rely on shorter-lifespan Ag/AgCl electrodes or expensive materials such as platinum or IrO₃. The earpieces are then coupled with wireless, discreet electronics capable of taking uninterrupted, low-noise neural measurements for over 40 hours [46] to form a wearable, in-ear ExG system.

- sentence in the abstract: "this work showcases the end-to-end design of the first wireless, in-ear, dry-electrode drowsiness monitoring platform". The work is not showing an end-to-end "design", but rather the design of one part (electrodes) and integration with other pre-existing components (WAND) in a complete system.

We have now reworded the sentence and subsequent sentence to:

To improve neural wearable usability, scalability, and enable discreet use in daily and itinerant environments, this work showcases an end-to-end system for the first wireless, in-ear, dry-electrode earpiece for monitoring drowsiness. The proposed platform integrates additive manufacturing processes for gold-plated dry electrodes, user-generic earpiece designs, wireless electronics, and low-complexity machine learning algorithms.

It no longer claims to be a novel design for an end-to-end system, rather it is an end-to-end instantiation utilizing wireless earpiece for monitoring drowsiness (which is similar to our new, amended title based off Reviewer 3's third comment).

- sentence in the summary: "We have reported the design, fabrication, assembly, and evaluation of a wireless, wearable, in-ear ExG platform" should be adapted to reflect which parts of the work are about design/fabrication, and which parts are about integration/assembly

This is also a good opportunity for clarification on our part. The summary's first sentence now reads:

We have reported the design and fabrication of in-ear dry electrodes along with the assembly, and evaluation of a wireless, wearable, in-ear ExG platform for offline drowsiness detection in never-before-seen users.

- an important thing missing is a photo showing the whole system. Now, there is Figure 2 with the photo of the electrode isolated. Then Figure 4 shows only a sketch, with the actual connectivity not really visible. I suggest including a real photo of the whole setup (electrodes, cabling, headband, wandmini...) under actual experimental conditions

This is a fair point. We have now added a full photo of our experiment setup to our supplement (supplemental methods: experimental recording setup). Along with text describing what is being shown.

Subjects comfortably sat comfortably in front of the base station radio and host laptop while wearing the WANDmini headband and ear EEG earpieces (Fig. 7s). Earpieces sizes were selected per subject based off comfort and electrode-skin impedance. The most comfortable earpieces with the lowest average impedance would be used for the rest of the trial. The headband straps were also tightened to a comfortable point where WANDmini would not move if the user moved their head.

Figure 7s: Experimental setup with WANDmini, the earpieces, base station radio, and laptop.

- About the measurements. Given the strong focus of the work on the electrodes, providing long-term measurements (collected over multiple days) is encouraged to support the claims, to understand how fast do they degradate

Thank you for pointing this out. We have used this as an opportunity to take more data showcasing how earpiece impedance does not change over time with heavy use. This results below have been added to supplement section I.b. (Electrode impedance comparisons) and showcase no noticeable change in impedance across 7 days. This is consistent with our expectation that gold is an inert metal that should not react with oils or skin detritus.

The ear-skin impedance also does not noticeably degrade over time. Electrode impedance was measured the day they were completed and one week after (Fig 2s). When comparing measurements across the same earpiece and same subject (N=4), ESI indicated no meaningful difference as expected with gold plated electrodes. It is important to note that during the week between ESI measurements, the electrodes underwent heavy use in various ExG exploratory tasks and multiple drowsiness trials, repeatably exposing then to cerumen, cleaning solution, and various other skin detritus.

Figure 2s: In ear electrode impedance magnitude and phase before and after a week of repeated use (N=4 for both measurements). Measurements were conducted using the same earpiece and same subject.

- in section 2, when the authors discuss the WANDmini, they say "System power is dominated by the Bluetooth transmission...". Please be quantitative and report here the power consumption of the system

This was an oversight. We have now updated the WANDmini discussion with power numbers, battery life, and an updated specification table (copied below):

System power is dominated by the microcontroller and Bluetooth transmission (98.3%) thus making unused channels immaterial from a power perspective. With the NMIC and WANDmini power consumptions, 700 μ W and 46 mW, respectively, a 3.7 V 550 mA battery can provide ~44 hours of runtime. In summary, the NMIC is lower power, higher channel count, and lower noise than commonly used commercial neural frontends (e.g. ADS1298/1299) making it ideal for use in channel elastic in-ear EEG prototypes. NMIC and WANDmini specifications are listed in Table 1.

Table 1: Relevant system, WANDmini, and NMIC specifications.

Maximum Recording Channels	64
Recording Channels Used	11
Reference Location	Right Cymba
Ground Location	Right Mastoid
Input Range	100 mVpp
ADC Resolution	15 bits
ADC Sample Rate	1 kSps
Noise Floor	$70nV/\sqrt{Hz}$
Wireless Data Rate	2 Mbps
NMIC Power	700 μ W
WANDmini Power	46 mW
Battery Life	44 Hours

3. the choice for the WAND system seems to be related to the availability of this system in the author's lab. The system has a resolution of only 15bits, when EEG data are commonly sampled with better resolutions (the ADS1298 and 1299 from TI, most commonly used for EEG recordings, have 24 bit resolutions). The in-ear measurements are especially challenging due to the small inter-electrode distance, hence better resolutions are required.

Thank you for bringing up this important point. The ADS 1299 and 1298 are indeed well suited for wet electrode applications. When we selected our neural front end, we paid special attention to power, system flexibility, input range, and noise floor requirements for our specific system (ADC resolution is insufficient on its own). WANDmini's neural recording circuit (Cortera NMIC [1]) is better optimized for low-power wearable prototypes, dissipating only $\sim 8 \mu\text{W}$ per channel while the ADS 1299 and 1298 dissipate 5.25 mW and 1.75 mW respectively, orders of magnitude more than the NMIC, whose lower power consumption significantly improves battery life.

WANDmini's 64 channels gave us channel count flexibility when designing the system. Of the 64, we use 11, therefore a more optimized system is certainly possible. The ADS 1299/98 only have 8 channels, therefore we would need to use two chips thereby increasing the size of the system in addition to the power dissipation. At its lowest noise setting, the ADS 1299 achieves a noise floor of $0.4 \mu\text{Vrms}$ (for a minimum bandwidth of 524 Hz, or $17.5\text{nV}/\sqrt{\text{Hz}}$) and an input range ranging of 87 mV. This dynamic range corresponds to 16 bits. Digitizing with 24 bits does not give any more information, since the ADC either covers a range higher than the input range or digitizes the noise floor.

We have shown in our previous work [2] that our dry electrodes contribute on average $\sim 2 \mu\text{Vrms}$ over 500 Hz (or $90 \text{nV}/\sqrt{\text{Hz}}$) of thermal noise, which is higher than the ADS 1299 noise floor. With the noise contributed by the electrodes, we calculate that the system can digitize 14 bits for the same maximum input. WANDmini has a noise floor of $1.6 \mu\text{Vrms}$, which is below the noise floor of the electrodes, and has a 100 mV input range, corresponding to ~ 14 bits. The 15-bit resolution of the NMIC ADCs perfectly meets the dynamic range requirements of the system. Furthermore, the entire NMIC chip consumes 700uW, which is only 0.2% of the ADS1299's power consumption. Given the noise floor contribution of the electrodes, it is not advantageous to use a recording circuit with more than 15 bits of resolution since it costs both additional power and data rate in the wireless link.

[1] B. C. Johnson *et al.*, "An implantable 700uW 64-channel neuromodulation IC for simultaneous recording and stimulation with rapid artifact recovery," *Symposium on VLSI Circuits*, 2017.

[2] R. Kaveh *et al.*, "Wireless User-Generic Ear EEG," in *Transactions on Biomedical Circuits and Systems*, vol. 14, no. 4, pp. 727-737, Aug. 2020.

To reflect this decision process, the text in section II.b. has been updated with the following:

With the NMIC and WANDmini power consumptions, 700 μW and 46 mW, respectively, a 3.7 V 550 mA battery can provide ~ 44 hours of runtime. In summary, the NMIC's significantly lower power than common commercial neural frontends (e.g. ADS1298/1299), high channel count, and sufficiently low noise floor makes it ideal for use in modular in-ear EEG prototypes. NMIC and WANDmini specifications are listed in Table 1 and further detailed in Supplement section II.h.

Furthermore, the dynamic range calculations have been detailed in Supplement section II.h and are copied below:

h. System Resolution and Noise Floor Calculations

This system was assembled to achieve a high degree of channel flexibility, low power dissipation, and most importantly, meet the required dynamic range of a dry electrode system. The dry electrode earpieces contribute an average $\sim 2 \mu\text{V}_{\text{rms}}$ over 500 Hz (500 Hz being the bandwidth of interest for ECG, EMG, EEG, and EOG). If the maximum input signal is assumed to be 100 mV peak to peak, our required dynamic range can be calculated with the following equation:

$$\text{Dynamic Range} = 10\log\left(\frac{\text{Power}_{\text{maximum signal}}}{\text{Power}_{\text{minimum signal}}}\right)$$

Assuming a sinusoidal input, the power of our maximum signal is given by:

$$\text{Power}_{\text{sinusoid}} = \left(\frac{V_{\text{amp}}^2}{2}\right)$$

Then assuming our electrode thermal noise dictates our minimum signal power, the power of our minimum signal is given by:

$$P_{\text{minimum signal}} = 2\mu\text{V}_{\text{rms}}^2$$

The resulting dynamic range (84.95 dB) can then be used to calculate the required number of bits to effectively digitize signals (effective number of bits – ENOB) across our input range using the following:

$$\text{ENOB} = \frac{\text{Dynamic Range} - 1.76}{6.02} = 13.82 \text{ bits}$$

1.76 is defined as the quantization error of an ideal ADC while 6.02 is the conversion factor between decibels (log10) to bits (log2). WANDmini, with an input range of 100 mV, $1.6 \mu\text{V}_{\text{rms}}$ noise floor, and 15 bit resolution is sufficient for the required dynamic range.

4. when the authors mention the filtering to remove power line noise: I suggest to include, between brackets, the frequency of the power line noise (60 Hz, since I guess the authors are based in the U.S.). This is helpful for readers located in areas where power lines are at 50 Hz, who might find doubtful the filtering up to 50 Hz.

This is a good point to make things clearer for readers outside North and South America! We've updated text in in Section IV.g. to the following:

To remove power-line interference (60 Hz in North America) while maintaining as much EEG activity as possible, both the recorded and re-referenced EEG channels were bandpass filtered from 0.05 - 50Hz.

5. About feature extraction. The authors state that only the top 20 features are used in training and validation. I would suggest adding a feature-importance analysis, to understand how much each of these 20 features contributes to the final results.

This would indeed help readers understand our feature selection process with more clarity. To make things classifier agnostic, Section IV.i. includes a discussion on which features were not only the least correlated with each other but also deemed more important for logistic regression. The pertinent text is copied below:

The same feature type was also selected for multiple channels (e.g. the top 20 features would include alpha band power from channel 1, 5, and 10). Contralateral channels (where sense and reference electrodes are in different ears) were always weighted higher than ipsilateral channels. The most used features (in order of importance) are shown in Table 4....

Spectral features associated with eye movement, relaxation, and drowsiness were the most important for model training. Furthermore, the previous epoch's features were also generally important.

6. About feature extraction. Deployment does not require only the classifier, but also the feature extractor. There seem to lack comments about the depoloyability of the whole feature extraction pipeline. Some features also involve complex calculations (like standard deviations), who are not trivial to be implemented on low-power MCUs. I suggest to include a discussion about the actual deployability of the whole end-to-end signal processing pipeline

Reviewer 1 was also curious about deployability limitations regarding different demographics. We have taken both of your comments to heart and expanded our summary to discuss important next steps and optimizations required to deploy a truly wireless drowsiness monitor. The updated text is below:

Our results are promising for the development of the next generation of standalone wearables that can monitor brain and muscle activity in work environments and in everyday, public scenarios. To realize these standalone, wireless systems, future work requires integrating these classifiers on-chip for real-time brain-state classification and miniaturizing all the hardware into a pair of earbuds. Furthermore, the hardware would need to support online classification to allow for full-day, itinerant use. Lastly, it would be important to take this miniaturized hardware and implement a user-study with a wider demographic. By monitoring in-ear EEG across individuals aged 18 – 65+, further age specific models can be investigated. If a monolithic model is unable to classify drowsiness stereotypes across such a large age range, it would be interesting to provide models with context such as age, gender, known sleep disorders, and previous night's sleep quality. Furthermore, the feature selection performed in this work suggests that simpler calculations such as bandpower ratios are sufficient for drowsiness classification. If this remains the case across larger demographics then feature extractors can ignore computationally expensive features such as standard deviation, different entropy measures, etc. to reduce power in embedded classification scenarios. With aforementioned integration, a pair of ear ExG buds would significantly enable long term, daily recording ExG without interrupting a user's day or

stigma. These measurements would enable an entirely new era of research for tracking long-term cognitive changes from disorders such as depression, Alzheimer's, narcolepsy, or stress.

7. About the ML models. I appreciate that the authors provided extra details about the model complexity, and it's nice that they indicate that they would fit on low power microcontrollers. At the same time, if possible, I would suggest the authors to consider including the deployment as part of this work (on a low-power microcontroller of their choice - ideally on the microcontroller featured on the WANDmini) so that the overall paper appears more complete.

Thank you again for the suggestion of providing more complexity details. We have even moved that text to a 'Classifier Architecture Comparison' section in our results (Section II.c.iii.4). As far as deployment, we agree that deployment of these classifiers directly onto a microcontroller would be very exciting future work and are currently working towards that goal.

8. About the ML models. The performance of the SVM on seen-before users is about 93%, and on never-seen-before users it's again about 93%. More in depth analyses are needed to demonstrate that the model and number of subjects on which the training was performed is sufficient to generalize across subjects. I suggest to perform progressive additions of new subjects to see at which point a saturation of the performance occurs. E.g. start with training with data of one single subject, then do training with data from two subjects, then do training with data from 3 subjects... showing how the testing performance (on unseen subjects) progresses. Essentially it means generating a plot of accuracy vs #subjects used for training.

Classifier performance vs. subjects is indeed an interesting point. To delve deeper into the minimum number of subjects required before cross-validation performance begins to saturate, we performed the suggested experiment across all three models and plotted a comparison below. Until a dataset consists of 6 subjects (5 in the testing set, 1 in the validation set), all three models struggle to achieve reasonable specificity and classify drowsy events. It's important to note that in this experiment, each subject only had a single trial's worth of data in the data set (to maintain fairness across all subjects regardless of data availability). An important next step would be to expand the demographic make-up of the subject population to include individuals older than 30 years old. This will be part of our future work.

This experiment has been added to supplemental results section I.g. with the following text:

To explore the minimum number of subjects required before cross-validation performance begins to saturate, the classifiers were trained with an increasing number of subjects in the training set (use a leave-one-user-out training/cross-validation approach). Until a dataset consists of 6 subjects (5 in the testing set, 1 in the validation set), all three models struggle to achieve reasonable specificity and classify drowsy events. It's important to note that in this experiment, each subject only had a single trial's worth of data in the data set (to maintain fairness across all subjects regardless of data availability).

Figure 10s: User-specific cross-validation accuracy as the number of subjects in the training set is increased.

9. About the ML models. The authors say "In the user-generic leave-one-trial-out case, average classification accuracy was higher and ranged from 91.4% to 93.2% when cross-validating across the 34 trials. This is most likely due to the increased amount of data available for training". This result gives a hint that the data collected on single subjects were not sufficient for an effective subject-specific training, doesn't it? It would be nice, similar to the comment above, to have an analysis of the required number of trials on a per-subject basis to reach best accuracies, i.e., having an accuracy vs #trials plot.

The authors agree that more subject-specific data would indeed improve subject-specific training/testing. To observe the effect of larger subject specific training sets, we trained classifiers on an increasing number of trials for the 4 subjects with the most data (5 trials each). Across all models, cross-validation accuracy continues to improve as the trials in the training set increase. This is consistent with our previous intuitions that subject specific training/validation performance would benefit from more data.

This experiment has been added to the supplemental results (section I.f.) with the following text:

To observe the effect of larger subject specific training sets, we trained classifiers on an increasing number of trials for the 4 subjects with the most data (5 trials each). Across all models, cross-validation accuracy continues to improve as the trials in the training set increase.

Figure 9s: User-specific cross-validation accuracy as the number of trials per user are increased.

Reviewer 3:

1. Figure 3s axis label should be Time (min), not Time (s).

Thank you for pointing this out, we have fixed this plot!

2. My point 4 about “artifact-free neuromodulation”. The authors justify this label by referring to a previous paper title (“A wireless and artifact-free 128-channel neuromodulation device...”) but a clearly wrong, false label does not get any better by repeating it. Even worse, continued use may raise expectations in uneducated readers (eg journalists) the authors can never ever fulfill. Please adhere to scientific standards, not advertising. “Artifact-free” should be removed because it is plain wrong whenever real data acquisition, where ground truth is generally not known, is involved. Every even half-decent biomedical signal textbook includes discussion of the concept of noise, and in real recordings the physiologically plausible portion of the recorded signal is generally not known, hence we somehow have to estimate/validate signal (and noise, which includes, but is not limited to “artifacts”). I cannot recommend publication unless the authors discontinue using this false statement.

Per your recommendation we have removed the terms ‘artifact-free’ and introduce WANDmini with the following text:

ExG was recorded using an existing compact, wireless recording platform affixed to a headband (Figure 4a). The platform, known as WANDmini, is a wireless neural recording frontend built for and already deployed in previous in-ear EEG studies [46]. It is adapted from a system originally designed for electrocorticography and comprises a custom neural recording circuit [69], [89] (NMIC, Cortera Neurotechnologies, Inc.), a microcontroller, and a Bluetooth radio for wireless transmission.

To clarify, the specific work in [46] was focused on removing stimulation artifacts only (the timing of which is known a priori) from neural recording with specific analog circuit and post-processing techniques. The publication does not claim to eliminate noise or any other type of artifact. In this work, we do not use the stimulation portion of WANDmini and therefore do not employ any techniques to reduce stimulation artifacts.

3. False impression of an online, closed-loop system. Similar to other reviewers, I was initially confused with the manuscript and expected, based on the paper title and figure 1, a portable closed loop (i.e. online) system running on portable hardware. Paper title (“a...monitor”) and figure 1 (showing a smartphone with actual data coming in and life feedback to the participant) are still misleading. I suggest changing the title to better guide the reader and avoid any confusion in the first place (eg “A wireless ear EEG to monitor drowsiness” or “Hardware for a wireless ear EEG drowsiness monitor” “A wireless ear EEG to monitor drowsiness”). In addition, figure 1 should be removed or at least moved to supplemental material, as this figure illustrates the envisioned solution, which is still far beyond the empirically presented work.

Thank you for your feedback and the suggested changes to the title and figure 1. We have updated the paper title to “Wireless Ear EEG to Monitor Drowsiness”, updated figure 1 to remove the phone, and confirmed that ‘online classification’ or ‘closed loop system’ do not appear in any pertinent text.

The new figure 1 (shown below) shows a cleaned-up version of our presented system with an earpiece, a desktop machine, and offline notifications about what neural signals correspond to drowsiness.

Reviewer 4:

1. Transparent machine learning. The word transparent now only appears twice in the article, once in a heading and once in the summary. Given this, it would probably just be better to remove it all together.

Thank you for pointing this out, we have removed transparent from the remaining places in article.

2. Isolation. No information on isolation is included in the revised text. Instead, the reply refers to power-line interference which is not a concern. I would expect to see the Keysight unit being battery powered with current limiting resistors added. The E4980A can output up to 20mA in the no fault condition, and so a lot of care is needed, probably double signing off that the correct current range was selected. I think it's important to include these details to help ensure others follow suitable procedures.

Thank you for the clarification, we did indeed set a maximum current limit (a setting built into the LCR meter) to prevent any sensation or injury during the impedance monitoring process (0.5 mA). The text in Section II.a.iv. has been updated with the following:

All measurements were performed with an LCR meter (E4980 A, Keysight) powered by a wall outlet and arranged as a two-point probe where a single electrode is considered a single probe. The LCR meter was configured with a current limit of 0.5 mA to prevent sensation or injury. While the LCR meter is designed to achieve high accuracy (within 3%) even in the presence of powerline interference, electrode cables were shielded by ground wires to further minimize interference.

It is important to note that the LCR meter setup and powered in accordance with the user manual (powered by wall-outlet) which achieves accuracy within 3% according to the data sheets.

3. Correlated KSS scores. A lot of the reply focuses on physiological measures, which was not the question. For correlated KSS scores, it is good that a preliminary experiment was carried out. Maybe it would be simplest to include details of this in supplementary material. At present, in the modified text two references are used to support the choice. However [41] makes no reference to KSS, and [93] appears to be a mistake, that paper is on a very different topic. I've not gone to the effort of doing a detailed search, but I'm surprised no-one in the sleep research community has done an investigation into how KSS scores correlate with different frequencies of asking, that you could cite. At a minimum you could add a brief entry on this potential limitation to the discussion.

Thank you so much for pointing this out. The discrepancy and reference error are from a mistake that occurred when we were copying our updated text between the response letter and the manuscript. The references were meant to be to the following three texts:

[45] C. Schwendeman, R. Kaveh, and R. Muller, "Drowsiness Detection with Wireless, User-Generic, Dry Electrode Ear EEG," in *IEEE Engineering in Medicine and Biology Society, EMBS.*, 2022, pp. 9–12.

[90] J. Geiger Brown *et al.*, "Measuring subjective sleepiness at work in hospital nurses: validation of a modified delivery format of the Karolinska Sleepiness Scale," *Sleep and Breathing*, vol. 18, no. 4, pp. 731–739, Dec. 2014.

[91] K. Kaida *et al.*, "Validation of the Karolinska sleepiness scale against performance and EEG variables," *Clinical Neurophysiology*, vol. 117, no. 7, pp. 1574–1581, Jul. 2006.

The text (with fixed references is copied below):

Every five minutes, the user was prompted to enter a Likert item according to the Karolinska Sleepiness Scale (KSS). This scale is frequently used to evaluate subjective sleepiness and ranges from 0 = "extremely alert", to 10 = "extremely sleepy, fighting to stay awake"[90]. Queue intervals (60 seconds and 5 minutes) were selected based off initial experimentation and previous works that demonstrated a balance between minimizing disturbances and frequent datapoints [45], [91].

REVIEWERS' COMMENTS

Reviewer #2 (Remarks to the Author):

I thank the authors for the corrections, which improved the quality of the manuscript. I still have the following comments and concerns:

1) As already mentioned in the previous round of comments, I must insist to avoid using the terminology "this work showcases an end-to-end system ...", "... the proposed platform..." since an end-to-end system / a platform is not what the contributions of the paper is about. The paper presents novel electrodes and uses an existing data acquisition platform for collecting data from them. A careful reader would see these as overstatements, so I encourage the authors to kindly make these corrections.

2) Figure 10s shows a drop in sensitivity when subject 7 is included in the analysis. However, table supplementary 1 shows that subject 3 has some challenges in reaching high sensitivity. I suspect the drop on #subject=7 is because that's when you introduce in the analysis subject #3? Please elaborate further. Also, it would be better to report in Fig. 10s average metrics, i.e., for the $x=2$ you should report the average value among all possible combinations of 2 subjects, $x=3$ etc. This should reduce the sensitivity value for low number of subjects (instead of a perfect 100%)

3) the photo of the subject wearing the device should be put in the main manuscript, not in the supplementary information. If there is a concern of the total number of figures, Fig 1 does not add much to the discussion and in case could be dropped.

4) Table 2: the authors also use one wet electrode. So the table should be updated replacing "dry" with "dry+wet" (or similar)

First, the authors would like to express their gratitude to the editor and reviewers for providing further suggestions and constructive feedback to the authors. Below is the list of questions and comments raised by the reviewers alongside our responses. For the convenience of the reviewers both the modifications made to the manuscript and our responses below are shown in green.

Reviewer 2:

1) As already mentioned in the previous round of comments, I must insist to avoid using the terminology "this work showcases an end-to-end system ...", "... the proposed platform..." since an end-to-end system / a platform is not what the contribution of the paper is about. The paper presents novel electrodes and uses an existing data acquisition platform for collecting data from them. A careful reader would see these as overstatements, so I encourage the authors to kindly make these corrections.

To be more precise and clearer, we have updated the language in question to the following text:

Neural wearables can enable life-saving drowsiness and health monitoring for pilots and drivers. While existing in-cabin sensors may provide alerts, wearables can enable monitoring across more environments. Current neural wearables are promising but most require wet-electrodes and bulky electronics. This work showcases in-ear, dry-electrode earpieces used to monitor drowsiness with compact hardware. The employed system integrates additive-manufacturing for dry, user-generic earpieces, existing wireless electronics, and offline classification algorithms. Thirty-five hours of electrophysiological data were recorded across nine subjects performing drowsiness-inducing tasks. Three classifier models were trained with user-specific, leave-one-trial-out, and leave-one-user-out splits. The support-vector-machine classifier achieved an accuracy of 93.2% while evaluating users it has seen before and 93.3% when evaluating a never-before-seen user. These results demonstrate wireless, dry, user-generic earpieces used to classify drowsiness with comparable accuracies to existing state-of-the-art, wet electrode in-ear and scalp systems. Further, this work illustrates the feasibility of population-trained classification in future electrophysiological applications.

The abstract now specifies that the work's primary novelty is the in-ear EEG capable earpiece that is used to monitor drowsiness. The 'employed' system description in the following specifies that pre-existing hardware was used. The claim of a novel system has been removed and the word system only refers to how the novel earpiece was used to record data.

2) Figure 10s shows a drop in sensitivity when subject 7 is included in the analysis. However, table supplementary 1 shows that subject 3 has some challenges in reaching high sensitivity. I suspect the drop on #subject=7 is because that's when you introduce in the analysis subject #3? Please elaborate further. Also, it would be better to report in Fig. 10s average metrics, i.e., for the x=2 you should report the average value among all possible combinations of 2 subjects, x=3 etc. This should reduce the sensitivity value for low number of subjects (instead of a perfect 100%)

This is an excellent point. To run this analysis more holistically, we reran this experiment to calculate the average value for each number of subjects. Furthermore, we included all the data from each subject (as opposed to a single trial per each subject as was done previously). The new supplemental figure is shown below:

There are two notable changes, the average accuracy of 2 users has improved and is around 65 - 75%. This makes sense given the average 2 user case has significantly more training data now (average of 3 hours vs. 1 hour). The other notable change is that performance saturates at 6-7 subjects as was shown earlier. The accompanying text has been updated to reflect the new procedure and these conclusions.

3) the photo of the subject wearing the device should be put in the main manuscript, not in the supplementary information. If there is a concern of the total number of figures, Fig 1 does not add much to the discussion and in case could be dropped.

Thank you for the suggestion. We have now updated Figure 2 to have a photo of an earpiece inserted into an ear (shown below).

This is consistent with the precedent set by many existing nature portfolio works such as <https://doi.org/10.1038/s41551-023-01095-1>, <https://doi.org/10.1038/s41467-023-39814-6>, and <https://doi.org/10.1038/s41551-018-0323-x> that all place experimental photos in the supplement and not in the main body of the paper.

4) Table 2: the authors also use one wet electrode. So the table should be updated replacing "dry" with "dry+wet" (or similar)

This is a good point. Indeed, we use one wet electrode for the system ground. This should be included in the table and the pertinent section has been updated to specify we have a wet ground electrode. The updated table section is copied below:

	Format	In-ear	In-ear	In-ear	Scalp	Muse Headband	In-ear
Electrodes	Single/both ears	single	Single	Single	-	-	Both
	Sense/Ref Electrodes	Wet	Wet	Wet	Wet	Dry	Dry
	Ground Electrode	Wet	Wet	Wet	Wet	Dry	Wet
	# Channels	1	2	1	30	4	11
	Generic	Yes	Yes	Yes	Yes	Yes	Yes
	Assembly material	Metallic	Foam	Silicone	Metallic	Plastic	3D printed polymer
	Electrode	--	Ag/AgCl wire	Ag + Cu	--	Au	Au